# Bacterial RNA promotes proteostasis through inter-tissue communication in *C. elegans*

Emmanouil Kyriakakis [1] ✉, Chiara Medde [1], Danilo Ritz [1], Geoffrey Fucile[1,2], Alexander Schmidt [1] & Anne Spang [1] ✉

Life expectancy has been increasing over the last decades, which is not matched by an increase in healthspan. Besides genetic composition, environmental and nutritional factors influence both health- and lifespan. Diet is thought to be a major factor for healthy ageing. Here, we show that dietary RNA species improve proteostasis in *C. elegans*. Inherent bacterial-derived double stranded RNA reduces protein aggregation in a *C. elegans* muscle proteostasis model. This beneficial effect depends on low levels of systemic selective autophagy, the RNAi machinery in the germline, even when the RNA is delivered through ingestion in the intestine and the integrity of muscle cells. Our data suggest a requirement of inter-organ communication between the intestine, the germline and muscles. Our results demonstrate that bacterial-derived RNAs elicit a systemic response in *C. elegans*, which protects the animal from protein aggregation during ageing, which might extend healthspan.

Humans and all living organisms rely on nutrients for growth, reproduction, movement and survival, with key nutritional pathways being evolutionary conserved across species. It is generally accepted that the type and concentration of nutrients influence the healthspan and life expectancy of eukaryotes. However, it remains unclear what combination of nutrients is most beneficial. Over the years, *C. elegans* has been proven to be an important and reliable model for nutrient-dependent health and lifespan studies, with major discoveries being confirmed across species[1–5]. The influence of dietary restriction on longevity was first assessed in rodents as early as 1917, with *C. elegans* later emerging as a key model to explore the underlying mechanisms in greater depth. Today, its benefits are widely accepted in mammals and even humans[3,6–9]. Furthermore, pioneering studies in *C. elegans* have unveiled the important role of cellular protein homeostasis (or proteostasis) in diseases and ageing[10–15]. Since proteostasis deteriorates during ageing, finding ways to safeguard or even extend proteostasis emerges as a key concept to prevent, or at least ameliorate, age-associated diseases, such as cardiovascular disease, neurodegenerative diseases, late-onset neuromuscular disorders, sarcopenia and others. Ample scientific evidence suggests that specific dietary interventions provide a promising approach to maintain proteostasis and improve health during ageing.

To answer how diet and which dietary components influence cellular and organismal fitness and life expectancy in a reliable and expeditious way, we investigated *C. elegans* and its bacterial diet. *C. elegans* nematodes are reared on monoxenic bacterial cultures that are easy to grow and to genetically manipulate. Utilizing this simple, tractable animal model, we show that a mixed diet of two *E. coli* strains promotes *C. elegans* fitness. Importantly, we demonstrate that bacterially expressed ribonuclease 3 influences the accumulation of protein aggregates in *C. elegans* body-wall muscles via a cell-non-autonomous mechanism involving intestinal uptake of bacterial-derived RNA species, the RNAi machinery, selective autophagy and proper muscle function. We also show that communication across tissues and cell types, such as intestine, germline and neurons, plays an important role in the regulation of proteostasis in body-wall muscles. Overall, our findings suggest bacterial-derived dietary cues influence organismal fitness by eliciting a protective response during stress and reveal how diet-derived RNA species promote proteostasis in *C. elegans*.

[1]Biozentrum, University of Basel, Basel, Switzerland. [2]sciCORE Center for Scientific Computing and Center for Data Analytics, University of Basel, Basel, Switzerland. ✉e-mail: emmanouil.kyriakakis@unibas.ch; anne.spang@unibas.ch

## Results

### A mixed bacterial diet promotes *C. elegans* fitness

In the laboratory, *C. elegans* are usually reared on two different *E. coli* strains: OP50, an *E. coli* B strain, and HT115, a K-12-derived strain. It was previously shown that these strains differ in their metabolic and nutrient profile[16]. For example, OP50 leads to vitamin B12 deficiency in *C. elegans*[5,17,18]. We first investigated the effect of diet on organismal fitness and lifespan (Fig. 1a). As previously reported, we did not observe any considerable difference in lifespan between the OP50 and HT115 diets (Fig. 1b)[16]. However, worms fed on OP50 produced significantly higher number of progeny, which also developed faster than worms on the HT115 diet (Fig. 1c, d and Supplementary Fig. 1a). This beneficial effect came at the cost of reduced healthspan at advanced age, since a larger fraction of OP50-fed worms displayed impaired movement compared to their HT115-fed counterparts (Fig. 1e). These data indicate there were apparent benefits and trade-offs accompanying each diet. Hence, we reasoned that a mixture of both diets could exert beneficial effects. Indeed, the benefits of OP50 were still maintained even if it constituted only 10% of the diet, while the fitness in older worms was improved even beyond the level of feeding on HT115 alone (Fig. 1c–e, Supplementary Movies 1–4). Thus, bacterial diets differentially affect development, reproduction and fitness. Combining both diets also combined the benefits of each individual diet and improved the healthspan of *C. elegans*.

### Dietary cues protect from muscle proteotoxicity

Next, we aimed to uncover how diets affect fitness and healthspan. Motility is a fitness measure in *C. elegans* and is linked to the function of body-wall muscles[15,19,20]. Polyglutamine (polyQ) expansions have been used to assess cellular dysfunction in *C. elegans* body-wall muscles in response to proteotoxicity[15]. PolyQ40-YFP aggregate formation in body-wall muscles of worms can be used as a readout for proteostasis decline. Worms contained numerous polyQ-aggregates early in adulthood when reared on OP50, the number of which increased with age (Fig. 1f). In contrast, HT115-fed worms started to form aggregates much later in adulthood and to a lesser extent (Fig. 1f). The strong delay in aggregate formation in body-wall muscle cells might serve as indication for the increased mobility of aged animals on HT115 diet. These differences were not due to OP50's vitamin B12 deficiency (Supplementary Fig. 1b). As expected, polyQ24-expressing worms did not show any diet-dependent aggregate formation (Supplementary Fig. 2a). To corroborate the diet-dependent effect on proteostasis, we expressed amyloid-beta (Aβ1-42) in body-wall muscle cells, which has been previously shown to lead to paralysis in worms[21]. Again, HT115 led to a much later onset of paralysis compared to OP50 (Supplementary Fig. 2b). Mixed diets also improved the fitness of polyQ40-expressing animals, as we observed beneficial effects on the number of aggregates, number of progeny and development (Fig. 1g, h and Supplementary Fig. 2c, d). Thus, we detected diet-dependent proteostasis dysregulation in body-wall muscles, for which the polyQ40 *C. elegans* can serve as a sensor.

### Autophagy protects from diet-dependent protein aggregation

One explanation for the positive dietary effect of the HT115 bacterial diet could be the upregulation of cytoprotective mechanisms, such as autophagy. Autophagy positively influences health and lifespan by removing damaged organelles and protein aggregates[22]. Moreover, it has been shown that autophagy inhibits the accumulation of polyQ40 aggregates in *C. elegans* and protects from proteotoxicity[23]. Similarly, we found that knockdown of key autophagy factors increased the accumulation of polyQ40 aggregates on the HT115 diet (Fig. 2a, b). Moreover, the offspring of mothers with defective autophagy were developmentally arrested (Supplementary Fig. 3a), and this effect was dependent on the expression of polyQ repeats in

muscle cells (Supplementary Fig. 3b–d). Specifically, with shorter, non-toxic repeats like Q0 and polyQ24, no developmental defects were observed. However, at polyQ35, where toxicity begins, there was a noticeable delay, and with the highly toxic polyQ40, the worms arrested entirely. It is plausible that the proteotoxicity of polyQ40 in muscle cells triggers a signal under these conditions, which is inherited by the offspring. Moreover, these data indicate that HT115 might induce autophagy, which is beneficial for the animal. To test this possibility, we measured autophagy induction using *C. elegans* LC3 fused to GFP (LGG-1::GFP) in hypodermal seam cells and the relative autophagosomes to autolysosomes ratio using LC3 fused to GFP and mCherry (mCherry::GFP::LGG-1) in muscle cells, as read-out for systemic induction and in body-wall muscles, respectively. Autophagy was significantly induced by the HT115 diet compared to OP50 as measured by the number of autophagosomes present in hypodermal seam cells and suggested by the relative GFP/mCherry ratio in body-wall muscles (Fig. 2c–f). A high ratio shows a greater number of autophagosomes exist in the muscle cells compared to autolysosomes, suggesting a reduction of the turnover of autophagosomes in lysosomes. Protein aggregates are usually removed through a selective autophagy pathway termed aggrephagy. To test whether HT115 might induce aggrephagy, we knocked down E3 ligases implicated in the ubiquitination of aggrephagy clients, PELI-1 and CHN-1, the aggrephagy receptors SQST-1 (p62) and TLI-1 (Tollip/Cue) and the aggrephagy adapter WDFY-3 (Alfy)[24,25]. In all cases, we observed an increase in polyQ40 aggregate formation (Fig. 2g, h and Supplementary Fig. 3e, f), indicating that aggrephagy in *C. elegans* body-wall muscle cells is activated to prevent aggregate formation. This effect was not due to the presence of dsRNA itself, as knockdown of unrelated genes did not increase aggregate formation (Supplementary Figs. 3g and 14h). Similar results were obtained in a *tli-1* KO strain (Supplementary Fig. 3h, i). TLI-1 and SQST-1 appear to have at least partially overlapping functions, as the combined knockdown increased aggregate formation over the individual knockdowns (Fig. 2i). Our data so far indicate that the HT115 bacterial diet protects from protein aggregation by inducing autophagy, and that the OP50 diet cannot induce the response to similar levels.

### Innate immunity pathways are not activated by bacterial diets

The positive dietary effect on proteostasis could also be due to stimulation of the innate immune response in worms. Induction of innate immunity has been observed with live pathogenic bacteria[26,27]. To test this possibility, we fed polyQ40 worms with UV-killed OP50 and HT115. However, the diet-dependent aggregate formation remained unchanged (Supplementary Fig. 4a, b). We next tested whether a bacterial secreted factor could be responsible for the differences between OP50 and HT115 and would induce innate immunity, which was not the case, as the secretome of the bacteria had no beneficial effect on the worm proteostasis (Supplementary Fig. 4c, d). Finally, we tested the induction of innate immunity more directly with transcriptional reporters for several immunity response genes. While those genes were induced by a pathogenic strain of *Pseudomonas aeruginosa* (PA14), neither OP50 nor HT115 elicited a response under the conditions tested (Supplementary Fig. 5). Thus, it is unlikely that the innate immune response is a prominent driver of diet-dependent aggregate formation.

### Ribonuclease-dependent bacterial RNAs promote proteostasis

A key difference between OP50 and HT115 is that HT115 can be used for RNA silencing experiments by feeding, while OP50 cannot. HT115 lacks a functional ribonuclease 3 (*rnC*), which recognizes dsRNA species and cleaves them with high specificity to produce smaller dsRNA fragments[28,29]. Therefore, ectopically expressed dsRNA in HT115 is stable and can be transferred to worms when they feed on the bacteria. To explore whether the presence or absence of ribonuclease 3 is

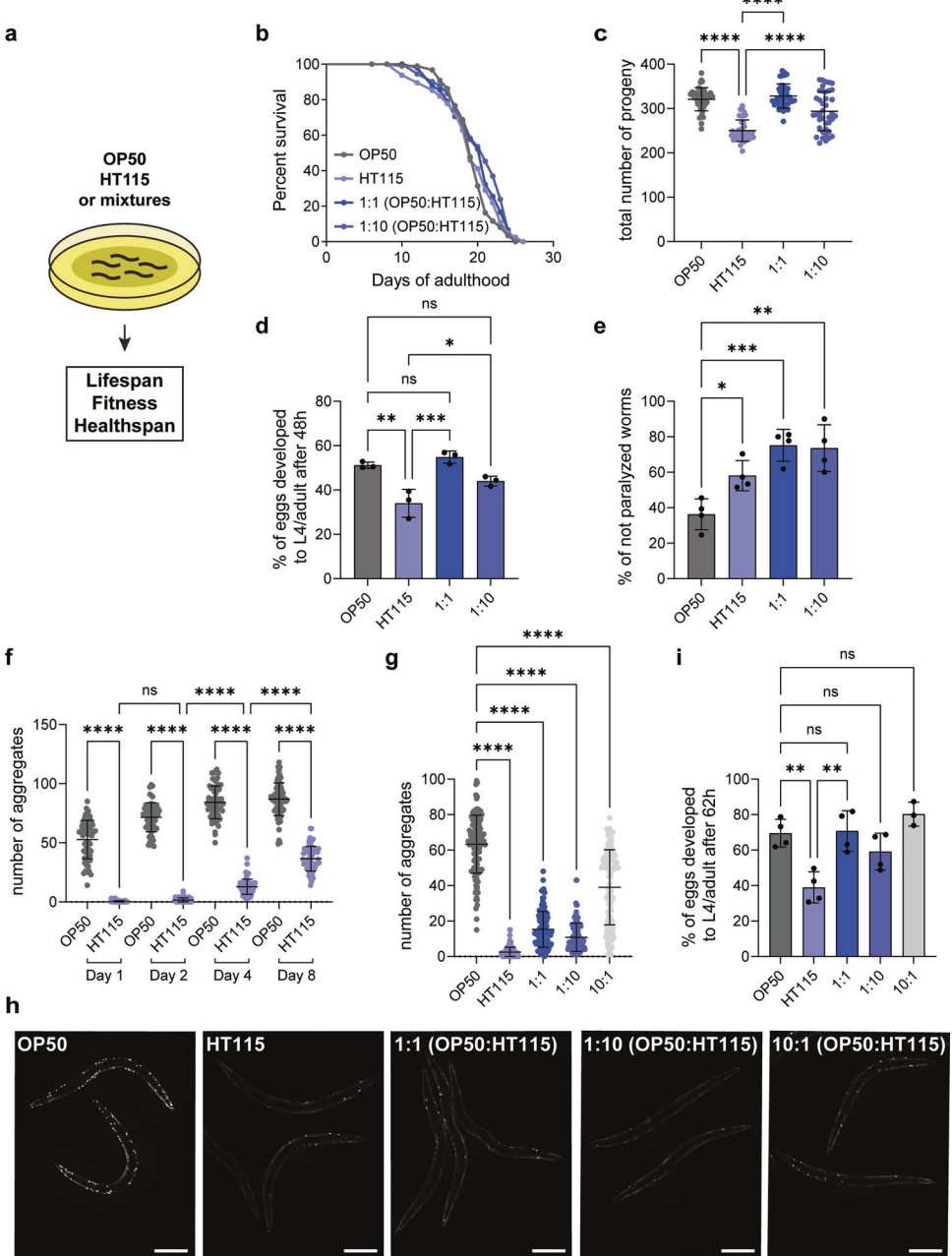

**Fig. 1 | Dietary mixtures promote fitness and proteostasis in the body wall muscles of *C. elegans*. a** Schematic representation of the main methodology. **b** Lifespan curves of wild-type (wt) worms cultured on OP50, HT115 and bacterial mixtures. Bacterial mixtures were used in 1:1 (50% OP50 and 50% HT115) and 1:10 (10% OP50 and 90% HT115) ratios. Statistical analysis for lifespan curves was performed with the Log-rank (Mantel–Cox) test and the summary is shown in Table 1. **c** Brood size of wt worms on different bacterial lawns. The total number of progeny per worm is shown. Number of worms, *n* = 40 for all conditions. OP50 vs. HT115 *P* < 0.0001, OP50 vs. 1:1 *P* = 0.819, OP50 vs. 1:10 *P* = 0.0009, HT115 vs. 1:1 *P* < 0.0001, HT115 vs. 1:10 *P* < 0.0001. **d** Developmental rate of wt worms on different bacterial lawns was measured as the percentage of eggs that developed into L4/adult stages 48 h after egg laying. Data are from three independent biological replicates. Number of worms, *n* = 751 (OP50), *n* = 619 (HT115), *n* = 914 (1:1), *n* = 899 (1:10). OP50 vs. HT115 *P* = 0.0022, OP50 vs. 1:1 *P* = 0.7773, OP50 vs. 1:10 *P* = 0.2029, HT115 vs. 1:10 *P* = 0.0493, HT115 vs. 1:1 *P* = 0.0006. **e** Percentage of not paralyzed (18–20 days old) wt worms on different bacterial diets was assessed. Data are from four independent biological replicates. Number of worms *n* = 25 (OP50), *n* = 32 (HT115), *n* = 27 (1:1), *n* = 30 (1:10). OP50 vs. HT115 *P* = 0.286, OP50 vs. 1:1 *P* = 0.0004, OP50 vs. 1:10 *P* = 0.0006. **f** Quantification of polyQ40::YFP

fluorescent foci of worms on OP50 or HT115 diet during ageing. The number of aggregates per worm is shown. Number of worms, *n* = 63 (OP50 Day 1), *n* = 71 (HT115 Day 1), *n* = 65 (OP50 Day 2), *n* = 68 (HT115 Day 2), *n* = 71 (OP50 Day 4), *n* = 69 (HT115 Day 4), *n* = 80 (OP50 Day 8), *n* = 68 (HT115 Day 8). HT115 Day 1 vs. HT115 Day 2 *P* = 0.9985, for the rest *P* < 0.0001. **g** Quantification of polyQ40::YFP fluorescent foci of 2-day-old worms on different bacterial diets. The number of aggregates per worm is shown. Number of worms, *n* = 117 (OP50), *n* = 128 (HT115), *n* = 112 (1:1), *n* = 128 (1:10), *n* = 105 (10:1). All *P* values shown <0.0001. **h** Representative images of 2-day-old polyQ40::YFP-expressing worms on different bacterial diets. Scale bars in all panels are 200 μm. **i** Developmental rate of polyQ40 expressing worms on different bacterial lawns was measured as the percentage of eggs that developed into L4/adult stages 62 h after egg laying. Data are from four independent biological replicates. Number of worms, *n* = 883 (OP50), *n* = 1030 (HT115), *n* = 1193 (1:1), *n* = 1058 (1:10), *n* = 932 (10:1). OP50 vs. HT115 *P* = 0.0027, OP50 vs. 1:1 *P* > 0.9999, OP50 vs. 1:10 *P* = 0.6560, OP50 vs. 10:1 *P* = 0.6878, HT115 vs. 1:1 *P* = 0.0019. All values represent mean ± SD from at least three independent experiments. One-way ANOVA with Sidak's multiple comparison test was used for all except for the lifespan. ns *P* > 0.05, **P* < 0.05, ***P* < 0.01, ****P* < 0.001, *****P* < 0.0001. Source data are provided as a Source Data file.

**Table 1 | Statistical analysis of survival assays**

| Genotype | Diet | Median survival ± SD (days of adulthood) | Total deaths/censored | P value |
|---|---|---|---|---|
| Wild-type (N₂) | OP50 | 20.67 ± 1.528 | 262/57 | – |
| | HT115 | 17.33 ± 1.528 | 259/59 | ns vs. OP50 |
| | 1:1 mix | 17.67 ± 2.887 | 262/59 | ns vs. OP50 or HT115 |
| | 1:10 mix | 18.00 ± 2.646 | 261/57 | ns vs. OP50 or HT115 |

Two sided, one-way ANOVA with Sidak's multiple comparison test was used.

important for the dietary difference in proteostasis in worms, we reared worms on the OP50(xu363) strain, in which *rnC* gene is mutated[30]. Strikingly, loss of *rnC* protected worms from polyQ40 aggregates (Fig. 3a, b), and these effects were not due to differences in polyQ40 transgene expression (Supplementary Fig. 6a–d). Re-introduction of wild-type *rnC*, but not of two different catalytically inactive mutant versions[29,31] into OP50(xu363) or HT115 led to aggregate formation in animals, confirming the detrimental effect of bacterial *rnC* on worm proteostasis (Fig. 3c, d, Supplementary Fig. 6e, f). The HT115 parental strain (W3310), which still carries a functional *rnC* gene, caused aggregate formation in muscle cells, validating that indeed the loss of bacterial *rnC* improves *C. elegans* proteostasis (Supplementary Fig. 6g, h). Finally, we wanted to test the general applicability of our findings. The *E. coli* strain Nissle 1917 is a non-pathogenic bacterial strain used in the clinical setting to treat gastrointestinal conditions due to its probiotic properties. We deleted *rnC* in Nissle 1917. PolyQ40 expressing worms on *rnC*-ablated Nissle 1917 diet showed a significantly reduced number of aggregates in their body-wall muscles compared to worms fed with WT Nissle 1917 (Fig. 3e, f). The effect was somewhat less dramatic than with OP50(xu363) or HT115, probably due to the strong tendency of Nissle 1917 to form biofilms. Nevertheless, taken together, our data establish that loss of bacterial ribonuclease 3 has a positive effect on proteostasis in *C. elegans* muscle cells.

We confirmed that, similar to HT115 diet data, autophagy, and more specifically aggrephagy, is required to maintain the low polyQ40 aggregation effect on OP50(xu363) diet (Supplementary Fig. 7a–c). Moreover, mutation of *rnC* in OP50 did not negatively influence brood-size or development, but positively influenced the muscle morphology, the pumping rate and the motility of worms in advanced age (Supplementary Figs. 7d–i and 8a–f). Thus, the OP50(xu363) diet combines the advantages of the OP50 and HT115 diets with respect to organismal fitness. Our data suggest that bacterially derived RNA species have a positive systemic effect on *C. elegans* proteostasis and fitness in general. To corroborate these findings, we injected RNA from bacteria or genomic DNA into the gonad of polyQ40-expressing *C. elegans*. Only bacteria-derived RNA reduced aggregate formation in muscle cells (Fig. 3g and Supplementary Fig. 9). The fact that injections of isolated RNA from all three bacterial strains had a similar effect on aggregate accumulation suggests that they all contain the effective dsRNA species. We assume that the size of the dsRNA species matters for the uptake and processing by the intestinal cells. Previous work showed that ingestion of long dsRNA is more effective than shorter pieces, but the length of dsRNA had no effect when injected into the gonad[32,33]. These data suggest that loss of one single bacterial gene, *rnC*, provides essential RNA species through the intestine that are critical for the reduction of aggregates in *C. elegans*.

## The RNAi machinery and the germline are required to promote proteostasis

*C. elegans*, like many other eukaryotes, has evolved a system to defend itself against foreign RNA species, the RNAi machinery[34]. Therefore, we tested whether bacterial-derived RNAs act through the RNAi machinery to promote proteostasis. To that end, we knocked down key

components of the RNAi pathway: the RNA transporters SID-1 and SID-2 and the Argonaute proteins RDE-1 and ERGO-1 using the HT115 diet. In all these cases, the number of aggregates in muscle cells was increased (Fig. 4a). A similar result was obtained upon silencing of the RNA-dependent RNA polymerase EGO-1, which is required for the systemic effect of RNAi in worms (Fig. 4a). These results were confirmed by mutants in RNAi pathway components (Fig. 4b–f), irrespective of the diet. In addition, silencing of *sid-1* induced paralysis of old worms (Supplementary Fig. 8a, b). Thus, our data suggest that bacterial-derived RNA is recognized by the RNAi machinery and that this machinery is linked to the motility of aged worms and the reduction of polyQ40 aggregation.

We have shown above that administration of bacterial RNA through the intestine by feeding and into the gonad by injection prevents aggregate formation in body-wall muscles. Therefore, we wondered whether the RNAi machinery needed to be active in both tissues. PPW-1 is a germline-specific Argonaute[35]. In the *ppw-1* mutant, the number of aggregates remains high across all three diets (Fig. 4f), indicating that even RNA delivery into the intestine requires the germline RNAi components for the protective effect. However, the germline RNAi machinery was not sufficient, because a *C. elegans* strain in which RNAi is only active in the germline still accumulated polyQ40 aggregates (Fig. 4g)[36]. Consistent with these findings, polyQ40 aggregates also formed in animals when the RNAi machinery was only active in the intestine (Fig. 4h)[37]. These results suggest that a functional germline is indispensable to block polyQ40 aggregate accumulation in the body-wall muscles of *C. elegans*. Moreover, communication across tissues involving intestinal cells, the germline and muscle cells, is most probably required for bacterially derived RNA species to protect from protein aggregates.

Next, we determined whether any or a specific RNA would elicit the beneficial effect. If it were a specific one, this bacterial RNA would affect the gene expression of a selective group of genes in the animal. We performed total RNA seq and smRNA seq on the three bacterial strains using both total RNA and a library enriched for small RNA molecules. On the total RNA level, HT115 and OP50 are distinct, with the majority of the variance captured by inter-strain differences. Only a minor fraction of the total variance is captured by the contrast of OP50 versus OP50(xu363) (Supplementary Fig. 10a). Among the set of 271 genes, which were consistently differentially expressed when contrasting HT115 versus OP50 and OP50(xu363) versus OP50 (Supplementary Fig. 10b), there is no significant enrichment of known Gene Ontology (GO) terms at any level. However, when considering the subset of 23 genes which are consistently up-regulated in HT115 and OP50(xu363) compared to OP50 there are enrichments in GO biological pathways, in particular related to RNA and protein metabolism (Supplementary Fig. 10c). RNA-Seq of the small RNA library similarly yielded few differences between the genotypes (Supplementary Fig. 10d). We employed de novo transcriptome assembly to increase sensitivity, however among the expressed contigs we identified only three which were differentially expressed between OP50 and OP50(xu363) and mapped to the *C. elegans* reference genome (Table 2). Administration of dsRNA based on these candidates did not affect the phenotype (Supplementary Fig. 10e). Thus, it is unlikely that the bacterial RNA silences a gene or subset of genes specifically, but

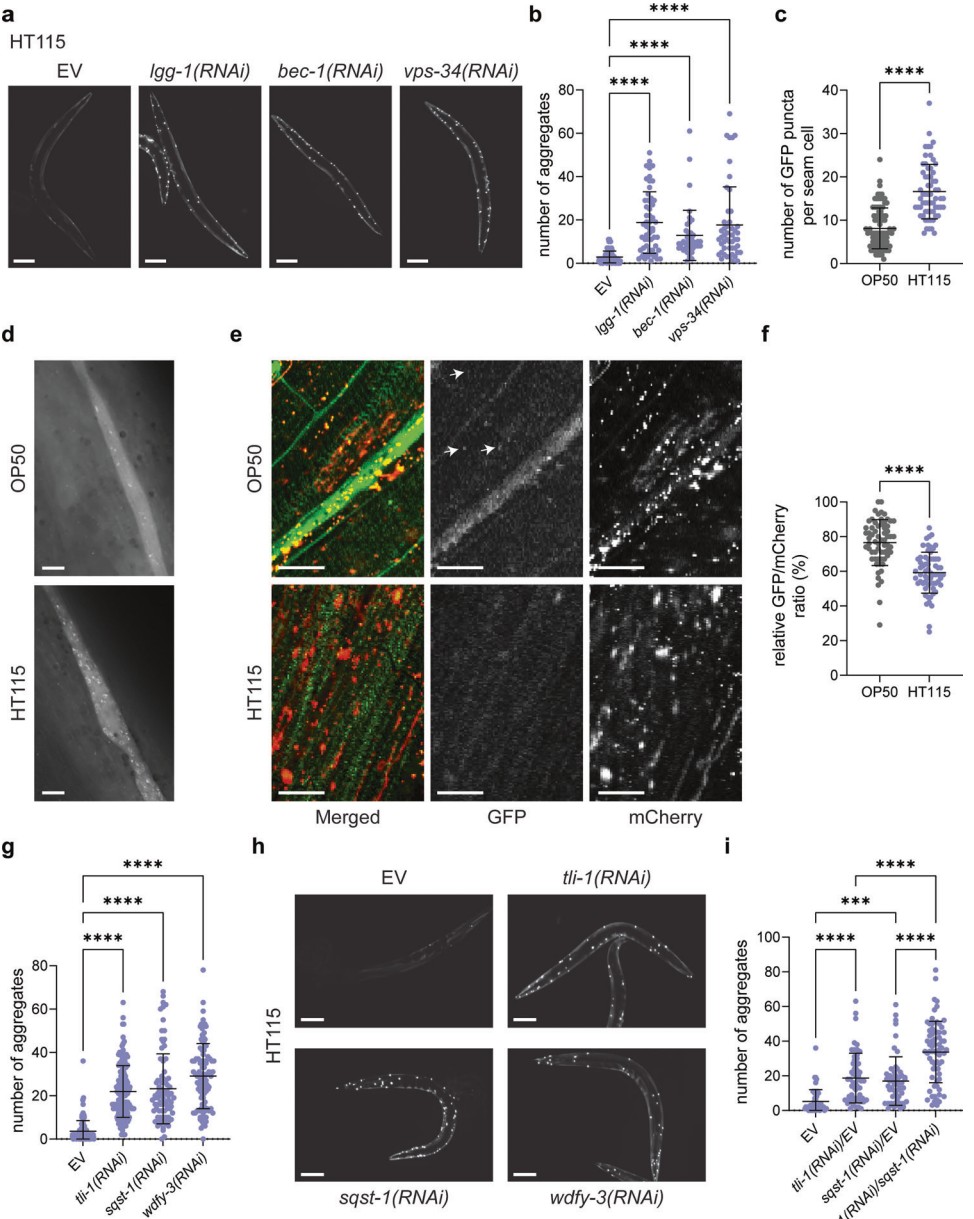

**Fig. 2 | Autophagy protects from protein aggregation in the body wall muscles of *C. elegans*. a** Representative images of 2-day-old polyQ40::YFP-expressing worms on HT115 diet, treated with empty vector (EV), *lgg-1(RNAi), bec-1(RNAi)* and *vps-34(RNAi)*. Scale bars in all panels are 100 μm. **b** Quantification of polyQ40::YFP fluorescent foci of 2-day-old worms on HT115, treated with empty vector (EV), *lgg-1(RNAi), bec-1(RNAi)* and *vps-34(RNAi)*. The number of aggregates per worm is shown. Number of worms, *n* = 78 (EV), *n* = 60 (*lgg-1(RNAi)*), *n* = 41 (*bec-1(RNAi)*), *n* = 49 (*vps-34(RNAi)*). All *P* values are <0.0001. **c** Quantification of GFP::LGG-1 positive puncta per seam cell of worms on OP50 and HT115. Number of seam cells, *n* = 93 (OP50), *n* = 63 (HT115). *P* < 0.0001. **d** Representative images of hypodermal seam cells, of p*lgg-1*GFP::LGG-1-expressing L4 animals on OP50 or HT115. Scale bar is 10 μm. **e** Representative images of body wall muscle cells, of 8-day-old p*lgg-1* mCherryGFP::LGG-1-expressing animals on OP50 or HT115. Scale bar is 10 μm. Arrows show representative mCherry and GFP-positive puncta (autophagosomes). Representative images from three independent experiments are shown. **f** Quantification of the relative GFP/mCherry LGG-1 positive puncta in muscle cells of worms reared on OP50 and HT115. *P* < 0.0001. **g** Quantification of polyQ40::YFP

fluorescent foci of 2-day-old worm on HT115, treated EV, *tli-1(RNAi), sqst-1(RNAi)* and *wdfy-3(RNAi)*. The number of aggregates per worm is shown. Number of worms, *n* = 142 (EV), *n* = 156 (*tli-1(RNAi)*), *n* = 85 (sqst-1(RNAi)), *n* = 98 (*wdfy-3(RNAi)*). All *P* values are <0.0001. **h** Representative images of 2-day-old polyQ40::YFP-expressing worms on HT115 diet, treated with EV, *tli-1(RNAi), sqst-1(RNAi)* and *wdfy-3(RNAi)*. Scale bars in all panels are 200 μm. **i** Quantification of polyQ40::YFP fluorescent foci of 2-day-old worms treated with EV, *tli-1(RNAi)* and *sqst-1(RNAi)* diluted with an equal amount of the EV and a mixture of equal amounts of *tli-1(RNAi)* and *sqst-1(RNAi)*. The number of aggregates per worm is shown. Number of worms, *n* = 51 (EV), *n* = 55 (*tli-1(RNAi)/EV*)), *n* = 44 (sqst-1(RNAi)/EV), *n* = 70 (*tli-1(RNAi)/sqst-1(RNAi)*). EV vs. *tli-1(RNAi)/EV P* < 0.0001, EV vs. *sqst-1(RNAi)/EV P* = 0.0001, *tli-1(RNAi)/EV* vs. *tli-1(RNAi)*/sqst-1(RNAi) *P* < 0.0001, sqst-1(RNAi)/EV vs. *tli-1(RNAi)/sqst-1(RNAi)* *P* < 0.0001. All values represent mean ± SD from at least three independent experiments. One-way ANOVA with Sidak's multiple comparison test (**b, g, i**) and Mann−Whitney *t*-test (**c, f**) were used. ***P* < 0.001, *****P* < 0.0001. Source data are provided as a Source Data file.

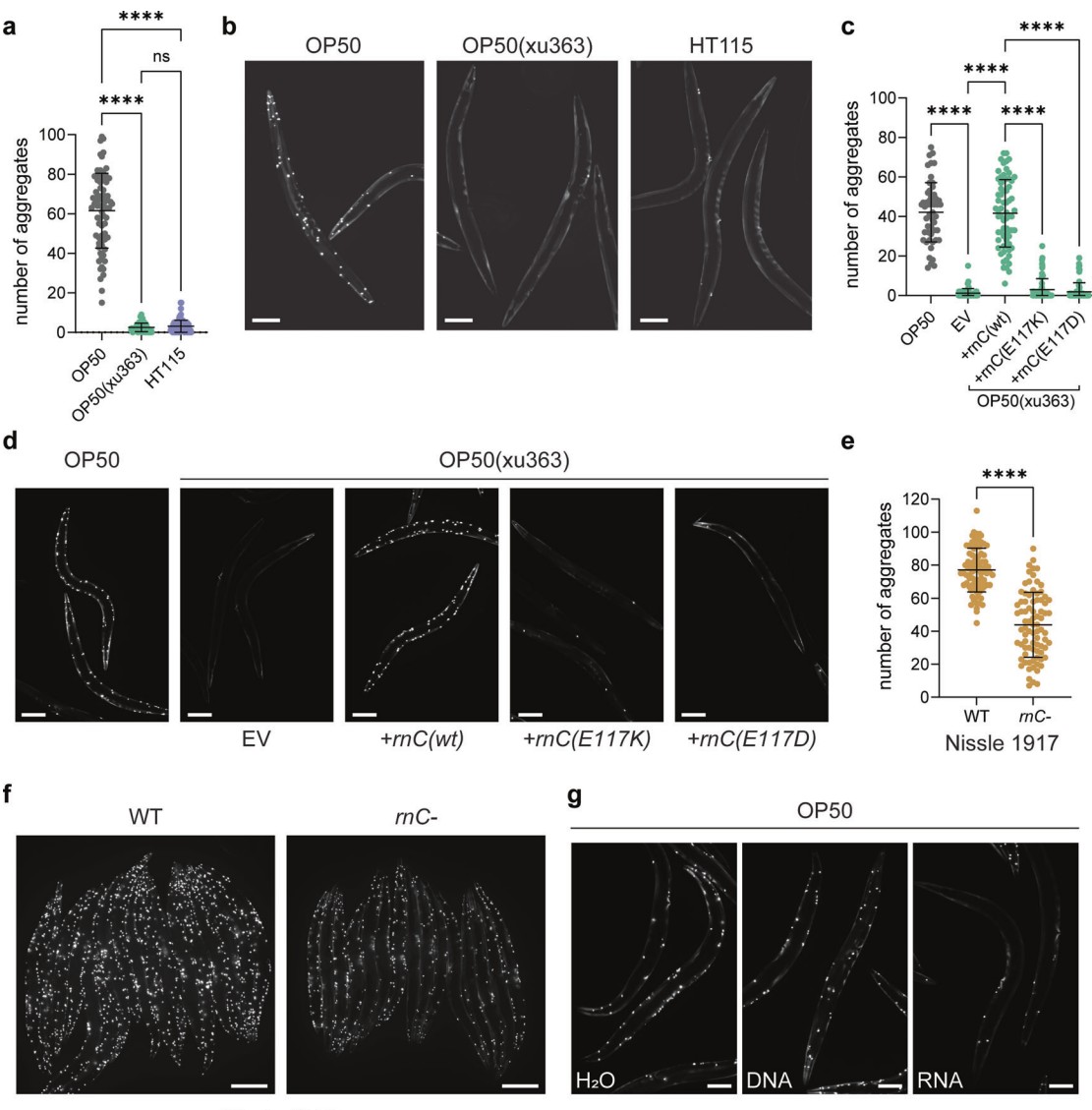

**Fig. 3 | Ribonuclease 3-dependent bacterial-RNA species protect from polyQ40 protein aggregation. a** Quantification of polyQ40::YFP fluorescent foci of 2-day-old worms on OP50, OP50(xu363) and HT115 bacterial diets. The number of aggregates per worm is shown. Number of worms, $n = 73$ (OP50), $n = 78$ (OP50(xu363)), $n = 82$ (HT115). Values represent mean ± SD from three independent experiments. One-way ANOVA with Sidak's multiple comparison test was used. OP50 vs. OP50(xu363) and OP50 vs. HT115 with $P < 0.0001$, OP50(xu363) vs. HT115 $P = 0.9798$, $P < 0.0001$. **b** Representative images of 2-day-old polyQ40::YFP-expressing worms on OP50, OP50(xu363) and HT115 bacterial diets. Scale bar is 100 μm. **c** Quantification of total polyQ40::YFP fluorescent foci of 2-day-old worms on OP50 and OP50(xu363) containing the empty vector (EV) or expressing the wt ribonuclease 3 (+rnC(wt)) and catalytically inactive ribonuclease 3 (+rnC(E117K), +rnC(E117D)). The number of aggregates per worm is shown. Number of worms, $n = 54$ (OP50), $n = 67$ (EV), $n = 68$ (+rnC(wt)), $n = 65$ (+rnC(E117K)), $n = 61$ (+rnC(E117D)). Values represent mean ± SD from three independent experiments. One-way ANOVA with Sidak's multiple comparison test was used. All with $P < 0.0001$. **d** Representative images of 2-day-old polyQ40::YFP-expressing worms on OP50 and OP50(xu363) expressing the wt (+rnC(wt)) or catalytically dead (+rnc(E117K) or +rnC(E117D)) ribonuclease 3. EV serves as the control vector. Scale bar is 100 μm. **e** Quantification of polyQ40::YFP fluorescent foci of 2-day-old worms on wt and ribonuclease-depleted (rnC-) Nissle 1917 *E. coli*. The number of aggregates per worm is shown. Number of worms, $n = 86$ (WT), $n = 82$ (rnC-). Values represent mean ± SD from three independent experiments. Mann–Whitney *t*-test was used. $P < 0.0001$. **f** Representative images of 2-day-old polyQ40::YFP-expressing worms on wt and ribonuclease 3-depleted (rnC-) Nissle 1917 *E. coli*. Scale bar is 200 μm. **g** Representative images of 2-day-old poly-Q40::YFP-expressing worms on OP50 diets, descendants of worms in which their gonads were injected with water (H$_2$O), DNA or HT115-derived RNA. Scale bar is 100 μm. Source data are provided as a Source Data file.

rather suggests that bacterial RNA elicits a low level of a more general stress response.

## Muscle contraction and sarcomere integrity are required for the beneficial effect

Nevertheless, the dietary bacterial RNA prompted a systemic response and, therefore, we determined the proteome of WT and polyQ40 expressing worms reared on OP50, OP50(xu363) or HT115 diets. We detected diet-dependent changes in the proteomes of both WT and polyQ40 expressing worms, most importantly also between OP50 and OP50(xu363) (Fig. 5a–d and Supplementary Fig. 11a–f). We focused on the beneficial effect in our proteostasis model. In total, we found 194 proteins to be significantly altered in polyQ40-expressing worms grown on HT115 or OP50(xu363) compared to OP50 (Fig. 5e). GO term analysis revealed an enrichment of GO terms related to sarcomere organization and muscle function (Supplementary Fig. 11g, Table 3). The functional protein association network uncovered 12 muscle-related proteins that were clustering (Fig. 5f). These proteins were not

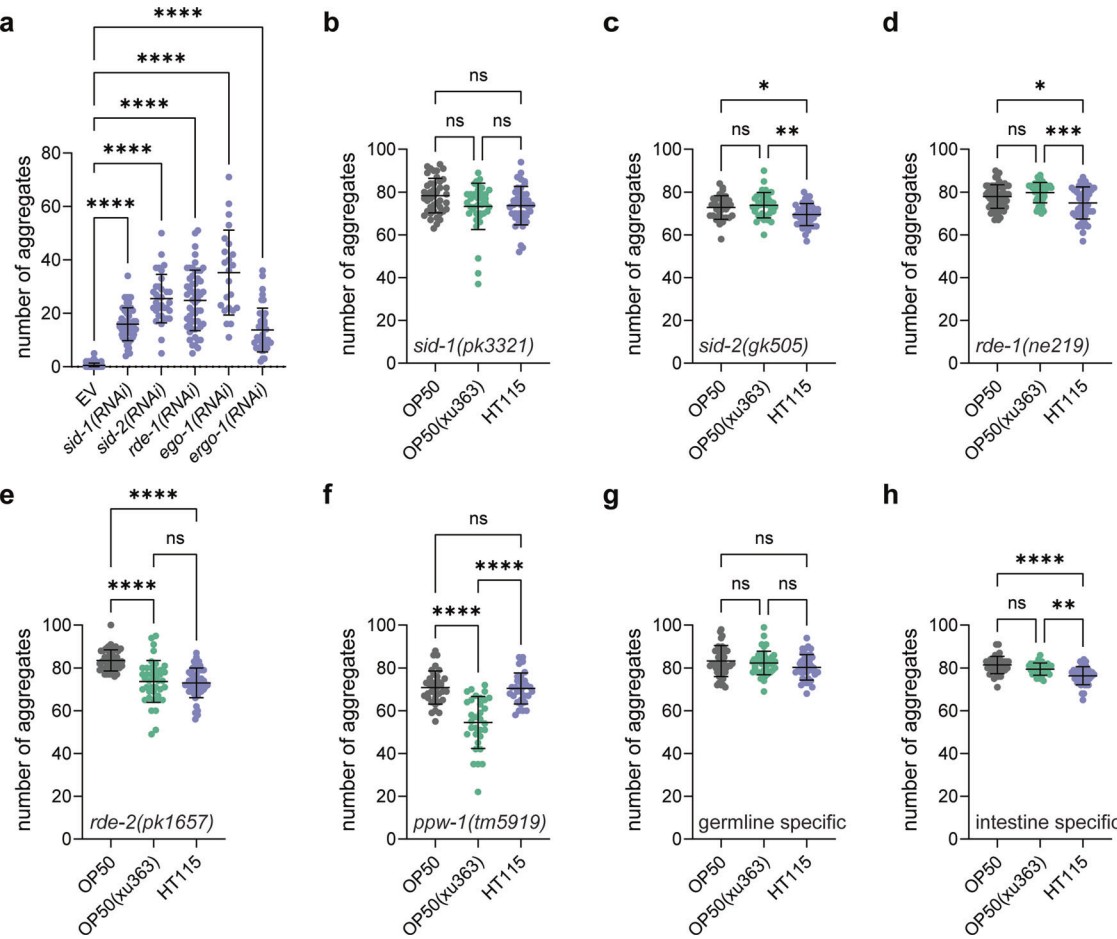

**Fig. 4 | The RNAi machinery and the germline are required to protect from protein aggregation. a** Quantification of polyQ40::YFP fluorescent foci of 2-day-old worms on HT115, treated with empty vector (EV), *sid-1(RNAi), sid-2(RNAi), rde-1(RNAi), ego-1(RNAi)* and *ergo-1(RNAi)*. The number of aggregates per worm is shown. Number of worms, *n* = 91 (EV), *n* = 51 (*sid-1(RNAi)*), *n* = 34 (*sid-2(RNAi)*), *n* = 54 (*rde-1(RNAi)*), *n* = 23 (*ego-1(RNAi)*), *n* = 41 (*ergo-1(RNAi)*). All with *P* < 0.0001. **b–f** Quantification of polyQ40::YFP fluorescent foci of 2-day-old worms on OP50, OP50(xu363) and HT115. The number of aggregates per worm is shown in *sid-1(pk3321)*, number of worms, *n* = 43 (OP50), *n* = 38 (OP50(xu363)), *n* = 41 (HT115). OP50 vs. OP50(xu363) with *P* = 0.535, OP50 vs. HT115 *P* = 0.0724, OP50(xu363) vs. HT115 *P* = 0.9978 (**b**), *sid-2(gk505)*, number of worms, *n* = 32 (OP50), *n* = 34 (OP50(xu363)), *n* = 42 (HT115), OP50 vs. OP50(xu363) with *P* = 0.8177, OP50 vs. HT115 *P* = 0.0346, OP50(xu363) vs. HT115 *P* = 0.0024 (**c**), *rde-1(ne219)*, number of worms, *n* = 65 (OP50), *n* = 55 (OP50(xu363)), *n* = 47 (HT115) with OP50 vs. OP50(xu363) *P* = 0.2516, OP50 vs. HT115 *P* = 0.0263, OP50(xu363) vs. HT115 *P* = 0.0002 (**d**), *rde-2(pk1657)*, number of worms, *n* = 43 (OP50), *n* = 41

(OP50(xu363)), *n* = 46 (HT115) with OP50 vs. OP50(xu363) *P* < 0.0001, OP50 vs. HT115 *P* < 0.0001, OP50(xu363) vs. HT115 P = 0.9642 (**e**) and *ppw-1(tm5919)*, number of worms, *n* = 35 (OP50), *n* = 32 (OP50(xu363)), *n* = 31 (HT115) with OP50 vs. OP50(xu363) *P* < 0.0001, OP50 vs. HT115 *P* = 0.9965, OP50(xu363) vs. HT115 *P* < 0.0001 (**f**) strains. **g** Germline, number of worms, *n* = 45 (OP50), *n* = 44 (OP50(xu363)), *n* = 39 (HT115) with OP50 vs. OP50(xu363) *P* = 0.8742, OP50 vs. HT115 *P* = 0.0958, OP50(xu363) vs. HT115 *P* = 0.3607 and **h** intestine number of worms, *n* = 39 (OP50), *n* = 35 (OP50(xu363)), *n* = 41 (HT115) with OP50 vs. OP50(xu363) *P* = 0.0889, OP50 vs. HT115 *P* < 0.0001, OP50(xu363) vs. HT115 P = 0.0016, RNAi-specific mutant strains were used to quantify polyQ40::YFP fluorescent foci of 2-day-old worms of OP50, OP50(xu363) and HT115. The number of aggregates per worm is shown. Values represent mean ± SD from three independent experiments. One-way ANOVA with Sidak's multiple comparison test was used. ns *P* > 0.05, \**P* < 0.05, \*\**P* < 0.01, \*\*\**P* < 0.001, \*\*\*\**P* < 0.0001. Source data are provided as a Source Data file.

significantly altered in WT worms (Fig. 5a–d and Supplementary Fig. 11a–f), indicating that the bacterial-derived RNA can elicit a context-dependent response and increase muscle function. However, we cannot exclude the possibility that these gene expression changes are merely a consequence of reduced polyQ40 aggregation, rather than a causative factor.

To distinguish between these hypotheses, we performed knockout and knockdown experiments on key muscle genes. Lesion of two giant sarcomeric proteins, UNC-22 (twitchin) and UNC-89 (obscurin), or UNC-27 (the ortholog of troponin I) increased the number of protein aggregates (Fig. 5g–i and Supplementary Fig. 13a–c), suggesting that the observed changes are associated with the induction of aggregation rather than being a consequence of it. Sarcomere, the basic unit of muscles, contracts following calcium influx. Disrupting calcium influx in the body-wall muscles by

silencing *unc-2, egl-19* or *cca-1*, which encode subunits of three voltage-dependent calcium channels, increased the number of aggregates compared to the control (Fig. 5j and Supplementary Fig. 13d–f). The voltage-dependent calcium channels open when muscle membranes depolarize upon neuroendocrine stimulation. This stimulation could either happen through neurotransmitter release or neuropeptide secretion and processing[38–40]. Mutants defective in the neurotransmitter release and acetylcholine synthesis (*unc-13* and *cha-1 cho-1*), but not in neuropeptide secretion and processing (*unc-31* and *egl-21*), lost the diet-dependent protective effect from polyQ40 aggregate formation in muscle cells (Fig. 5k, l and Supplementary Fig. 14a–f). Likewise, blocking neurotransmission through mutation or silencing of the postsynaptic acetylcholine receptors UNC-38 and UNC-29 caused aggregate formation independent of the diet (Fig. 5m, n and Supplementary Fig. 14g). In

**Table 2 | Sequences of the three expressed contigs, which are differentially expressed between OP50 and OP50(xu363)**

>TRINITY_DN17059_c0_g1_i1 len = 161 path = [0:0–160]
ACCCCATCGTAATTAATCGGCTTCAACGGAAGCTGGGCTCTGATGAATCCCCTAATGATTTTGGTAAAAATCATTAAGTTAAGGTGGATA-
CACATCTTGTCATATGATCTCGGGAAAAGCGTTGGTGACCAAAGGTGCCTTTTATCATCACTTTAAAAATA

>TRINITY_DN17136_c0_g1_i1 len = 596 path = [0:0–595]
TTTTTAGCGTTTATATCTGAAGGTTGGTTAGTTTTCCCTGTTTTAATTTTATTGGCTGGTGGTGGGATCGCTTTACCTGCATTA-
CAGGGAGTGATGTCTATCCAAACAAAGAGTCATCAGCAAGGTGCTTTA-
CAGGGATTATTGGTGAGCCTTACCAATGCAACCGGTGTTATTGGCCCATTACTGTTTGCTGTTATTTA-
TAATCATTCACTACCAATTTGGGATGGCTGGATTTGGATTATTGGTTAGCGTTTTACTGTATTATTATCCTGCTATCGATGACCTT-
CATGTTAACCCCTCAAGCTCAGGGGAGTAAACAGGAGACAAGTGCTTAGTTATTCGTCACCAAATGATGTTATTCCGCGAAATA-
TAATGACCCTCTTGATAACCCAAGAGGGCATTTTTTACGATAAAGAAGATTTAGCTTCAAA-
TAAAACCTATCTATTTTATTTATCTTTCAAGCTCAATAAAAAGCCGCGGTAAATAGCAA-
TAAATTGGCCTTTTTTATCGGCAAGCTCTTTTAGGTTTTTCGCATGTATTGCGATATGCATAAACCAGCCATTGAGTAAGTTTTTAAGCACATCATCATCATAAG

>TRINITY_DN3492_c0_g1_i1 len = 173 path = [0:0–172]
TCAGCGCAATTGATAGGCCAAATTCCCGCAACGGTGTGGGTGCTATTTACCGAAAATCGTTTTGGATGGAA-
TAGCATGATGGTTGGCTTTTCATTAGCGGGTCTTGGTCTTTTACACTCAGTATTCCAAGCCTTTGTGGCAGGAAGAATAGCCACTAAATGGGGCGAAAAAAC

contrast, silencing acetylcholine receptors in motor or sensory neurons or the homomeric ACR-16 in body-wall muscles had no effect (Supplementary Fig. 14h). These results provide evidence for the requirement of inter-tissue communication (Supplementary Fig. 14i) to maintain functional body-wall muscles and protect from protein aggregation. Moreover, we find that ribonuclease 3-deficient bacteria increased the levels of proteins required for muscle function. Reducing or abrogating the function of these proteins reversed the positive effect of the diet, and aggregates accumulated.

## Discussion

Here, we provide evidence that atypical dietary components, in particular bacterial-derived RNA species, regulate proteostasis and promote organismal health in *C. elegans*. Bacterial RNA species are taken up and processed by *C. elegans* intestinal cells and the germline to promote muscle function and protect from toxic protein aggregates *via* a mechanism that requires the RNAi pathway and autophagy induction. For the beneficial effect, inter-organ communication of the intestine, the germline and muscles is required (Supplementary Fig. 15). Whether neuronal input acts in parallel or is also part of this communication system remains to be determined. These results suggest that loss of proteostasis is not solely an internal cellular problem, but non-cell autonomous mechanisms are likewise important.

It has been shown previously that bacteria-derived RNAs can specifically affect gene expression of a subset of genes and behavior in *C. elegans*[41,42]. In a groundbreaking study, it was shown that pathogen-derived P11 sRNA is taken up by *C. elegans* and processed through the canonical RNAi pathway, to downregulate specifically the *maco-1* gene to initiate pathogen avoidance[41]. The mechanism we uncovered in this study is different, as we did not identify any specific bacterial RNA that would directly affect gene expression by acting as siRNA or by influencing transcription in *C. elegans*. These findings are mirrored by the proteome analysis in *C. elegans*, which did not reveal significant upregulation of major stress-responsive mechanisms nor a specific protein target. A major difference between our and previous studies is that we use a proteostasis model, which might already challenge the animals and therefore enables them to better and faster adapt to stressful environments. Accordingly, our proteomic analyses revealed an increase in the concentration of proteins required for muscle function, specifically in the muscle proteostasis model and not in the WT. Nevertheless, we also observed a positive effect on body-wall muscle morphology and pharyngeal muscle function in wild-type animals, suggesting a benefit not only in proteotoxically challenged animals. Bacterial RNA(s) induced the expression of muscle-specific genes to confer protection under stress conditions, suggesting that this trans-kingdom mechanism serves as a response, which is particularly beneficial during adverse

and pathological conditions, indicating that organ-specific responses could be triggered through dietary RNA and further supports the existence of broader, systemic effects. In support of this notion, we found systemic induction of selective autophagy. The relationship between proteostasis and autophagy is complex and context-dependent. Moderate levels of autophagy are considered beneficial for cellular health. In our proteostasis model, autophagy is induced by the diet-derived RNA species, and this is sufficient to promote proteostasis. Overall, the distinct response between control and proteotoxically challenged worms suggests an adaptive mechanism that depends on bacterial RNA and also the homeostatic state of cells and the whole organism. We propose a model in which the diet-derived RNA species elicit a basal stress response that would prime the organism to deal better with the onset of protein aggregation in our proteostasis model and therefore reduce and delay aggregate formation in body-wall muscle cells. The stress response remains at a low level, which we assume is sufficient to trigger proteoprotective mechanisms on the cellular level and thereby reduce aggregate formation. This delay in aggregate formation may be the underlying cause of the increase in healthspan in our proteostasis model. The low level of stress induction is supported by our findings that we did not observe any significant upregulation of autophagic or other stress-response related proteins in our *C. elegans* proteomics analysis when the animals were fed the different bacterial diets.

It has been previously shown that diet may promote proteostasis and lifespan and protect from neurodegeneration in *C. elegans*[1,2,43,44]. The transcriptional responses of *C. elegans* highly depend on the bacterial diet, and different diets direct unique transcriptional signatures[2,45]. In this study, we demonstrate that the transcriptional changes between the OP50 and HT115 diets are not responsible for the promotion of proteostasis and protection from toxic protein aggregates in *C. elegans*. However, the deletion of a single bacterial gene (*rnC*) in several distinct bacterial strains (OP50, HT115, Nissle1917) resulted in minimal transcriptional changes between the bacterial strains but significant changes in the *C. elegans* proteostasis model. Thus, the accumulation of dsRNA species that cannot be processed in the absence of ribonuclease 3 (*rnC*) is sufficient for the proteoprotective effects observed in *C. elegans*.

In this study, we move beyond the strict definition of nutrients and we identified non-traditional components, such as RNA species, that promote proteostasis and fitness. Bacteria do not behave solely as a nutrient source and this interspecies model may be relevant in understanding the relationship between humans and their microbiome and how it impacts physiology and disease.

In particular, in the advent of RNA as therapeutics, it is conceivable that dietary small RNA will prove useful as an intervention to extend healthspan in humans.

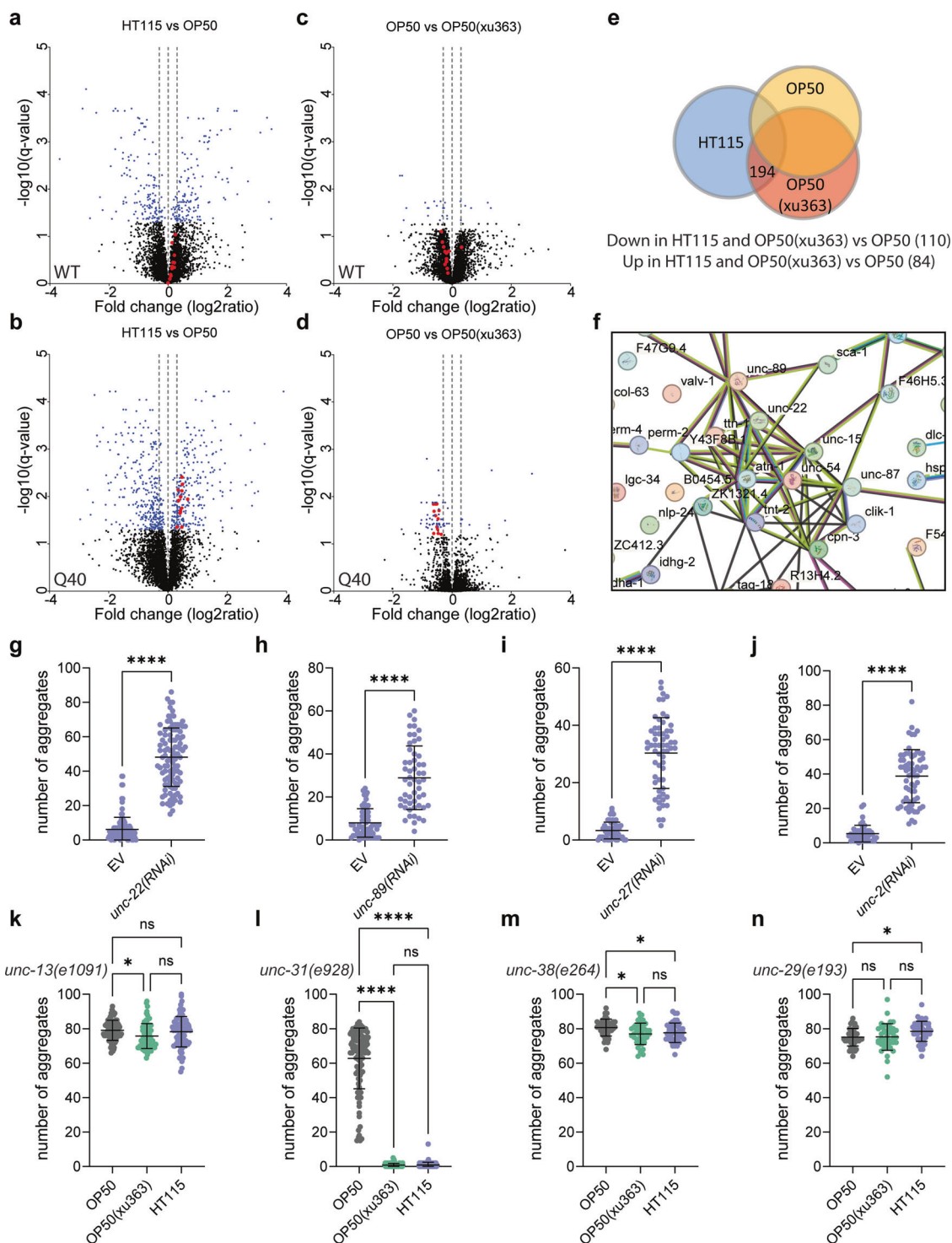

## Methods

### Nematode strains and growth conditions

Standard rearing conditions were used for maintaining *C. elegans* strains. All experiments were performed at 20 °C on nematode growth media (NGMs) agar supplemented with *Escherichia coli* (OP50, OP50(xu363), HT115, W3310, Nissle 1917, or mixtures of OP50 and HT115) unless otherwise stated. All bacterial strains were carrying the empty vector (EV) plasmid (pL4440), which served as the control for RNAi experiments but also for selection purposes, unless otherwise indicated. For RNAi experiments, worms were placed on NGM plates seeded with IPTG-induced HT115(DE3) or OP50(xu363) bacteria transformed with the gene-specific RNAi construct. OP50(xu363) is an

OP50-derived RNAi-competent strain, and HT115 is an RNAi-competent strain that derives from the W3310 strain. All bacteria grew at similar rates, and we ensured an excess of food for the worms during each experiment and for each diet. For *egl-19(RNAi)* experiments, the bacterial cultures were diluted 10 times with the EV plasmid to minimize developmental defects and sterility. Clones of interest were obtained from the Ahringer RNAi bacterial library or generated in the lab. The following nematode strains were used in the study: N2: wild-type Bristol isolate, AM141: rmIs133 [punc-54Q40::YFP], AM138: rmIs130 [unc-54p::Q24::YFP], AM140: rmIs132 [unc-54p::Q35::YFP], CL4176: smg-1(cc546) I; dvIs27[myo-3p::A-Beta (1-42)::let-851 3'UTR) + rol-6(su1006)], MAH14: daf-2(e1370) III; adIs2122 [lgg-1::GFP + rol-

**Fig. 5 | Muscle function and neurotransmission protect against the accumulation of protein aggregates. a**–**d** Volcano plots of total quantified proteins showing significant increase or decrease in content in WT (**a**, **c**) and polyQ40-expressing (**b**, **d**) strains on OP50, OP50(xu363) or HT115 bacteria. In blue are the proteins with q values less than 0.05. UNC-89, UNC-22, TTN-1, UNC-15, ATN-1, UNC-54, UNC-87, ZK1321.4, Y43F8B.1, TNT-2, CPN-3, and CLIK-1 proteins from the STRING analysis are shown in red. Horizontal dotted lines are at −0.3, 0, and +0.3 fold change. **e** Venn diagram illustrating the intersections among proteins of worms grown on OP50, OP50(xu363) or HT115 bacteria. 194 proteins were significantly altered between OP50 and both HT115 or OP50(xu363)-fed worms. **f** STRING functional protein association network analysis of the upregulated proteins is shown. A cluster of 12 muscle-related proteins is formed. **g**–**j** Quantification of polyQ40::YFP fluorescent foci of 2-day-old worms on HT115, treated with empty vector (EV), *unc-22(RNAi)* (**g**), *unc-89(RNAi)* (**h**), *unc-27(RNAi)* (**i**) and *unc-2(RNAi)* (**j**). Number of worms, $n = 118$ (EV), $n = 104$ (*unc-22(RNAi)*) with $P < 0.0001$ (**g**), $n = 63$ (EV), $n = 54$ (*unc-89(RNAi)*) with $P < 0.0001$ (**h**), $n = 49$ (EV),

$n = 61$ (*unc-27(RNAi)*) with $P < 0.0001$ (**i**), $n = 45$ (EV), $n = 64$ (*unc-2(RNAi)*) with $P < 0.0001$ (**j**). **k**–**n** Quantification of polyQ40::YFP fluorescent foci of 2-day-old worms on OP50, OP50(xu363) and HT115. The number of aggregates per worm is shown in *unc-13(e1091)* (**k**), *unc-31(e928)* (**l**), *unc-38(e264)* (**m**) and *unc-29(e193)* (**n**) mutant strains. Number of worms, $n = 89$ (OP50), $n = 79$ (OP50(xu363)), $n = 89$ (HT115) with OP50 vs. OP50(xu363) $P = 0.0113$, OP50 vs. HT115 $P = 0.8484$, OP50(xu363) vs. HT115 $P = 0.804$ (**k**), $n = 100$ (OP50), $n = 108$ (OP50(xu363)), $n = 104$ (HT115) with OP50 vs. OP50(xu363) $P < 0.0001$, OP50 vs. HT115 $P < 0.0001$, OP50(xu363) vs. HT115 $P = 0.9996$ (**l**), $n = 42$ (OP50), $n = 42$ (OP50(xu363)), $n = 42$ (HT115) with OP50 vs. OP50(xu363) $P = 0.0106$, OP50 vs. HT115 $P = 0.0490$, OP50(xu363) vs. HT115 $P = 0.9298$ (**m**), $n = 43$ (OP50), $n = 41$ (OP50(xu363)), $n = 45$ (HT115) with OP50 vs. OP50(xu363) $P = 0.9996$, OP50 vs. HT115 $P = 0.0362$, OP50(xu363) vs. HT115 $P = 0.0501$ (**n**). Values represent mean ± SD from three independent experiments. Mann–Whitney *t*-test (**g**–**j**) or one-way ANOVA with Sidak's multiple comparison test (**k**–**n**) were used. ns $P > 0.05$, *$P < 0.05$, ****$P < 0.0001$. Source data are provided as a Source Data file.

**Table 3 | Key proteins that are similarly altered by OP50(xu363) and HT115 but differently by OP50 during proteotoxic stress**

| Protein name | Function |
| --- | --- |
| UNC-89 | Obscurin, required for normal muscle cell architecture and function |
| UNC-22 | Twitchin kinase, regulates sarcomere organization |
| TTN-1 | Titin, enables actin filament and myosin binding activity |
| UNC-15 | Paramyosin, structural component of the muscle thick filament |
| ATN-1 | Actinin, contributes to muscle structure by anchoring actin filaments to the Z-disk |
| UNC-54 | Myosin, motor protein, drives muscle contraction by interacting with actin filaments |
| UNC-87 | Calponin-like, participates in muscle structure and sarcomere assembly |
| CPN-3 | Calponin, plays a role in regulating muscle contraction and actin–myosin interaction |
| CLIK-1 | Calponin-like, predicted to be involved in actin filament organization |
| TNT-2 | Troponin T, predicted to enable tropomyosin binding activity |
| ZK1321.4 | Predicted to enable actin binding activity and muscle alpha-actinin binding activity |
| Y43F8B.1 | Predicted to be involved in muscle development |

6(su1006)], RB1473: tli-1(ok1724) (6 times outcrossed), VP303: rde-1(ne219) V; kbIs7 [nhx-2p::rde-1 + rol-6(su1006)], NR350: rde-1(ne219) V; kzIs20 [hlh-1p::rde-1 + sur-5p::NLS::GFP], KP2018: egl-21(n476) IV, DA509: unc-31(e928) IV, CB1091: unc-13(e1091) I, ppw-1(tm5919), DCL569: mkcSi13 [sun-1p::rde-1::sun-1 3′UTR + unc-119(+)] II; rde-1(mkc36) V, NL3321: sid-1(pk3321) V, WM27: rde-1(ne219) V, NL3531: rde-2(pk1657) I, VC1119: dyf-2&ZK520.2(gk505) III, CB193: unc-29(e193) I, CB904: unc-38(e264) I, RM1743: cha-1(md39) cho-1(tm373) IV, AY101: acIs101 [F35E12.5p::GFP + rol-6(su1006)], AU133: agIs17 [myo-2p::mCherry + irg-1p::GFP] IV, AU306: agIs44 [Pirg-4::GFP::unc-54–3′ UTR; Pmyo-2::mCherry], MAH19: rrf-1(pk1417) I; myo-3(st386)V; stEx30 [myo-3p::GFP::myo-3 + rol-6(su1006)], MAH215: sqIs11 [lgg-1p::mCherry::GFP::lgg-1 + rol-6]. To generate double mutants, AM141 males were mated to hermaphrodites carrying the mutation of interest. The presence of the respective mutations was checked phenotypically or by genotyping.

### Constructs generated

For the construction of *tli-1(RNAi)* plasmid, the following primers (Microsynth) were used: 5′-TCTAGAAACCAAAACAAATACTGATCTTC CGT-3′ (FW) (with XbaI restriction site) and 5′-ACCGGTCTCTCGGC TGCTGCTGTCATCT-3′ (RV) (with AgeI restriction site). The amplified *tli-1* genomic region was ligated into the pL4440 vector upon digestion with XbaI and AgeI. For the construction of *rnC*(wt) expression plasmid, the following primers(Microsynth) were used: 5′-CCTGTGGATCC ATGAACCCCATCGTAATTAATCG-3′ (FW) (with BamHI restriction site) and 5′-CCTGTCAGCTGTCATTCCAGCTCCAGTTTTTTC-3′ (RV) (with PvuII restriction site). The amplified *rnC* region was ligated into the pL4440 vector upon BamHI and PvuII digestion, which leaves only one

T7 promoter. For the construction of *rnC*(E117D) expression plasmid, the following primers (Microsynth) were used: 5′-CCTGTGGATCCATG AACCCCATCGTAATTAATCG-3′ (FW) (with BamHI restriction site) and 5′-TAATGCATCGACGGTGTCGGCGA-3′ (RV) and also the 5′-CCTGT CAGCTGTCATTCCAGCTCCAGTTTTTTC-3′ (RV) (with PvuII restriction site) and 5′-TCGATGCATTAATTGGTGGCGTATT-3′. The two amplified regions were combined by fusion PCR using the following primers (Microsynth): CCTGTGGATCCATGAACCCCATCGTAAT-TAATCG-3′ (FW) (with BamHI restriction site) and 5′-CCTGTCAGCT GTCATTCCAGCTCCAGTTTTTTC-3′ (RV) (with PvuII restriction site). The amplified *rnC* region carrying the E117D point mutation was ligated into the pL4440 vector upon BamHI and PvuII digestion, which leaves only one T7 promoter. For the construction of *rnC*(E117K) expression plasmid, the same strategy was followed: the following primers (Microsynth) were used: 5′-CCTGTGGATCCATGAACCCCATCGTAAT-TAATCG-3′ (FW) (with BamHI restriction site) and 5′-TAATGCTTTT-GACGGTGTCGGCGA-3′ (RV) and the 5′-CCTGTCAGCTGTCATTCC AGCTCCAGTTTTTTC-3′ (RV) (with PvuII restriction site) and 5′-TCAAAGCATTAATTGGTGGCGTATT-3′. For the construction of plasmids containing the sequences of the three expressed contigs which are differentially expressed between OP50 and OP50(xu363) that were identified from the RNA seq analysis, the following primers (Microsynth) were used: (For hit 1) 5′-ctcgaattcACCCCATCGTAATTAATCGG-3′ (FW) and 5′-ctcgaattcTATTTTTAAAGTGATGATAAAAGGC-3′ (RV), (for hit 2) 5′-ctcgaattcTTTTAGCGTTTATATCTGAAGG-3′ (FW) and 5′-ctcgaattcCTTATGATGATGATGTGCTTAAA-3′ (RV), (for hit 3) 5′-ctgcgaattcTCAGCGCAATTGATAGGC-3′ (FW) and 5′- ctggaattcGTTTTTT CGCCCCATTTAG-3′ (FW), all containing EcoRI restriction site at the 5′. The amplified bacterial regions were ligated into the pL4440 vector

upon digestion with EcoRI. These plasmids were used to generate dsRNA, which was used to test their efficiency to modulate aggregate accumulation.

### Lifespan assays

Lifespan assays were performed at 20 °C. Synchronized animal populations were generated by bleaching (hypochlorite treatment) gravid adult animals of the desired strain. Eggs were then placed on NGM plates with the different bacterial diets until the L4 larval stage, when they were again placed on the same diets. Their progeny was grown until the L4 larval stage and then transferred to fresh plates in groups of 20–25 worms per plate for a total of 100–120 individuals per condition (day 0 of adulthood). Animals were transferred to freshly-made RNAi plates every 2 days until the 12th day of adulthood and every 3 days until the end of the experiment. Animals were transferred to fresh plates every 2–3 days thereafter and examined every day for touch-provoked movement and pharyngeal pumping, until death. Worms that died owing to internally hatched eggs, an extruded gonad or desiccation due to crawling on the edge of the plates were censored and incorporated as such into the data set. Each survival assay was repeated at least twice, and figures represent typical assays. Survival curves were created using the product-limit method of Kaplan and Meier.

### Brood size determination

Synchronized animal populations were generated by bleaching (hypochlorite treatment) of gravid adult animals. Eggs were then placed on NGM plates with the different bacterial diets and were grown on the same diet for at least two generations. Ten L4 worms were picked and placed into separate NGM plates containing the corresponding diet. After the first 36 h, worms were moved daily to fresh plates until no more eggs were laid. The number of progeny was scored in each plate, and statistical analyses were performed using Sidak's multiple comparisons test following one-way ANOVA. Total or daily brood sizes are reported. Each brood size determination assay was repeated four times.

### Developmental rates

Synchronized animal populations were generated by bleaching (hypochlorite treatment) of gravid adult animals. Eggs were then placed on NGM plates with the different bacterial diets and were grown on the same diet for at least two generations. Each time, L4 worms were used to obtain synchronized worm populations. Approximately 12 2-day-old worms were used for egg laying for 2–3 h on fresh plates. The adult worms were removed, and the number of eggs was determined. When ~50% of the worms started reaching the L4/adult stage, we counted the total number of progeny that reached the L4/adult stage (for different strains, different time point was used due to developmental differences between strains). We performed statistical analyses using Sidak's multiple comparisons tests following one-way ANOVA. The assay was repeated at least three times.

### Analysis of polyQ protein aggregation

For the analysis of polyglutamine aggregation in body-wall muscle cells, we used the AM141 (polyQ40) and AM138 (polyQ24) strains. Synchronized animal populations were generated by bleaching (hypochlorite treatment) of gravid adult animals. Eggs were placed on NGM plates with the different bacterial diets and were grown on the same diet for at least two generations. L4 worms were used to obtain synchronized worm populations. Same-age adult worms were collected, immobilized with levamisole, before mounting on coverslips for microscopic examination with a Zeiss Axioplan 2 epifluorescence microscope. Protein aggregates, identified as visible bright puncta within muscle cells, in whole animals, were quantified with the help of ImageJ software.

### Motility assay

Synchronized nematodes were grown normally on NGM media plates containing the different diets for at least 2 generations. When worms reached 16–20 days of age, they were gently touched with an eyebrow. Worms not responding to touch were considered dead and were excluded from the analysis. Worms responding to touch but did not move were scored as paralyzed and the percentage of paralyzed worms per condition was evaluated. Each analysis was performed three times.

### Paralysis assay

To assay β-amyloid toxicity, we used the CL4176 temperature-sensitive strain that expresses β-amyloid in the body-wall muscle of *C. elegans*, leading to paralysis. Synchronized animal populations were generated by bleaching (hypochlorite treatment) gravid adult animals. Eggs were placed on NGM plates with the different bacterial diets at 15 °C. At the L4 stage, 12 animals were transferred to fresh plates for two days. Then, egg laying was performed for ~3–4 h. The mothers were removed and the plates containing only the eggs were placed back at 15 °C for 48 h, at which point the plates were shifted at 23 °C. Approximately 24 h later, and every 12 h, paralysis was scored. The percentage of paralyzed animals per condition is plotted against the time since temperature shifting. Each analysis was repeated at least three times.

### Pharyngeal pumping rate assay

To assess pharyngeal pumping rate the movement of the grinder (in the terminal bulb) was monitored every 20 s. Synchronized nematodes were grown normally on NGM media plates containing the different diets for at least 2 generations. When worms reached 8 days old pumping rate was monitored. Each analysis was performed at least three times.

### Myosin organization assay

To monitor myosin organization and muscle integrity 8-day-old MAH19 worms were used. The animals were grown for at least two generations on OP50, OP50(xu363) or HT115 diet. Muscles were classified into three categories depending on their muscle morphology (normal, intermediate and severe phenotype). Analysis was repeated three times.

### Supernatant isolation and supplementation

Overnight OP50 and HT115 bacterial cultures (6 ml) were centrifuged for 10 min at 16,000 × *g* at 4 °C. Bacterial supernatants were centrifuged for another 5 min and sterile-filtered using 0.2 μm filters. One ml of each supernatant (or LB medium as a control) was used to overlay NGM media plates. The pellets were once washed with LB medium (5 ml), resuspended in 0.5 ml cold LB medium and spotted onto NGM media plates (100 μl). Plates were exposed to UV light for 20 min to kill the bacteria. Worms were reared on these plates for two generations, and polyQ40::YFP protein aggregates were monitored.

### Methylcobalamin supplementation

The bacterial growth media was supplemented with exogenous methylcobalamin (Sigma), a vitamin B12 analog, to a final concentration of 25 μg/ml. OP50 bacteria were grown for two hours at 37 °C, and spotted onto NGM media plates. Worms were reared on these plates for two generations, and polyQ40::YFP protein aggregates were monitored at 2-day-old adults. To make sure that the methylcobalamin supplementation is effective, we monitored survival and developmental rate in the presence of hydrogen peroxide (Thermo Scientific) (0, 10 and 20 mM). The hydrogen peroxide-induced death was significantly rescued in the presence of methylcobalamin (Supplementary Fig. 1c). Their developmental rate was also significantly increased in the presence of methylcobalamin (Supplementary Fig. 1d, e).

## Autophagy measurements

Autophagy was measured in muscle cells of 8-day-old worms and in hypodermal seam cells of L4 worms according to guidelines[46]. Autophagosome number in hypodermal seam cells was assessed by using the GFP::LGG-1 reporter strain MAH14 grown on OP50 and HT115 bacterial diets for at least two generations. Approximately 20–30 L4-staged animals were collected, anaesthetized with 0.1% sodium azide and mounted on agarose pads for microscopic observation. The number of the GFP::LGG-1-positive autophagic puncta was quantified. Autophagosome and autolysosome numbers in hypodermal seam cells were assessed by using the mCherry::GFP::LGG-1 reporter strain MAH215 grown on OP50, OP50(xu363) and HT115 bacterial diets for at least two generations. Approximately 20 8-day-old animals were collected, anaesthetized with 0.1% sodium azide and mounted on agarose pads for microscopic observation using a confocal microscope. The number of GFP::LGG-1-positive puncta and the number of the mCherry::LGG-1-positive puncta were measured. The percentage of GFP-positive (autophagosomes) versus mCherry-positive (autophagosomes and autolysosomes) was quantified. Bigger puncta are present in tissues other than muscle and were disregarded for the analysis.

## *rnC* knockout in Nissle 1917

*E. coli* Nissle 1917 *rnc* knock-out strain was created by Red/ET recombination. The protocol was adapted from Datsenko and Wanner[47]. The wild-type Nissle 1917 strain was cultured in LB overnight at 37 °C, 200 rpm. The next day, the kanamycin-resistant pKD4 cassette was amplified by PCR using the following primers (Microsynth) 5'-CATCGTAATTAATCGGCTTCAACGGAAGCTGGGCTACACTTGTAGGCTGGAGCTGCTTCG-3' and 5'-CTGACCTGGCAGTGGATAGTAAATTCCTGATCGTGCGCTTATGGGAATTAGCCATGGTCC-3'. The primers contain overhangs corresponding to the neighboring sequences of the *rnc* gene. In parallel, the pKD46 plasmid was transformed into wild-type Nissle 1917 by electroporation. The next day, the PCR product was transformed into Nissle 1917 by electroporation. Adding 1 mM L-arabinose induced the ƛ recombinase expressed from the pKD46 plasmid, which led to the exchange of the *rnc* gene with the pKD4 cassette at the corresponding overhangs. Clones of Nissle 1917, where the *rnc* gene was knocked out and replaced by the kanamycin-resistant pKD4 cassette, were picked after kanamycin selection. Knock-out was further confirmed by PCR using the following primers (Microsynth) 5'-CTGAAGCGAATCTGGTCGGT-3' and 5'-CACTTTGTTCACCGCGAGGA-3'.

## Bacterial RNA isolation

A colony of HT115 bacteria was inoculated in LB medium and grown overnight at 37 °C in a shaking incubator. Next day 0.5 ml of the overnight culture were inoculated in 10 ml medium containing Tryptone (0.25%), NaCl (0.3%), Cholesterol (5 µg/ml), CaCl$_2$ (1 mM), MgSO$_4$ (1 mM), KPO$_4$ (25 mM), Ampicillin (100 µg/ml), Nystatin (100 U/ml), and grown overnight at 23 °C in a shaking incubator. Bacterial pellets were obtained after 3 min centrifugation at 1500×*g*. RNA was isolated from the pellets using the Quick-RNA Fungal/Bacterial Kit from Zymo Research and stored at −20 °C till use.

## Total RNAseq

Ribosomal RNA depletion was performed on 300 ng *E. coli* total RNA using NEBNext rRNA Depletion Kit Bacteria (Cat#E7850L, NEB, Ipswich, MA, USA). Following elution in 8 µl water, 1 µl of eluate for monitoring the depletion of ribosomal RNA on TapeStation instrument (Agilent Technologies, Santa Clara, CA, USA) using the High Sensitivity RNA ScreenTape (Agilent, Cat# 5067-5579), 6 µl of eluate were then mixed with Fragment, Prime Finish Mix provided in the TruSeq Stranded Total RNA Library Prep Gold Kit (Cat# 20020599, Illumina, San Diego, CA, USA) used for completing library preparation, in conjunction with the TruSeq RNA UD Indexes (Cat# 20022371, Illumina). 15 cycles of PCR were performed. Libraries were quality-checked on the Fragment Analyzer (Agilent Technologies, Santa Clara, CA, USA) using the Standard Sensitivity NGS Fragment Analysis Kit (Cat# DNF-473, Agilent Technologies), revealing good quality of libraries (average concentration was $72 \pm 46$ nmol/l and average library size was $317 \pm 37$ base pairs). Samples were pooled to equal molarity. The pool was quantified by Fluorometry using the QuantiFluor ONE dsDNA System (Cat# E4871, Promega, Madison, WI, USA) and sequenced Single-Reads 76 bases (in addition: 8 bases for index 1 and 8 bases for index 2) on NextSeq 500 using the NextSeq 500 High Output Kit 75-cycles (Illumina, Cat# FC-404-1005). Flow lanes were loaded at 1.8 pM. 1% PhiX was included in the pool. Primary data analysis was performed with the Illumina RTA version 2.11.3. This Nextseq run compiled a large number of reads (on average per sample: $22.5 \pm 11.7$ million pass-filter reads).

## RNAseq for small RNAs

The kit QIAseq FastSelect −5S/16S/23S (Cat# 335921, Qiagen, Hilden, Germany) was used for inhibiting the amplification of ribosomal RNA during library preparation, which was then performed from 120 ng total RNA of *E. coli* total RNA using SMARTer smRNA-Seq Kit for Illumina (Cat# 635029, Takara Bio, Shiga, Japan). Libraries were quality-checked on the Fragment Analyzer (Agilent Technologies, Santa Clara, CA, USA) using the High Sensitivity NGS Fragment Analysis Kit (Cat# DNF-474, Agilent Technologies), revealing good quality of libraries (average concentration was $0.94 \pm 0.22$ nmol/l and average library size was $191 \pm 5$ base pairs). Samples were pooled to equal molarity. The pool was quantified by Fluorometry using the QuantiFluor ONE dsDNA System (Cat# E4871, Promega, Madison, WI, USA) and sequenced Single-Reads 76 bases (in addition: 8 bases for index 1 and 8 bases for index 2) on NextSeq 500 using the NextSeq 500 High Output Kit 75-cycles (Illumina, Cat# FC-404-1005). Flow lanes were loaded at 1.8 pM. 1% PhiX was included in the pool. Primary data analysis was performed with the Illumina RTA version 2.11.3. This Nextseq run compiled a large number of reads (on average per sample: $24.0 \pm 6.2$ million pass-filter reads).

## RNA-seq data analysis

For the total RNA-Seq, reads were mapped against the Ensembl *E. coli* K12 DH10B reference genome distributed by iGenomes using STAR v2.7.9[48]. Read counts were summarized using the featureCounts function of Subread package v2.0.3. The matrix of uniquely mapped read counts was filtered for features with at least 10 reads in at least 3 samples. Read normalization was computed using the blind variance stabilizing transform as implemented in DESeq2 v1.40.2[49], and used as input for clustering by principal component analysis as implemented by prcomp in R v4.3.0.

For the short read RNA-Seq: Illumina sequencing reads were preprocessed by removing the first three nucleotides and polyA trimming using CutAdapt v3.4[50] as per the manufacturer's instructions, followed by 3′ quality trimming using Trimmomatic v0.39[51]. Adapter-trimmed reads were pooled and used as input to the Trinity de novo assembly pipeline v2.11.0[52]. The short reads were then aligned against the collection of contigs using Bowtie v1.2.3 (-n 1 -l 10)[53]. Reads were summarized using featureCounts. 4296 contigs had at least 10 reads in at least 3 samples. Uniquely mapped reads for these features were considered for differential expression using the Wald test as implemented in DESeq2. Significantly differentially expressed genes were considered for differences across the genotype contrasts greater than 2-fold with an adjusted *p*-value of ≤0.01. BLASTn was used to identify differentially expressed contigs that map to the *C. elegans* genome.

### Injections

Total bacterial RNA (85–100 ng/µl), genomic DNA (70 ng/µl) in water, and water as vehicle control were microinjected directly in the syncytium region of polyQ40-expressing *C. elegans* germlines cultured on OP50 diets. Similarly, 100 ng/µl of each dsRNA generated by in vitro transcription was used in a mixture to inject polyQ40-expressing *C. elegans* germlines cultured on OP50 diets. In each case, ~20 young adult worms were injected. The progeny of injected worms was monitored under the microscope. For each condition, about 60 2-day-old progeny grown on OP50 were used. Injections were repeated at least 3 times.

### In vitro transcription of bacterial segments

To synthesize dsRNA from the plasmids containing the three bacterial segments that were identified to be differentially expressed between OP50 and OP50(xu363) by the smRNAseq analysis, we used the MEGAscript™ T7 Kit (ThermoFisher Scientific). In short, we prepared the template DNA by linearizing all three plasmids (digests with HaeII). We obtained shorter regions that contain the DNA segments of interest between two T7 polymerases. Following the manufacturer's protocol, we assembled the transcription reaction and generated dsRNAs that were subsequently recovered by phenol:chloroform extraction and isopropanol precipitation. The pellets were re-suspended in water and frozen till the day of injection.

### Proteome analysis

Synchronized N2 and AM141 worms were placed on 4 plates and left to lay ~500 eggs. Once the worms reached the adult stage were collected in M9 buffer. Floating eggs were removed, and the remaining adults were washed with M9 and placed back in NGM plates. The next day, the 2-day-old worms were washed 3 times with M9, and the worm pellet was flash frozen in liquid nitrogen and stored at −80 °C. Worms were resuspended in 5% SDS, 10 mM Tris(2-carboxyethyl)phosphine hydrochloride (TCEP), 0.1 M TEAB and lysed by sonication using a PIXUL multi-sample sonicator (Active Motif) with pulse set to 50, PRF to 1, process time to 20 min and burst rate to 20 Hz, followed by a 10 min incubation at 95 °C. Lysates were TCA precipitated according to a protocol originally from Luis Sanchez (https://www.its.caltech.edu/~bjorker/TCA_ppt_protocol.pdf) as follows. One volume of TCA was added to every 4 volumes of sample, mixed by vortexing, incubated for 10 min at 4 °C, followed by the collection of precipitate by centrifugation for 5 min at 23,000×*g*. Supernatant was discarded, pellets were washed twice with acetone precooled to −20 °C and the washed pellets were incubated open at RT for 1 min to allow residual acetone to evaporate. Pellets were resuspended in 2 M Guanidinium HCl, 0.1 M Ammonium bicarbonate, 5 mM Tris(2-carboxyethyl)phosphine hydrochloride solution (TCEP), phosphatase inhibitors (Sigma P5726&P0044), and proteins were digested as described previously (PMID:27345528). Shortly, proteins were reduced for 60 min at 37 °C and alkylated with 10 mM chloroacetamide for 30 min at 37 °C. After diluting samples with 0.1 M ammonium bicarbonate buffer to a final Guanidinium HCl concentration of 0.4 M, proteins were digested by incubation with sequencing-grade modified trypsin (1/100, w/w; Promega, Madison, Wisconsin) for 12 h at 37 °C. After acidification using 5% TFA, peptides were desalted using C18 reverse-phase spin columns (Macrospin, Harvard Apparatus) according to the manufacturer's instructions, dried under vacuum and stored at −20 °C until further use.

Dried peptides were resuspended in 0.1% aqueous formic acid and subjected to LC–MS/MS analysis using an Exploris 480 Mass Spectrometer fitted with a Vanquish Neo (both Thermo Fisher Scientific) and a custom-made column heater set to 60 °C. Peptides were resolved using a RP-HPLC column (75 µm × 30 cm) packed in-house with C18 resin (ReproSil-Pur C18–AQ, 1.9 µm resin; Dr. Maisch

GmbH) at a flow rate of 0.2 µl/min. The following gradient was used for peptide separation: from 4% B to 10% B over 5 min to 35% B over 45 min to 50% B over 10 min to 95% B over 1 min followed by 10 min at 95% B to 5% B over 1 min followed by 4 min at 5% B. Buffer A was 0.1% formic acid in water and buffer B was 80% acetonitrile, 0.1% formic acid in water.

The mass spectrometer was operated in DIA mode with a cycle time of 3 s. MS1 scans were acquired in the Orbitrap in centroid mode at a resolution of 120,000 FWHM (at 200 *m/z*), a scan range from 390 to 910 *m/z*, a normalized AGC target set to 300% and maximum ion injection time mode set to Auto. MS2 scans were acquired in the Orbitrap in centroid mode at a resolution of 15,000 FWHM (at 200 *m/z*), precursor mass range of 400–900, quadrupole isolation window of 12 *m/z* with 1 *m/z* window overlap, a defined first mass of 120 *m/z*, normalized AGC target set to 3000% and a maximum injection time of 22 ms. Peptides were fragmented by higher-energy collisional dissociation (HCD) with collision energy set to 28% and one microscan was acquired for each spectrum.

The acquired raw files were searched using the Spectronaut (Biognosys v17.4) directDIA workflow against a *C. elegans* database (consisting of 26585 protein sequences downloaded from Uniprot on 20220222) and 392 commonly observed contaminants. Detailed search settings are depicted in the supplementary file named SpectronautSettings. Quantitative fragmented ion data (F.Area) was exported from Spectronaut and analyzed using the MSstats R package v.4.7.3. (https://doi.org/10.1093/bioinformatics/btu305). Data was normalized using the default normalization option "equalizedMedians", imputed using "AFT model-based imputation" and *p*-values and *q*-values for pairwise comparisons were calculated as implemented in MSstats.

### Generation of heatmaps and PCA of proteomic data

The principal component analysis (Supplementary Fig. 12a, b) was done based on log2 protein intensities calculated by MSstats data-Process function using prcomp from the stats package (v. 4.4.0). Missing protein intensities were replaced by zero. Points are colored by experimental group.

The heatmaps (Supplementary Fig. 12c, d) were generated based on log2 protein intensities calculated by MSstats dataProcess function using heatmap.2 function from the gplots package (v. 3.1.3.1). Data was centered by subtracting the respective column means from each value. Row and column-wise clustering was performed by calculating the distance matrix based on Pearson correlation across different proteins (rows) and across samples (columns), using the cor and as.dist function of the stats package (v. 4.4.0). Hierarchical cluster analysis was done based on the ward.D2 algorithm using hclust from the stats package. Heatmap colors are based on the RdYlBu palette of the RColorBrewer package (v. 1.1-3), ranging from 75% of minimal to 75% of maximal log2 protein intensities to reduce the impact of outlier values on the color gradient.

### GO term analysis and functional association networks

GO term analysis and protein-protein interaction networks enrichment analysis were performed with the use of ShinyGO (ver. 0.77) (http://bioinformatics.sdstate.edu/go/) with a 0.01 FDR cutoff and STRING (ver. 12.0) (https://string-db.org/cgi/input?sessionId=b0IlBAPfUcQa&input_page_show_search=on). A *q*-value of <0.05 was used to filter significant changes prior to the pathway analyses. Proteins between −0.3 and +0.3 fold change (log2ratio) were excluded from the analysis.

### Western blotting

Worm lysates were prepared as follows. 100 2-day-old synchronized N2 worms on OP50 and AM138 and AM141 on OP50, OP50(xu363) or HT115 for at least 3 generations, were transferred into 50 µl M9 buffer

and snap frozen with liquid nitrogen. Worms were thawed, mixed with 5x Laemmli buffer and boiled for 10 min at 95 °C. Lysates were subjected to standard SDS–polyacrylamide gel electrophoresis and electroblotted onto nitrocellulose. Anti-GFP (TP401, Torrey Pines, 1:5000) for the detection of polyQ40::YFP and anti-α-tubulin (T5168; Sigma-Aldrich; 1:2000) primary antibodies were used. Quantification of signals was performed using ImageJ.

### mRNA quantification and RT-PCR analysis

Total RNA from 100 2-day-old synchronized N2 worms on OP50 and AM141 on OP50, OP50(xu363) or HT115 for at least 3 generations was extracted by using the TRIzol reagent (Invitrogen). DNase treatment was performed using the RQ1 RNase-Free DNase kit (Promega). cDNA synthesis was performed by using the Goscript reverse transcription mix (Promega). Quantitative real-time PCR was performed using the GoTaq qPCR master mix kit (Promega) in a qTOWER$^3$G system (Analytikjena). The following primer (Microsynth) pairs were used: for measuring Q40::YFP mRNA levels: FW-5′-GTACAACTACAACAGCCA-CAACG-3′ and RV-5′-GTGTTCTGCTGGTAGTGGTCG-3′ and for measuring *pmp-3* mRNA levels as control: FW-5′-ATGATAAATCAGC GTCCCGAC-3′ and RV-5′-TTGCAACGAGAGCAACTGAAC-3′.

### Statistics and reproducibility

Statistical analyses and graphs were prepared using the Prism software package (version 9; GraphPad Software; https://www.graphpad.com). Data are reported as the mean values ± standard deviation (SD). For statistical analyses, *p* values were calculated by unpaired Student's *t*-test, one-way and two-way ANOVA with multiple comparisons test. All statistical tests were two-sided. The significance was determined by the *p*-values: $*p < 0.05$, $**p < 0.01$, $***p < 0.001$, $****p < 0.0001$ and n.s. = not significant $p > 0.05$. No statistical method was used to pre-determine sample size. No data were excluded from the analyses. The experiments were randomized, since worms were randomly placed on the various treatments/diets to minimize selection bias. The investigators were not blinded to allocation during experiments and outcome assessment.

### Reporting summary

Further information on research design is available in the Nature Portfolio Reporting Summary linked to this article.

## Data availability

The authors declare that the main data supporting the findings of this study are available within the article and its supplementary information files. RNAseq data have been uploaded to NCBI GEO with accession code GSE261167. Proteomic data have been deposited to the ProteomeXchange Consortium (https://www.proteomexchange.org/) via the MassIVE partner repository (https://massive.ucsd.edu/) with MassIVE data set identifier MSV000094244 and ProteomeXchange identifier PXD050390. Source data are provided with this paper.

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

## Acknowledgements

We thank Pascal Ankli, Jessica Arnegger, Tara Thorsen, Vanessa Brullo, Sheuli Begum, Jordi Greoles Cano, Sorismonde Manchester Vareille, Isabella Santi, Louise Larsson, Médéric Diard and Dora Stetak for technical support. We also thank the Genomics Facility Basel of the University of Basel and the Department of Biosystems Science and Engineering, ETH Zurich, for carrying out the Next-Generation Sequencing and the Imaging Core facility (IMCF, Biozentrum, University of Basel) and in particular Dr. Alexia Loynton-Ferrand for the technical assistance provided on the LSM880 point scanning confocal microscope. Genomics computations were performed at sciCORE (http://scicore.unibas.ch/) scientific computing center at University of Basel. We are grateful to Susan E. Mango, Ian G. Macara and Nikolaos Charmpilas for critical reading of the manuscript. Some nematode strains used in this work were provided by the Caenorhabditis Genetics Center, which is funded by the National Center for Research Resources of the National Institutes of Health and S. Mitani (National Bioresource Project) in Japan. We thank Read Pukkila-Worley and Andy Fire for providing the AU306 strain and plasmid vectors, respectively. This work is supported by Swiss National Science Foundation grant 185127 (ASpang), Swiss National Science Foundation grant 197779 (ASpang), Swiss National Science Foundation grant 219513 (ASpang), The Novartis Foundation for Medical-Biological Research grant #20C179 (ASpang), University of Basel grant FoFo 3BZ5106 (ASpang) and the University of Basel.

## Author contributions

Conceptualization: E.K. and A. Spang. Methodology: E.K., C.M., and G.F. Investigation: E.K., C.M., G.F., D.R., and A. Spang. Visualization: E.K., C.M., and G.F. Funding acquisition: E.K. and A. Spang. Supervision: E.K., A. Schmidt (proteomic data), and A. Spang. Writing original draft: E.K., A. Spang. Writing review and editing: E.K. and A. Spang.

## Competing interests

The authors declare no competing interests.
