## [Transparent Peer Review file · Nature Communications]

Bacterial RNA promotes proteostasis through inter-tissue communication in *C. elegans*

Corresponding Author: Professor Anne Spang

Version 0:

Reviewer comments:

Reviewer #1

(Remarks to the Author)

Summary

In this manuscript, Kyriakakis et al. propose that RNA species in bacteria can reduce protein aggregation in *C. elegans*. Furthermore, the authors demonstrate that this anti-aggregation mechanism requires RNAi machinery, autophagy, and inter-tissue communication.

The experiments were adequately carried out, and the data were nicely presented with consistent color coding. RNA-induced anti-aggregation is interesting and robust, with clear differences among different bacteria. Moreover, they provided the detailed molecular mechanisms underlying the anti-aggregation effect of bacteria using RNAi and genetic mutants. However, I have some concerns, as detailed below.

Major comments

1. My biggest concern is the physiological relevance of this work because the anti-aggregation phenotype is seen only in the combination of the non-natural bacteria (HT115 or mutated OP50) and aggregation of artificially introduced polyQ40::YFP. This experimental design raises the possibility that the detailed molecular mechanism proposed in this work is only applicable to an artificial setting. The authors examined a few physiologically relevant phenotypes. However, (1) HT115 with anti-aggregation did not have any effect on the lifespan (Fig. 1B); (2) HT115 only had a marginal effect on paralysis at a very later stage (Day 18-20), where 70-80% of animals were already dead (Fig. 1E); (3) HT115-fed animals had less brood size than OP50-fed ones (Fig. 1D and 1I). These results raise the question of the beneficial effect of RNA in a natural setting.
2. Authors clearly demonstrated that ribonuclease 3 in bacteria plays a critical effect on the anti-aggregation phenotype (Fig. 3). However, they could not narrow down what kind of RNA species in HT115 play a role despite the attempt to find candidates using RNAseq (Fig. S12). As a result, their major statement, "RNA is important," relies on a single experiment of RNA injection in Fig. 3G. Thus, the statement should be strengthened by additional experiments. I understand that narrowing down the specific RNA species is challenging and that no specific RNA species might be required. However, they should address whether specific RNA species from HT115 are necessary or if an increased amount of dsRNA in HT115 is essential. They can examine whether RNAs were increased in HT115 and OP50(xu363), compared to OP50, AND whether or not RNAs from OP50 have beneficial effects at the same concentration as HT115-derived RNAs when injected. Through these experiments, they can address whether the species or amount of RNAs matter in the anti-aggregation phenotype. Furthermore, they should provide a quantification of those data and Fig. 3G data because these are critical data to support their notion.
3. Line 32: The authors claimed that bacterial RNAs cause a low level of stress but lack the evidence. Provide the data or reference to support this notion.
4. Line 121: The authors use empty vectors for the control of RNAi. However, I do not think this should be interpreted with caution because all RNAi bacteria should have more dsRNAs than EV-bacteria in the first place, which makes the data interpretation complicated. I mean, dsRNAs might affect the aggregates independently from the specific effect of RNAi. Therefore, they should test several random RNAi bacteria to show that these RNAi do not affect the aggregation. I understand they provided the RNAi that had no effect in Figure S15H. However, this point should be addressed when introducing RNAi (Line 121) to avoid confusing readers.
5. Line 130: If the autophagy is for removing aggregates, they should test the induction of the autophagy in muscles before

testing the systemic induction because the aggregation is induced in muscles.

6. There are several misaligned panels, as described in Minor Comments. The authors might want to present the results more carefully to increase the credibility of this work.

Minor comments

1. Line 95: Add a reference to show the link between motility and muscle function.
2. Line 160-163: Add references for vitamin 12 auxotrophy and ribonuclease.
3. Figure 3E: Interestingly, Nissle 1917-fed animals had more aggregates than OP50-fed animals, and the number of aggregates did not reduce to 0 in *mrc*- mutant, unlike OP50(*xu363*). Does it mean that Nissle 1917 has another pro-aggregation factor resistant to bacterial ribonuclease 3-induced anti-aggregation? Maybe the authors can briefly discuss that.
4. Line 215: Show the references for germline-specific RNAi and intestine-specific RNAi strains.
5. Line 466: Specify the definition of "aggregates" if it was scored by human eyes.
6. Line 469: Add the number of animals examined.
7. Line 619-: Species names in References should be italicized.
8. Figure 1 legend: (A)... where performed -> were performed?
9. Figure 2A: *bec-1*(RNAi) panel is misaligned
10. Figure 2E: A few letters in the background (T1?)
11. Figure 2F: *wdfy-3*(RNAi) panel is misaligned
12. Figure 2 legend: Describe the statistics for the quantification (Figs. 2B,D,E,G)
13. Figure 4B-H: The comparisons between wt and each mutat should be made in the same bacterial condition.
14. Figure S7 legend: *P. aeruginosa* should be italicized.
15. Figure S9 legend: *E. coli* should be italicized.
16. Figure S10 legend: The Figure S9 legend was mistakenly placed here.
17. Figure S12 title: Describe RNA-seq of what.
18. Figure S12 legend: *E. coli* should be italicized.
19. Figure S15A: Panels were misaligned.

Reviewer #2

(Remarks to the Author)

The manuscript "Bacterial RNA promotes proteostasis through inter-tissue communication in *C. elegans*" by Emmanouil Kyriakakis and colleagues explores the impact of diet on healthy ageing in *C. elegans*. The study showed that *C. elegans* fed with HT115 bacteria have fewer polyQ40 aggregates in body wall muscle cells and exhibit less age-related paralysis compared to animals fed with OP50 bacteria. Moreover, an increase in LGG-1::GFP foci in seam cells and the aggravation of polyQ40 aggregation after silencing of aggrephagy genes in HT115-fed animals indicate a protective enhancement of autophagy in response to the HT115 diet. The beneficial effect of the HT115 diet appears to depend on the absence of ribonuclease III, an enzyme that cleaves dsRNA, in HT115 bacteria, since deletion of the *mrc* gene coding for ribonuclease III also rendered OP50 beneficial. Moreover, injection of dsRNA extracted from HT115 bacteria was sufficient to reduce polyQ40 aggregation. The authors also showed that the beneficial effect depends on the RNAi machinery. While the topic is intriguing, and the significance of this work would be high if dietary bacterial dsRNA actually triggered a systemic protective stress response that attenuated protein aggregation, slowed muscle atrophy and prolonged healthspan. However, the current data do not sufficiently support these conclusions. Several crucial controls are missing, requiring further experiments to validate the claims made. Therefore, this manuscript cannot be accepted in its current form.

1. My biggest concern is that there is hardly any polyQ40 signal in animals grown on HT115. This not only makes it very difficult to see the worms in the respective figures (an accompanying brightfield image would be helpful). It also indicates that not only aggregation but also the expression of polyQ40 seems to be reduced in these worms. If only aggregation was reduced, the soluble protein should still be visible. This suggests that the observed effects of bacterial dsRNA could stem from the endogenous system RNAi machinery affecting transgene expression. It is known that especially high copy transgenes can be affected by the endogenous RNAi machinery. This assumption is supported by the fact that mutations in RNAi machinery genes lead to a significantly higher number of polyQ40 aggregates (figure 4B: *sid-1* mutant animals have about 80 polyQ40 aggregates; in figure 3A and C WT animals have about 40-60 aggregates). Thus, the beneficial effect of HT115 could stem from induction of the endogenous RNAi machinery that silences the expression of the polyQ40 transgene.

Therefore, the following control experiments are necessary to prove that the HT115 diet indeed improves muscle proteostasis and does only affect the polyQ40 transgene:

A. Transgene expression must be tested by RT-PCR and Western Blot to make sure that polyQ40 RNA and protein levels are identical in animals fed with HT115 and OP50.

B. The effect of KD of the aggrephagy genes, and in particular the RNAi machinery must be tested in WT animals by assessing their effect on age -dependent paralysis. In addition, temperature-sensitive (*ts*) mutants (e.g. with the myosin(*ts*) [*unc-54(e1301)*] mutation) should be used as protein folding sensors and the effect of HT115 on *ts* phenotypes (movement, disruption of myosin filaments) should be assessed. If HT115 increases proteostasis capacity in muscle cells, then the muscle defect of *ts* mutant animals should be ameliorated.

2. Bacterial density should be controlled for by measuring OD of the bacteria in key experiments to make sure that the effect of different bacteria is not a result of reduced food availability (which could increase autophagy).

3. The authors claim systemic induction of autophagy, but only show increased LGG-1::GFP foci in the seam cells. The

authors should at least also assess the number of LGG-1::GFP foci in muscle cells, since they observe an effect on polyQ40 aggregation in this tissue. Moreover, increased LGG-1::GFP foci could either result from an induction of autophagy or from a reduced autophagic degradation. The authors should therefore be more cautious with their descriptions or should measure autophagic flux and show that lysosomal degradation is not affected. The fact that their proteomics data does not show an upregulation of autophagy argues that it is not induced systemically.

4. Figure 2E: why is there a “T1” in the figure?

5. Figure 3G: Quantification of polyQ40 aggregation is missing. To substantiate their claim, the authors should also inject RNA isolated from OP50 and OP50(xu363) and examine the effects on polyQ40 aggregation.

6. The text describing Figure 4F is misleading. In line 212, the authors write that “a mutant in PPW-1 accumulated polyQ40 aggregates in muscle cells, independent of the diet”. However, the data in Fig. 4F show that the influence of food is statistically significant.

7. Figure 5: It is surprising that the authors chose to focus their analysis on genes that are differentially expressed in polyQ40 vs. WT animals. It is known that polyQ40 affects muscle structure, and therefore it is not surprising that genes involved in muscle structure are upregulated, nor that KD of these genes worsens polyQ40 aggregation. However, the beneficial effect of bacterial dsRNA should be independent of polyQ40. polyQ40 should only be a proteostasis sensor (as they state in the beginning), and therefore genes affected by polyQ40 expression should not be considered. The correct comparison would therefore be between WT animals grown on OP50 vs HT115 or better OP50 vs OP50(xu363). Genes that are only expressed in animals grown on OP50(xu363) and not in animals grown on OP50 plates should be tested for their positive effect on proteostasis (and not only polyQ40 aggregation).

8. Statistics

Some of the mean values appear to differ only slightly, but are nevertheless statistically highly significant. Unfortunately, there is no information about which multiple comparison test was used in the figure legends. There is also no information about which correction was used for the multiple comparison test. Please specify.

Also, information about the number of animals tested in each replicate or experiment is missing. Please add.

9. Supplementary figures: all figures concerning one main figure should be combined to increase readability.

10. Figure S2: A positive control for methylcobalamin is missing. Is it actually taken up and active? The description in the methods section suggests that the assay was done differently than in the cited paper (e.g., different methylcobalamin concentration, dead vs. alive OP50). The authors need to provide evidence that methylcobalamin is active (e.g., by repeating some of the experiments from the cited paper assessing mitochondrial health).

11. In the results section (line 123), the text describes polyQ40 aggregation, but the figure S5D is labelled Q35: which strain was used in this experiment?

12. Figure S6B: The statistics should also include a comparison between dead and alive bacteria, as it was done in figure S6D.

13. Figure S9: there are two figures labelled S9.

14. Figure S13: this type of analysis should be part of the main figure and is more informative than the volcano plots in the main figure.

Reviewer #3

(Remarks to the Author)

In general, the manuscript entailed “Bacterial RNA promotes proteostasis through inter-tissue communication in *C. elegans*” described an interesting story about the role of bacterial-derived RNAs in *C. elegans*. DIA-based proteomics analyses were applied to investigate the effects of different diets on the proteomes of different types of worms. The experiments were well-designed and results were interesting. However, some issues in the proteomic parts should be clarified and supplied.

1. Since I can't get access to the raw MS data and identification information of the DIA-based proteomics, no information about the biological replicates of MS experiments and quantification results of the DIA proteomic analysis was provided in the manuscript.

2. For quantitative proteomic data analysis, detailed information should be provided, for example, what were the search parameters for Spectronaut? Whether cross-run normalization was applied for the data analysis? Whether imputation for missing values was performed and how?

3. The cutoffs for screening of differentially-expressed proteins in the DIA-based proteomic analysis wasn't provided in the manuscript.

4. Heatmap and principle components analysis are important to demonstrate the reproducibility and clusters of proteomic data. The author should provide this information in the manuscript.

5. In the manuscript, only 84 proteins that were significantly up-regulated were analyzed with GO enrichment analysis and

STRING network analysis. How about the 110 down-regulated proteins? What are the roles of these down-regulated proteins in the different diets?

Reviewer #4

(Remarks to the Author)

The authors of this manuscript "Bacterial RNA promotes proteostasis through inter-tissue communication in *C. elegans*" investigated how OP50 and HT115, two *E. coli* strains commonly used as food for *C. elegans* worms raised in the laboratory. A lot of efforts were put into this study, as can be seen from the number of displayed items (5 figures in the main text and 16 supp figures) and the quality of the figures. However, after careful reading of this manuscript, I think that the main text, abstract included, needs a major revision. My concerns are as follows.

Major concerns:

1. This study tries to establish a mechanism that starts with dsRNA in the bacterial diet (component #1). Ingested dietary dsRNA is processed by the RNAi machinery in the worm body (component #2), leading to activation of autophagy/aggrephagy (component #3) and consequently, less polyQ40::YFP aggregation in body wall muscles (BWMs) (component #4). This is thought to be a systemic effect that somehow requires intact RNAi function in the germline, in addition to that in the intestine. Component #1 is backed up by overwhelming evidence that the authors have gathered from elaborate experiments. Component #2 is also on solid footing, although the role of RNAi in the germline and other non-intestinal tissues is unclear. Component #3 is weak. The only data that supports it is an increase in the number of GFP::LGG-1 puncta in the seam cells of worms raised on HT115 relative to that of worms raised on OP50 (Fig. 2C-D). Whether autophagy/aggrephagy is activated in BWMs of worms raised on HT115 was not examined. The other data (Fig. 2 A-B, E-G, and Fig. S5), although sizable, only show that autophagy/aggrephagy is required for worms feeding on HT115 to keep polyQ40::YFP aggregation at a low level. They lend no direct support to activation of autophagy/aggrephagy in BWMs, neither do RNA-seq or proteomic data (Fig. S12 and Fig. S13). It may be right that autophagy is elevated in worms on HT115 diet and it is possible that enhanced autophagy is realized through post-translational regulation, but to this end there is not sufficient experimental evidence for it. Component #4 is all right by itself but becomes problematic when it is treated as if it were the same as health span, as discussed below. The wording of the abstract smoothed over the gaps to the effect that it's easy for readers to connect the dots in their mind when there is not yet a line connecting them.
2. Health span can be measured using different assays. The measurement of health span in this study relies almost exclusively on aggregation of polyQ40::YFP in BWMs. I think it is more appropriate to deemphasize health span and adjust the wording towards protection against proteotoxicity, proteostasis, or the like.
3. The experimental evidence in Fig. 5 does not, in my opinion, support the conclusion that "the dietary RNA mediates this protective effect by increasing the levels of key proteins required for muscle function," unless RNAi reduced the abundance of a targeted UNC protein to the same level as that found in worms reared on OP50. In addition, genes knocked down or mutated in Fig. 5 I-N do not correspond to proteins up-regulated in worms raised on HT115 or OP50(xu363).

Minor problems

1. L246, the section title "Body-wall muscle contraction and sarcomere integrity alleviate accumulation of aggregates" is difficult to understand.
2. L182-184, "We confirmed that similar to HT115, the loss of aggregate formation on OP50(xu363) diet was also due to autophagy, and more specifically to aggrephagy (Fig. S10)." Conclusion is less than precise, for it implies that autophagy is THE reason, rather than A reason. I think the data show that autophagy, more specifically aggrephagy, is required for the low-polyQ-aggregation effect of the HT115 or OP50(xu363) diet.
3. Grammar errors in the titles of Figure 5 and Figure S14.
4. Title of Fig. S9, change to "Ribonuclease 3 depletion protects *C. elegans* from protein aggregation."
5. Fig. S10 is mislabeled as Fig. S9 with a mismatched figure legend.
6. Remove "T1" from Fig. 2E.
7. Different laboratories reported different effects of HT115 vs OP50 on *C. elegans* lifespan and brood size. For example, in contrast to what's seen in this study, PMID 33159120 reported no difference in brood size but a markedly longer lifespan for worms on HT115 diet relative to OP50. Can the authors compare the different results and discuss what might be causing the discrepancy?

Version 1:

Reviewer comments:

Reviewer #1

(Remarks to the Author)

The authors have addressed all of my comments with additional experiments. The paper was significantly improved. I have a few minor comments:

-Line 45: I thought that the influence of dietary restriction on the lifespan was first reported in rodents in 1917 (Green et al., Nat Rev, 2022).
<https://www.nature.com/articles/s41580-021-00411-4>

-Figure 4B-F. What I meant by my original comment is as follows. I believe the authors want to compare the effects of diet on

WT and mutants. For example, the dietary effect in the wt (difference between OP50 and OP50(xu363) in Fig 3a) is much larger than the dietary effect in the rde-2 mutants (difference between OP50 and OP50(xu363) in Fig 4e). I had no problem with their conclusion from the beginning because the data was clear. However, OP50 and OP50(xu363) are statistically different in rde-2. Thus, I thought it might be better to compare the effects of diet on WT and mutants. Alternatively, OP50(xu363) condition can be compared between wt (Fig. 3a) and sid-1 (Fig. 3b) etc. If it is not appropriate to compare these, the authors can weaken the statement slightly.

Reviewer #2

(Remarks to the Author)

The manuscript "Bacterial RNA promotes proteostasis through inter-tissue communication in *C. elegans*" by Kyriakakis and colleagues explores the impact of the bacterial diet on proteostasis in *C. elegans*. The study showed that *C. elegans* fed with HT115 bacteria had fewer polyQ40 aggregates in body wall muscle cells and exhibited less age-related paralysis compared to animals fed with OP50 bacteria. Moreover, an increase in LGG-1::GFP foci formation and the aggravation of polyQ40 aggregation after silencing of aggrephagy genes in HT115-fed animals suggested a protective enhancement of autophagy in response to the HT115 diet. The beneficial effect of the HT115 diet appears to depend on the absence of ribonuclease III, an enzyme that cleaves dsRNA, in HT115 bacteria, since deletion of the rnc gene coding for ribonuclease III also rendered OP50 beneficial. The authors also showed that the beneficial effect highly depends on the RNAi machinery.

Although the authors have addressed some concerns in their revised submission by incorporating additional experimental data, several critical issues remain unresolved and require further clarification. Please find my additional comments below.

Regarding point 1A:

- Authors' answer: "We observe that polyQ40 aggregates accumulate over time, even in worms fed HT115, supporting that the polyQ40 transgene is expressed in these animals."

This observation is precisely what one would expect if polyQ40 protein expression is reduced. Given that the polyQ40 transgene is a high-copy transgene, RNAi-mediated KD would likely not fully suppress its expression.

- Authors' answer: "We also provide images of worms expressing polyQ24::YFP, driven by the same unc-54 promoter (Fig. S2a). The polyQ24 animals show similar expression across all dietary conditions, including HT115. This result suggests that dietary dsRNA is unlikely to significantly alter polyQ40 expression, as polyQ24 (with fewer Q repeats) maintains comparable expression levels across conditions."

Similarly, polyQ24::YFP is a high-copy transgene, and RNAi-mediated knockdown would likely result in only a slight reduction in its expression. A 20-30% reduction in expression would not be readily detectable by microscopy. However, even such a modest reduction could significantly delay aggregation, as this process is highly sensitive to protein levels.

- Authors' answer: "Additionally, our proteomic data for expression of the unc-54 promoter in the presence of HT115 and OP50(xu363) diets versus OP50 do not suggest a lower expression. If bacterial dsRNA affected transgene expression, it would also likely affect unc-54 levels. However, we do not observe a decrease in unc-54 expression in HT115 or OP50(xu363)-fed worms; if anything, we see a slight increase rather than a reduction in the AM141 polyQ40-expressing strain. Here are the quantified values: For AM141 strain $\log_2\text{FC HT115 vs OP50} = 0.39$, $\log_2\text{FC HT115 vs OP50(xu363)} = -0.07$, $\log_2\text{FC OP50 vs OP50(xu363)} = -0.46$ and for wt N2 strain: $\log_2\text{FC HT115 vs OP50} = 0.21$, $\log_2\text{FC HT115 vs OP50(xu363)} = 0.06$, $\log_2\text{FC OP50 vs OP50(xu363)} = -0.14$, indicating that the expression is relatively stable across diets in the wt animals."

Why would KD of polyQ40 also impact endogenous unc-54 levels? The KD is sequence-specific and would target only the cDNA coding for polyQ40::YFP, without influencing expression driven by the unc-54 promoter.

- Authors' answer: "Finally, to show this more directly. We performed Western blot analyses comparing polyQ40 protein levels in worms fed with OP50, OP50(xu363), and HT115 (Fig. S6 and below). As negative control we used wt N2 worms grown on OP50. Alpha-tubulin was used as loading control... These results indicate no significant differences in polyQ40::YFP expression, confirming that the observed dietary effects primarily impact aggregation rather than transgene expression levels. These combined findings suggest that the bacterial diet specifically influences the aggregation phenotype of polyQ40 rather than silencing its expression, supporting our interpretation that the observed reduction in aggregates is due to altered proteostasis rather than transgene suppression."

The authors' procedure for analyzing polyQ40 levels presents significant limitations. Notably, sonication (which they used for lysis) can promote protein aggregation (PMID: 15459333), and the high-speed centrifugation of lysates described in the methods is likely to pellet polyQ40 aggregates, potentially underestimating total protein levels in the Western blot analysis. This concern is reinforced by the observation of a single, distinct Q40-YFP band in their Western blots, characteristic of soluble proteins. In contrast, previous studies using this construct frequently report smears or protein fractions trapped in gel pockets during electrophoresis, which is typical for aggregation-prone proteins.

The authors must provide evidence that their lysis procedure does not exclude polyQ40 aggregates and confirm that the total protein, not just the soluble fraction, is loaded onto the gel. Centrifugation at lower speeds - or avoiding it entirely - should be considered, and the pellet fraction should also be analyzed to ensure the absence of polyQ40 aggregates. Additionally, performing the suggested RT-PCR analysis of polyQ40 expression could have offered valuable validation without the complications associated with protein aggregation.

Regarding point 1B:

The authors report a diet-dependent mortality in approximately 50% of CB1301 unc-54(e1301) mutant animals when fed on

HT115 bacteria, which raises concerns regarding their experimental conditions. This strain has been utilized in numerous studies, including RNAi experiments involving HT115 bacteria (e.g., PMID: 32217667, PMID: 39128917), without similar issues. Such mortality could indicate confounding factors unrelated to proteostasis, potentially compromising the reliability of their observations.

The alternative experiments presented by the authors to address these concerns are not convincing. The pharyngeal pumping rate, which they measured, is influenced by a variety of factors beyond proteostasis, including bacterial diet (PMID: 8462849). Therefore, it is not a reliable approach to assess proteostasis in the context of a different bacterial diet.

Additionally, the observed improvement in muscle morphology is minimal and correlates only marginally with the slightly reduced paralysis observed in aged wild-type worms. This provides insufficient evidence to conclude that proteostasis is substantially enhanced in HT115-fed non-transgenic animals and therefore my concern remains that the reported effects might be specific to the high-copy polyQ40 transgene.

To evaluate proteostasis in non-transgenic animals, the authors should consider testing additional ts mutants (e.g., as described in PMID: 22242008, PMID: 16469881) to verify that their findings are not limited to transgene-specific effects. Especially, since the study now emphasizes a role of the HT115 diet in enhancing proteostasis rather than improving aging, it is crucial to rigorously confirm that proteostasis is indeed improved.

Regarding point 3:

I do not agree with the authors' interpretation of the results in the provided figure. If we consider there is basal autophagy in OP50-fed animals, then in HT115-fed animals, there appear to be fewer (or even no?) GFP-positive autophagosomes and a similar number of only RFP-positive autolysosomes. Based on these observations, I would interpret this as a reduction in autophagic activity rather than an induction. To my knowledge, autophagic flux is considered increased when there is a rise in both yellow (GFP-RFP colocalized) and red (RFP-only) puncta in cells, while a blockade in autophagic flux is indicated by an increase in yellow puncta without a corresponding rise in red puncta (PMID: 18188003). This interpretation also aligns with the sequencing data, which do not indicate an induction of autophagy.

Regarding point 5:

The authors show that injection of RNA isolated from either bacterial strain (OP50, OP50(xu363) or HT1115) does lead to a reduction on polyQ40 aggregation in the progeny of injected animals. These results are important and should not be relegated to a supplementary figure. Instead, it should be prominently featured in Figure 3G, replacing the current panel, especially since the existing figure lacks quantification of PolyQ40. Additionally, the accompanying explanation requires revision, as the current wording is unclear and misleading.

The data presented highlight an effect on PolyQ40 aggregation in the progeny, which is likely independent of intertissue transport mediated by SID-1. This is because the RNA is already present in the stem cells and is distributed to progenitor cells through cytosolic diffusion rather than intercellular transmission. The claim that they "directly injected the RNA into the tissue" is incorrect. The RNA is injected only into the gonads of the animals, and for an effect to be observed in other tissues, such as muscle cells, intercellular RNA transmission via SID-1 would be required. However, such transmission is likely inefficient for the reasons the authors have described.

Regarding point 7:

Regarding the authors' finding that diet-dependent differences were observed only in polyQ40 worms and not in WT worms: the gene expression changes could simply be a consequence of reduced polyQ40 aggregation rather than a causative factor. As such, this observation may not provide substantial mechanistic insight. Nevertheless, the authors should explicitly discuss this possibility in the manuscript to allow readers to form their own conclusions.

Regarding point 8. Statistics:

In some experiments, the authors tested as many as 900 worms, which raises concerns about potential oversampling. In contrast, for other assays, they tested 30-40 worms, which seems more appropriate and consistent with standard practices in the field. The authors should clarify the rationale for these discrepancies in sample size.

Regarding point 10:

There is no reference to the methylcobalamin experiments in the text.

Reviewer #3

(Remarks to the Author)

The authors addressed the reviewer's comments by analyzing the data carefully, supplementing some data and revising the manuscript carefully. The responses and the revised manuscript are reviewed as satisfactory. It seems a good shape for publication in Nature Communication.

Reviewer #4

(Remarks to the Author)

The authors have addressed my concerns. I have no more questions.

Version 2:

Reviewer comments:

Reviewer #2

(Remarks to the Author)

Review of "Bacterial RNA promotes proteostasis through inter-tissue communication in *C. elegans*" by Kyriakakis and colleagues.

I appreciate the authors' efforts to address some of my concerns. However, my main concern remains: There is still no convincing evidence that the bacterial diet directly improves proteostasis of wild-type worms. The observed effects on polyQ40 aggregation could be attributed to interactions with the endogenous RNAi pathway (affecting the transgene and thereby only indirectly proteostasis).

Point-by-point response to the authors' rebuttal letter:

Regarding point 1 (proteostasis of WT animals):

The authors acknowledge that the pharyngeal pumping assay is not a direct readout of proteostasis. Therefore, to substantiate the claim that the bacterial diet enhances proteostasis in wild-type animals, the use of mutant strains expressing metastable proteins as folding sensors would be more appropriate. The explanation provided for the high mortality observed with the CB1301 unc-54(e1301) mutant strain (that it might result from the absence of using heat shock) is not entirely convincing. Heat shock would typically exacerbate, rather than alleviate, stress-induced mortality. Other groups, including our own, have successfully used this strain under standard growth conditions. Even if this particular strain was challenging in their hands, the use of an alternative folding sensor strain would have strengthened their conclusions. As it currently stands, the observed effects seem primarily associated with transgenic animals expressing polyQ (or A β), which might be particularly sensitive to influences from the endogenous RNAi machinery.

Regarding point 2 (Interpretation of the autophagy reporter):

I respectfully disagree with the authors' interpretation. To distinguish between induction and blockage of autophagy, measurement of autophagic flux would be necessary, which has not been performed. Therefore, the conclusion that the bacterial diet decreases polyQ aggregation via enhanced autophagy is not sufficiently supported. The fact that KD of autophagy genes increases polyQ aggregation alone does not allow such a conclusion.

Regarding point 5 (dsRNA uptake and SID-1 function):

The authors have misunderstood my original comment. In the manuscript, they state: "We assume that the size of the dsRNA species matters for the uptake and processing from the intestinal cells. Previous work showed that ingestion of long dsRNA is more effective than shorter pieces, but the length of dsRNA had no effect when injected into the gonad (33, 34)." The papers they cite in Ref. 33 and 34 found that the dsRNA transporter SID-1 transports more efficiently long dsRNA. However, both feeding and gonadal injection require SID-1 for the transport of dsRNA into muscle cells expressing polyQ40. Thus, the difference the authors observe cannot be explained solely by differences in dsRNA lengths.

Regarding point 8 (sample size and statistical power):

Although there were many progenies on the plates, the authors could have randomly selected fewer animals for the analysis. A smaller sample size, would have allowed for more meaningful statistics.

Version 3:

Reviewer comments:

Reviewer #1

(Remarks to the Author)

I am convinced by the author's rebuttal to Reviewer 2's remaining concerns.

Reviewer #2

(Remarks to the Author)

My main concern from the beginning has been that animals grown on HT115 show much weaker polyQ40 signal, suggesting that the bacterial diet may influence transgene expression, not just protein aggregation. Although the authors have added Western blot and RT-PCR data to address this issue, I do not find these experiments entirely convincing. Both techniques are highly sensitive to input amounts and prone to signal saturation, and the controls presented do not rule out expression-level differences with sufficient confidence.

I would also like to clarify that I never stated that these experiments alone would be sufficient to prove that HT115 bacteria improve proteostasis. On the contrary, I believe it is problematic to base the central claims of an entire study on a transgenic reporter, when the observed beneficial effects seem to be mediated by the RNAi machinery in the germline, which is known to efficiently suppress high copy repetitive transgenes. This is particularly relevant for the polyQ40 construct, which is not a neutral proteostasis reporter. It strongly affects proteostasis itself and even slightly reduced levels would immediately improve proteostasis simply by reducing the stress imposed by the transgene.

The authors' statement that a diet-induced RNAi effect on transgene expression would be "intriguing and impactful" completely misses the point. If the bacterial diet reduces expression of the polyQ40 protein itself, then the decrease in

aggregation is simply due to lower expression of the reporter. That would not only weaken the study's central conclusion, it would suggest that the observed effects are artifacts of transgene silencing.

A recent study illustrates this point clearly, showing that RNAi activation can silence high-copy repetitive transgenes: PMID: 40494360.

Moreover, the fact that gene expression changes linked to improved proteostasis are only seen in polyQ40-expressing animals fed HT115, but not in wild-type animals on the same diet, further suggests that HT115 selectively affects the transgenic strain. Reduced expression of the polyQ40 transgene would lower the proteotoxic stress and secondarily improve proteostasis. This would represent an indirect effect, mediated by modifying transgene expression, rather than a true enhancement of the proteostasis network.

Because of this concern, I suggested focusing more on wild-type or mutant strains that carry changes in endogenous genes but do not express transgenes. I am, of course, aware that such mutants are not "wild-type". My intention was simply to highlight that these strains are non-transgenic, and thus suitable for testing whether the observed effects occur independently of transgene silencing. I hope this lengthy clarification makes the point sufficiently clear so that it can no longer be misunderstood or dismissed based on imprecise wording.

I acknowledge that the authors attempted to perform the suggested experiment using a temperature-sensitive mutant strain to address this concern. However, I am surprised by their statement that these worms cannot be grown at 15 °C. *C. elegans* can be maintained and fed bacteria at this temperature, and RNAi remains effective under these conditions. If the beneficial effects of HT115 are indeed mediated by the RNAi machinery, they should still be detectable after growing them at 15 °C for two weeks and then shifting them to 20 °C or 25 °C for the assay. The authors now state that maintaining the worms at 20 °C is essential to observe the effect, but this is neither mentioned nor justified in the manuscript.

In any case, if this particular approach is not feasible for technical reasons, then alternative transgene-independent experiments must be pursued. It is not sufficient to report that the experiment failed and simply move on without attempting a different strategy. Some form of independent validation remains necessary, as this issue is central to the study's main claim. This was also the case for other experiments. Most of the experiments I suggested ended up not supporting the authors' original claims. Yet the response I get is that they did what was asked and therefore consider the matter resolved. The authors also imply that I keep changing my requests, which is not the case. My concerns have remained consistent throughout all review rounds. Unfortunately, instead of addressing these concerns directly, the authors appear more focused on dismissing the criticism.

Regarding point 2 (Interpretation of the autophagy reporter):

The authors have not toned down their interpretation of the autophagy data, nor have they adequately addressed the original concern. The data presented in Figure 2e show fewer GFP-positive foci but more RFP-positive foci in animals fed HT115 compared to those fed OP50. This pattern does not match the "high autophagy" scenario depicted in their schematic, which would require an increase in both GFP and RFP signals. Instead, the observed accumulation of RFP-positive lysosomes could reflect impaired degradation rather than increased autophagic activity.

Therefore, the conclusion that autophagy is beneficial is not supported by the data. Despite this, the authors still state in the abstract, "This beneficial effect depends on low levels of systemic selective autophagy..." which clearly suggests enhanced autophagic efficiency. Without direct evidence for increased flux, this interpretation is misleading. Moreover, this interpretation is consistently stated across the manuscript, not only in the abstract.

The remaining issues are not important enough to spend more time on.

REVIEWER COMMENTS

We sincerely thank the reviewers for their thoughtful comments and valuable suggestions, which have greatly contributed to strengthening the study through the inclusion of additional experiments.

Reviewer #1 (Remarks to the Author):

Summary

In this manuscript, Kyriakakis et al. propose that RNA species in bacteria can reduce protein aggregation in *C. elegans*. Furthermore, the authors demonstrate that this anti-aggregation mechanism requires RNAi machinery, autophagy, and inter-tissue communication.

The experiments were adequately carried out, and the data were nicely presented with consistent color coding. RNA-induced anti-aggregation is interesting and robust, with clear differences among different bacteria. Moreover, they provided the detailed molecular mechanisms underlying the anti-aggregation effect of bacteria using RNAi and genetic mutants. However, I have some concerns, as detailed below.

Major comments

1. My biggest concern is the physiological relevance of this work because the anti-aggregation phenotype is seen only in the combination of the non-natural bacteria (HT115 or mutated OP50) and aggregation of artificially introduced polyQ40::YFP. This experimental design raises the possibility that the detailed molecular mechanism proposed in this work is only applicable to an artificial setting. The authors examined a few physiologically relevant phenotypes. However, (1) HT115 with anti-aggregation did not have any effect on the lifespan (Fig. 1B); (2) HT115 only had a marginal effect on paralysis at a very later stage (Day 18-20), where 70-80% of animals were already dead (Fig. 1E); (3) HT115-fed animals had less brood size than OP50-fed ones (Fig. 1D and 1I). These results raise the question of the beneficial effect of RNA in a natural setting.

We thank the reviewer for this thoughtful feedback. We acknowledge that the anti-aggregation phenotype is observed in an artificially introduced polyQ40::YFP animal model. However, we believe this work valuable input for several reasons. For instance:

1. **Bacterial dsRNA's Impact:** Our key finding is that bacteria-derived dsRNAs can influence organismal physiology. This is a novel and only very limited data is available. Importantly there appears to be different ways how dsRNA can alter organismal physiology. While there are reports that dsRNA can act as siRNA under particular circumstances, our study shows that dsRNA can

systemically affect physiology by inducing low levels of autophagy that we interpret as inducing low levels of stress.

2. **Use of Clinically Relevant Strains:** While HT115 is not used in industry or physiology, we also show similar beneficial effects with probiotic Nissle 1917 bacteria, which are used in the clinics and are available in drug stores. This suggests the findings could have broader applicability beyond artificial settings.
3. **Accepted Model of Protein Aggregation:** The polyQ40::YFP model is widely used and accepted in research as a proxy for studying proteostasis and aggregation, despite being artificially introduced.
4. **Healthspan vs. Lifespan:** Although we did not observe lifespan changes, the significant improvement in motility in older animals (17-18 days old) points to benefits in healthspan, which is often considered more relevant than lifespan extension. Additionally, we observed a positive impact on pumping rate and on muscle morphology in 8-day old worms (post reproductive phase), further indicating a beneficial effect on physiology. These data are now shown included into the manuscript (Fig. S8) and below for your consideration.

For wt N2:

For polyQ40-expressing worms AM141:

For the muscle morphology, we used the MAH19 (*rrf-1(pk1417) I; myo-3(st386) V; stEx30 [myo-3p::GFP::myo-3 + rol-6(su1006)]*) strain, in which MYO-3 is fused to GFP and expressed under its own promoter, in a *myo-3* mutant background. Muscle morphology was classified into three categories (normal, intermediate and severe phenotype) (see below and Fig. S8e).

After growing the worms on the three diets muscle morphology was assessed (below and Fig. S8f).

We observe a significantly larger fraction of the normal phenotype in worms raised on both OP50(xu363) and HT115 diets versus on OP50, suggesting a positive effect of those diets.

We agree that it is unclear at this point whether our results are extrapolatable to humans, but we have to start somewhere. We would like to point out that the benefit of caloric restriction on lifespan was also first discovered in *C. elegans* and is now deemed to at least extend healthspan in humans.

To address the final point regarding the reduced brood size in HT115-fed animals compared to OP50-fed ones: as shown in Figure S7, this difference in brood size is not RNA-dependent. Specifically, OP50(xu363) and OP50 diets resulted in a similar brood size in wild-type (WT) animals. In the polyQ40-expressing background, we observe a small difference in brood size between OP50 and OP50(xu363), likely due to increased stress in these worms.

These results suggest that bacterial RNA does not influence every aspect of worm physiology, but rather exhibits some specificity, particularly in proteostasis and probably in cellular stress responses. It is important to recognize that OP50 and HT115 bacteria are inherently different even though they are both *E. coli* strains. Different *E. coli* strains have been shown to elicit different effects based on the genetic background, stress, or specific readouts by a number of studies. The difference in brood size does not contradict the aggregation phenotype we observe, and we want to clarify that we are not suggesting that bacterial dsRNA is a universal solution for all physiological issues. Rather, our focus is on its specific effect on proteostasis, which we find to be significant. We would also like to remind the reviewer that loss of proteostasis is a recognized hallmark of ageing. Yet, of course it is not the only one.

2. Authors clearly demonstrated that ribonuclease 3 in bacteria plays a critical effect on the anti-aggregation phenotype (Fig. 3). However, they could not narrow down what kind of RNA species in HT115 play a role despite the attempt to find candidates using RNAseq (Fig. S12). As a result, their major statement, “RNA is important,” relies on a single experiment of RNA injection in Fig. 3G. Thus, the statement should be strengthened by additional experiments. I understand that narrowing down the specific RNA species is challenging and that no specific RNA species might be required. However, they should address whether specific RNA species from HT115 are necessary or if an increased amount of dsRNA in HT115 is essential. They can examine whether RNAs were increased in HT115 and OP50(xu363), compared to OP50, AND whether or not RNAs from OP50 have beneficial effects at the same concentration as HT115-derived RNAs when injected. Through these experiments, they can address whether the species or amount of RNAs matter in the anti-aggregation phenotype. Furthermore, they should provide a quantification of those data and Fig. 3G data because these are critical data to support their notion.

We respectfully disagree with the statement that our conclusion about the importance of RNA relies solely on a single RNA experiment. First, we show that bacteria lacking the *rnC* gene, specifically HT115, mutated OP50 (OP50(xu363)), and the probiotic Nissle bacteria, which we have engineered to lack *rnC*, have a positive impact on the anti-aggregation phenotype, while feeding the worms bacteria like OP50 or W3310 which carry the *rnC* gene do not exhibit this effect and aggregates accumulate. Our data clearly demonstrate that RNA processing by ribonuclease 3 plays a crucial role for the anti-aggregation effect. Furthermore, we provide evidence that a mutated form of ribonuclease 3, which binds but cannot cleave dsRNA, has no beneficial effect in preventing protein aggregate formation. These results support the significance of dsRNA originating from bacteria for the beneficial effect on proteostasis.

Furthermore, we show that multiple components of the RNAi machinery are required for the observed phenotype. For example, SID-1, which is responsible for the intake of dsRNA, is necessary for this anti-aggregation effect, reinforcing the importance of RNA uptake in driving the phenotype.

Moreover, we directly show that RNA isolated from bacteria and injected into the gonads of worms leads to progeny with a reduced aggregation phenotype. While we have tried to identify a specific RNA species, it turned out that dsRNA per se and not a specific RNA species is responsible for the beneficial effect. We speculate that turning on the RNAi machinery may signal to other stress pathways including the proteostasis pathway. We admit we do not know yet how the signal is transmitted. We think, however, that solving this issue would constitute an entirely new study that goes beyond the focus of this manuscript.

The total RNA levels are not significantly different between the bacterial strains, since there was no significant difference when we isolated total RNA for the injection or RNAseq purposes. While we cannot rule out changes in specific RNA concentrations, we did conduct the experiment suggested by the reviewer, injecting RNA isolated from OP50. Interestingly, we found that injecting similar amounts of total bacterial RNA produced the same effect, whether from OP50, OP50(xu363), or HT115.

Although this result may seem counter intuitive, but our data align well with existing knowledge about the mechanism of RNAi in *C. elegans* (PMID: 12970568 & PMID: 26067272). Unlike in mammals, ingestion of long dsRNA is more effective than shorter pieces. Our findings suggest that while long dsRNA may be effective through feeding, the size of the dsRNA does not seem to matter when injected. Moreover, we cannot exclude that some RNAs may basepair during isolation and hence represent a dsRNA species potent of eliciting the beneficial proteostasis effect.

The reviewer appears to indicate that it was a failure that we could not identify a particular bacterial dsRNA or a particular *C. elegans* target which would be downregulated. We do not think this to be the case; we do not think that our analysis failed. First, of all the data is the data and there was just no bacterial RNA that could act as siRNA in *C. elegans* in our samples. We tested some poor matches, and they did not seem to have an effect. So, it is likely that there is not one specific *C. elegans* protein (or specific class) down regulated by the bacterial RNA. This was also confirmed by our proteome analysis, with the caveat that we of course do not detect 100% of the proteome. Having said this, we did see an increase of proteins beneficial for muscle function among which there were also heatshock proteins in polyQ40-expressing worms reared on HT115 and OP50(xu363), consistent with the notion the in the affected tissue an anti-stress response was mounted.

The results of aggregate formation upon injection of RNA isolated from different bacterial strains are now included in the manuscript Figure S9.

3. Line 32: The authors claimed that bacterial RNAs cause a low level of stress but lack the evidence. Provide the data or reference to support this notion.

As we do not have final proof of this notion, we moved it from the abstract.

4. Line 121: The authors use empty vectors for the control of RNAi. However, I do not think this should be interpreted with caution because all RNAi bacteria should have more dsRNAs than EV-bacteria in the first place, which makes the data interpretation complicated. I mean, dsRNAs might affect the aggregates independently from the

specific effect of RNAi. Therefore, they should test several random RNAi bacteria to show that these RNAi do not affect the aggregation. I understand they provided the RNAi that had no effect in Figure S15H. However, this point should be addressed when introducing RNAi (Line 121) to avoid confusing readers.

We would like to clarify that the empty vector (EV) control we used contains an approximately 200 bp segment, so it is not entirely devoid of dsRNA. Additionally, the RNAi constructs we used are designed to target specific genes, reducing the likelihood of off-target effects. We would like to point out that off-target effects in *C. elegans* are not an issue because much longer dsRNA pieces are used resulting in many siRNAs, increasing drastically the specificity compared to mammalian cells. Moreover, dsRNA-dependent RNA polymerase generates more dsRNA, which amplifies the siRNA pool in *C. elegans*.

As the reviewer suggests, to ensure that random dsRNA can equally reduce the aggregate burden and that the increase in aggregate formation by RNAi against components of the autophagic machinery is specific, we provide now two additional RNAi constructs that are not related to RNAi, autophagy or neuromuscular junction. *osm-3* is the *C. elegans* ortholog of human KIF17, a component of a kinesin complex and is expressed in ciliated neurons. The second gene we knocked down was *alh-6*, which encodes a mitochondrial matrix localized aldehyde dehydrogenase. In both cases we do not observe aggregate formation, confirming that a) dsRNA is reducing aggregate burden and b) in the absence of autophagy the beneficial effect of dsRNA on the prevention of aggregate formation is abrogated.

These data are now included in Figure S3g and are shown below for your convenience.

This is now mentioned in the same paragraph where the RNAi experiments are introduced as suggested by the reviewer. To avoid confusing readers, the following sentence has been added: "This effect was not due to the presence of dsRNA itself,

as knockdown of unrelated genes did not increase aggregate formation (Fig. S3g and S14h).”

5. Line 130: If the autophagy is for removing aggregates, they should test the induction of the autophagy in muscles before testing the systemic induction because the aggregation is induced in muscles.

To address the reviewer’s request, we assessed autophagy in muscle cells. We initially chose seam cells as a model because it is easier to monitor autophagy accurately in this distinct tissue at the L4 stage. Autophagy in muscle cells, by contrast, presents challenges: autophagy levels are generally very low in young muscle cells, which complicates quantification (see PMID: 25569839, PMID: 28675140). Additionally, the signal from other tissues can mask or overestimate the signal in muscle tissues. However, we conducted further analysis in muscle tissue by performing confocal microscopy on muscle cells of day 8 adult *C. elegans*. We used the MAH215 (mCherry::GFP::lgg-1) strain that expresses LGG-1 fused with mCherry and GFP under its own promoter. mCherry and GFP signal is present on autophagosomes, whereas when autolysosomes are formed the GFP signal is quenched due to the more acidic environment and only the mCherry is present. In this way we can differentiate autophagosomes from autolysosomes. Our findings indicate that HT115 (and OP50(xu363)) diet led to a decrease in the relative GFP/mCherry ratio suggesting an increase in the amount of LGG-1 that arrived in autolysosomes in muscle cells, when compared to the OP50 diet (Fig. 2e, f, Fig. S7c and below). Thus, we observed an increase in autophagy in muscle cells. Moreover, these results confirmed that the observed dietary effects on autophagy are not limited to a single cell type, supporting our hypothesis of a diet-driven modulation of systemic autophagic activity.

Arrows show some of the autophagosomes (both GFP and mCherry).

6. There are several misaligned panels, as described in Minor Comments. The authors might want to present the results more carefully to increase the credibility of this work.

We thank the reviewer for pointing out the misaligned panels. We have carefully reviewed the figures and made all necessary corrections as suggested. The panels are now properly aligned to ensure the results are presented clearly and accurately, enhancing the credibility of the work.

Minor comments

1. Line 95: Add a reference to show the link between motility and muscle function.

To support the link between motility and muscle function, we provide the following references that highlight the direct connection between muscle integrity and movement capacity in *C. elegans*. Laranjeiro et al., 2017 (PMID: 28395669) connects muscle mitochondrial health with motility in *C. elegans*, Morley et al., 2002 (PMID: 12122205) shows motility deficits linked to muscle aggregation, Gieseler et al., 2017 (PMID: 27555356) discusses how body wall muscles are essential for locomotion.

2. Line 160-163: Add references for vitamin 12 auxotrophy and ribonuclease.

We modified the text and we now provide some references as suggested by the reviewer.

3. Figure 3E: Interestingly, Nissle 1917-fed animals had more aggregates than OP50-fed animals, and the number of aggregates did not reduce to 0 in *rnc*-mutant, unlike OP50(xu363). Does it mean that Nissle 1917 has another pro-aggregation factor resistant to bacterial ribonuclease 3-induced anti-aggregation? Maybe the authors can briefly discuss that.

Thank you for this insightful comment. We do not know the exact mechanism why Nissle 1917-fed animals had more aggregates than OP50-fed animals, and why the aggregates were not more drastically reduced in the *rnc* mutant. There is, however, the possibility that Nissle 1917 contains an additional pro-aggregation factor that is resistant to the anti-aggregation effect of bacterial ribonuclease 3. Additionally, there is an important difference between OP50 and Nissle 1917 in that Nissle 1917 forms readily biofilms, which are hard to eat by *C. elegans*. The biofilm formation is actually one of the main reasons why Nissle1917 is used in the clinics. We reduced biofilm formation and still providing enough food by seeding higher concentrations of Nissle 1917 bacteria, by letting them divide only a few times. We consider the biofilm formation the likely cause for the less dramatic effect on aggregate formation than with OP50(xu363) worms. However, we would like to emphasize that despite the issue with biofilm formation, removing the *rnc* gene from Nissle 1917 has a positive effect on aggregation. As suggested, we now briefly mention this possibility in the results section.

4. Line 215: Show the references for germline-specific RNAi and intestine-specific RNAi strains.

The references have been added as suggested.

5. Line 466: Specify the definition of “aggregates” if it was scored by human eyes.

We now define aggregates as, visible bright puncta within muscle cells. The section has been modified accordingly.

6. Line 469: Add the number of animals examined.

The number of animals used in each experimental setup has been added to the figure legends.

7. Line 619-: Species names in References should be italicized.

Zotero automatically removed the italicized format. We tried to do our best to add the species names in the correct format manually.

8. Figure 1 legend: (A)... where performed -> were performed?

Thank you for spotting the spelling mistake.

9. Figure 2A: bec-1(RNAi) panel is misaligned

The panel has been aligned.

10. Figure 2E: A few letters in the background (T1?)

We apologize for the mistake; HT115 has been removed.

11. Figure 2F: wdfy-3(RNAi) panel is misaligned

The panel has been aligned.

12. Figure 2 legend: Describe the statistics for the quantification (Figs. 2B,D,E,G)

Figure 2 and its legend has been modified accordingly.

13. Figure 4B-H: The comparisons between wt and each mutat should be made in the same bacterial condition.

We are not entirely sure what this reviewer wants us to do. In Figure 4a, we performed RNAi experiments in the wild-type polyQ40 expressing strain to show that if we knockdown the RNAi machinery, we increase polyQ40 aggregate formation. While this could be seen as being complicated by performing RNAi on the RNAi machinery, we wanted to corroborate our findings in panel A with mutants in the RNAi machinery (panels B-F). In these mutants the beneficial effect of the diet was completely abrogated. The increase of aggregates on HT115 and OP(xu363) diet of the germline-specific argonaute mutant *ppw-1(tm5919)*, made us wonder whether there is inter-tissue communication between the germline and the intestine or whether RNAi functioning either in the intestine or in the germline would be sufficient to have the beneficial effect to avoid aggregate formation. Limiting the RNAi pathway to either the germline or the intestine was insufficient to reduce polyQ40 aggregate formation. This let us to conclude that there is inter-tissue communication between the intestine and the germline. We do not see how we can do the same experiment in a wild-type strain to reach the same conclusion. If the reviewer is worried about the penetrance and reproducibility of our data, we repeated the experiment of the different diets fed to wild-type animals expressing polyQ40 many times, often in parallel to the other experiments such as on the mutants. We see invariably that on OP50 polyQ40 aggregates form, as has been observed by many groups before, and

on OP50(xu363) and HT115 a very strong reduction of aggregates, in the bodywall muscles of the animals.

14. Figure S7 legend: *P. aeruginosa* should be italicized.

This has been corrected.

15. Figure S9 legend: *E. coli* should be italicized.

This has been corrected.

16. Figure S10 legend: The Figure S9 legend was mistakenly placed here.

This has been corrected.

17. Figure S12 title: Describe RNA-seq of what.

The title has been changed to: RNA-seq analysis of OP50, OP50(xu363), and HT115 *E. coli* strains.

18. Figure S12 legend: *E. coli* should be italicized.

This has been corrected.

19. Figure S15A: Panels were misaligned.

This has been corrected.

Reviewer #2 (Remarks to the Author):

The manuscript "Bacterial RNA promotes proteostasis through inter-tissue communication in *C. elegans*" by Emmanouil Kyriakakis and colleagues explores the impact of diet on healthy ageing in *C. elegans*. The study showed that *C. elegans* fed with HT115 bacteria have fewer polyQ40 aggregates in body wall muscle cells and exhibit less age-related paralysis compared to animals fed with OP50 bacteria. Moreover, an increase in LGG-1::GFP foci in seam cells and the aggravation of polyQ40 aggregation after silencing of autophagy genes in HT115-fed animals indicate a protective enhancement of autophagy in response to the HT115 diet. The beneficial effect of the HT115 diet appears to depend on the absence of ribonuclease III, an enzyme that cleaves dsRNA, in HT115 bacteria, since deletion of the *rnc* gene coding for ribonuclease III also rendered OP50 beneficial. Moreover, injection of

dsRNA extracted from HT115 bacteria was sufficient to reduce polyQ40 aggregation. The authors also showed that the beneficial effect depends on the RNAi machinery.

While the topic is intriguing, and the significance of this work would be high if dietary bacterial dsRNA actually triggered a systemic protective stress response that attenuated protein aggregation, slowed muscle atrophy and prolonged healthspan. However, the current data do not sufficiently support these conclusions. Several crucial controls are missing, requiring further experiments to validate the claims made. Therefore, this manuscript cannot be accepted in its current form.

1. My biggest concern is that there is hardly any polyQ40 signal in animals grown on HT115. This not only makes it very difficult to see the worms in the respective figures (an accompanying brightfield image would be helpful). It also indicates that not only aggregation but also the expression of polyQ40 seems to be reduced in these worms. If only aggregation was reduced, the soluble protein should still be visible. This suggests that the observed effects of bacterial dsRNA could stem from the endogenous system RNAi machinery affecting transgene expression. It is known that especially high copy transgenes can be affected by the endogenous RNAi machinery. This assumption is supported by the fact that mutations in RNAi machinery genes lead to a significantly higher number of polyQ40 aggregates (figure 4B: *sid-1* mutant animals have about 80 polyQ40 aggregates; in figure 3A and C WT animals have about 40-60 aggregates). Thus, the beneficial effect of HT115 could stem from induction of the endogenous RNAi machinery that silences the expression of the polyQ40 transgene.

Therefore, the following control experiments are necessary to prove that the HT115 diet indeed improves muscle proteostasis and does only affect the polyQ40 transgene:

A. Transgene expression must be tested by RT-PCR and Western Blot to make sure that polyQ40 RNA and protein levels are identical in animals fed with HT115 and OP50.

We acknowledge that in some of our current images, the signal from the polyQ40 is not very high in the HT115 or OP50(xu363) fed worms. This is because the signal of aggregates is very strong and we use the same exposure time for the same set of experiments. Below we provide enhanced images including Differential Interference

Contrast (DIC) images to make the animals' morphology clearer.

We do not think that the expression of the polyQ40 changes on the different diets because

- a. We observe that polyQ40 aggregates accumulate over time, even in worms fed HT115, supporting that the polyQ40 transgene is expressed in these animals.
- b. We also provide images of worms expressing polyQ24::YFP, driven by the same *unc-54* promoter (Fig. S2a). The polyQ24 animals show similar expression across all dietary conditions, including HT115. This result suggests that dietary dsRNA is unlikely to significantly alter polyQ40 expression, as polyQ24 (with fewer Q repeats) maintains comparable expression levels across conditions.

- c. Additionally, our proteomic data for expression of the *unc-54* promoter in the presence of HT115 and OP50(xu363) diets versus OP50 do not suggest a lower expression. If bacterial dsRNA affected transgene expression, it would also likely affect *unc-54* levels. However, we do not observe a decrease in *unc-54* expression in HT115 or OP50(xu363)-fed worms; if anything, we see a slight increase rather than a reduction in the AM141 polyQ40-expressing strain. Here are the quantified values: For AM141 strain \log_2FC HT115 vs OP50 = 0.39, \log_2FC HT115 vs OP50(xu363) = -0.07, \log_2FC OP50 vs OP50(xu363) = -0.46 and for wt N2 strain: \log_2FC HT115 vs OP50 = 0.21, \log_2FC HT115 vs OP50(xu363) = 0.06, \log_2FC OP50 vs OP50(xu363) = -0.14, indicating that the expression is relatively stable across diets in the wt animals.
- d. Finally, to show this more directly. We performed Western blot analyses comparing polyQ40 protein levels in worms fed with OP50, OP50(xu363), and HT115 (Fig. S6 and below). As negative control we used wt N2 worms grown on OP50. Alpha-tubulin was used as loading control.

These results indicate no significant differences in polyQ40::YFP expression, confirming that the observed dietary effects primarily impact aggregation rather than transgene expression levels. These combined findings suggest that the bacterial diet specifically influences the aggregation phenotype of polyQ40 rather than silencing its expression, supporting our interpretation that the observed reduction in aggregates is due to altered proteostasis rather than transgene suppression.

B. The effect of KD of the aggrephagy genes, and in particular the RNAi machinery must be tested in WT animals by assessing their effect on age-dependent paralysis.

We followed the recommendation and tested the effects of RNAi targeting autophagy and RNAi machinery genes in wild-type (*N2*) animals, as well as in AM141 polyQ40-expressing worms, to assess age-dependent paralysis.

When *vps-34*, a gene required for autophagy, was knocked down, we observed complete paralysis in older worms at a stage where control worms remained relatively active (Fig. S8 and below). Overall, *vps-34(RNAi)* animals were sicker than the control animals, further underscoring autophagy's critical role in maintaining health during aging. Knockdown of *sid-1*, which is integral to the RNAi machinery and the uptake of dsRNAs, also increased paralysis in worms vs. HT115. Similar results to HT115-fed worms were observed when worms were fed OP50(xu363). These data indicate the significant role of dsRNA on muscle function and paralysis resistance during aging.

We also checked it in AM141 animals, and similar effects were observed (Fig. S8 and below).

In addition, temperature-sensitive (ts) mutants (e.g. with the myosin(ts) [*unc-54(e1301)*] mutation) should be used as protein folding sensors and the effect of HT115 on ts phenotypes (movement, disruption of myosin filaments) should be assessed. If HT115 increases proteostasis capacity in muscle cells, then the muscle defect of ts mutant animals should be ameliorated.

We thank the reviewer for suggesting the use of temperature-sensitive (ts) mutants to test our hypothesis on proteostasis enhancement in muscle cells. We were very excited to try this out. Based on the recommendation, we ordered the *CB1301 unc-54(e1301)* strain, which harbors a temperature-sensitive mutation in the myosin gene (*unc-54*), from the *Caenorhabditis* Genetics Center (CGC).

It turned out the experiment was more complicated than we anticipated. We observed diet dependent mortality in that about 50% of *unc-54* mutant animals died as adults already at 20°C when fed on HT115. This death rate was reduced to about 10% on OP50 and OP50(xu363). While this diet-dependent mortality rate is interesting by itself, we did not follow it up. This also meant that other difference between the two types of *E. coli* strains impair worm physiology, for which there are already other examples in the literature (PMID: 32208418), but which also confounded the analysis.

We still observed that the surviving adults on HT115 diets had a slightly better motility than OP50. However, we did not observe the same trend with OP50(xu363) (figure below). Yet, we were only able to test 20°C and not higher temperatures because the already high mortality rate of *unc-54* mutant animals on HT115. Because of all these confounding issues, we decided to abandon this experiment and searched for other ways to show that muscle function is improved.

We turned to pharyngeal muscles, because there we can determine pharyngeal pumping rates. Pharyngeal pumping goes down during aging. We assessed pharyngeal pumping rates in eight-days old wild-type and polyQ40 animals fed on the different diets. In both wild-type and polyQ40 animals we observed a higher pharyngeal pumping rate when they were on a bacterial diet missing the ribonuclease 3 (rnC) gene (OP50(xu363) and HT115) compared to OP50, which expresses functional rnC. Thus, we can ameliorate muscle function with dsRNA diet, even when we do not challenge the animals with a proteotoxic stress. These data are included in the manuscript (Fig. S8c and d).

For wt N2:

For polyQ40-expressing worms AM141:

Furthermore, we monitored body wall muscle morphology of 8-day old worms. For this assay we used the MAH19 (*rff-1(pk1417) I*; *myo-3(st386) V*; *stEx30 [myo-3p::GFP::myo-3 + rol-6(su1006)]*) strain where the MYO-3 is fused with GFP and expressed under its own promoter, in a *myo-3* mutant background. Muscle morphology was classified into three categories (normal, intermediate and severe phenotype) (see below and Fig. S8e).

After growing the worms on the three diets muscle morphology was assed (below and Fig. S8f).

We observe a significant increase in the 'normal phenotype' on both OP50(xu363) and HT115 diets compared to the OP50 diet, suggesting a positive effect of those diets.

2. Bacterial density should be controlled for by measuring OD of the bacteria in key experiments to make sure that the effect of different bacteria is not a result of reduced food availability (which could increase autophagy).

We thank the reviewer for the valuable suggestion. There is always excess of food available on the plates and we never let them run out. In addition, we always ensure that the plates are similarly seeded for all bacterial diets, although it is true that OP50(xu363) grows a bit slower than the other strains. As requested, we have now measured the bacterial density in both the cultures and the NGM plates after seeding and washing off the bacteria. Our analysis shows no significant differences in bacterial density that could influence aggregation or autophagy induction. This indicates that the observed effects are not due to variations in food availability. In particular:

To feed the worms with bacterial cultures that are not intended to induce silencing, we streak the corresponding bacterial strain on an LB+amp plate and leave the bacteria to grow overnight at 37°C. Then single colonies are picked and inoculate in 3 ml of LB+amp. The next day, we apply 150 µl of this o/n culture on NGM plates.

The OD₆₀₀ of the overnight cultures was diluted 10 times in LB and then measured using a spectrophotometer.

OD₆₀₀ for 1/10 o/n cultures:

OP50 → 1st try 0.340, 2nd 0.317, 3rd 0.316 4th 0.298

OP50(xu363) → 1st try 0.331, 2nd 0.318, 3rd 0.328 4th 0.307

HT115 → 1st try 0.360, 2nd 0.368, 3rd 0.401, 4th 0.390

To be sure that the growth on the plates is not significantly different, we washed the cultures off the plates diluted three times in LB and again measured the OD₆₀₀.

OD₆₀₀ for cultures washed off the NGM plates and diluted three times:

OP50 → 1st try 0.519 2nd time 0.502 3rd 0.407

OP50(xu363) → 1st try 0.534 2nd 0.553 3rd 0.452

HT115 → 1st try 0.604 2nd 0.668 3rd 0.603

To feed the worm with RNAi constructs and induce silencing we use 30 µl of fresh overnight cultures in 2 ml LB. When about 3 h later and when the OD₆₀₀ is around 0.4

we add IPTG and apply 150 μ l in each plate. We then leave the plates overnight at room temperature and we use them the next day.

OD₆₀₀ for over-day cultures:

OP50 → 1st 0.420 2nd 0.208 3rd 0.302

OP50(xu363) → 1st 0.330 2nd 0.203 3rd 0.285

HT115 → 1st 0.417 2nd 0.380 3rd 0.354

If anything, the OP50(xu363) grows a bit slower than the rest of the strains, therefore it is unlikely that the effect of HT115 diet inducing autophagy is due to starvation of the animals because of low levels of food provided.

Again, to be sure that the growth on the plates is not significantly different, we washed the cultures off the plates, diluted them three times in LB and again measured the OD₆₀₀.

OD₆₀₀ for cultures washed off the NGM plates and diluted 3 times:

OP50 → 1st 0.578, 2nd 0.479, 3rd 0.482

OP50(xu363) → 1st 0.446, 2nd 0.403, 3rd 0.422

HT115 → 1st 0.544, 2nd 0.450, 3rd 0.498

We now mention in the material and methods section “All bacteria grew at similar rates, and we ensured excess of food for the worms during each experiment and for each diet”.

3. The authors claim systemic induction of autophagy, but only show increased LGG-1::GFP foci in the seam cells. The authors should at least also assess the number of LGG-1::GFP foci in muscle cells, since they observe an effect on polyQ40 aggregation in this tissue. Moreover, increased LGG-1::GFP foci could either result from an induction of autophagy or from a reduced autophagic degradation. The authors should therefore be more cautious with their descriptions or should measure autophagic flux and show that lysosomal degradation is not affected. The fact that their proteomics data does not show an upregulation of autophagy argues that it is not induced systemically.

To address the reviewer’s request, we assessed autophagy in muscle cells. We initially chose seam cells as a model because it is easier to monitor autophagy accurately in this distinct tissue at the L4 stage. Autophagy in muscle cells, by contrast, presents challenges: autophagy levels are generally very low in young muscle cells, which complicates quantification (see PMID: 25569839, PMID: 28675140). Additionally, the signal from other tissues can mask or overestimate the signal in muscle tissues. However, we conducted further analysis in muscle tissue by performing confocal microscopy on muscle cells of 8-day adult *C. elegans*. We used the MAH215 (mCherry::GFP::lgg-1) strain that expresses LGG-1 fused with mCherry and GFP under its own promoter. mCherry and GFP signals are both present on autophagosomes (yellow), whereas when autolysosomes are formed the GFP signal is quenched due to the more acidic environment and only the mCherry is fluorescing (red). In this way we can differentiate autophagosomes from autolysosomes. Our findings indicate that HT115 (and OP50(xu363)) diet led to a decrease in the relative GFP/mCherry ratio suggesting an increase of autolysosomes in muscle cells when compared to the OP50 diet (Fig. 2e, f, Fig. S7c and below). Thus, we observed an

increase in autophagy in muscle cells. Moreover, these results confirmed that the observed dietary effects on autophagy are not limited to a single cell type, supporting our hypothesis of a diet-driven modulation of systemic autophagic activity.

Arrows show some of the autophagosomes (both GFP and mCherry).

4. Figure 2E: why is there a “T1” in the figure?

We are sorry for this error. T1 comes from a “stray” HT115 label, hiding behind the graph and is now deleted.

5. Figure 3G: Quantification of polyQ40 aggregation is missing. To substantiate their claim, the authors should also inject RNA isolated from OP50 and OP50(xu363) and examine the effects on polyQ40 aggregation.

We thank the reviewer for raising this point. We performed the suggested experiment, injecting RNA from each bacterial strain, and found that RNA from all bacteria had a similar impact on aggregate levels in progeny of injected worms compared to the ones injected with water (Fig. S9 and below). We also see the distribution of progeny with different number of aggregates, since the efficiency of injection is not always the same in all progeny. We usually picked around 40-60 early progeny in each round of injection.

Although this initially seems to contradict our ingestion-based findings, it makes sense considering that all three bacterial strains contain the same dsRNA. However, a key difference lies in how this dsRNA is processed based on the presence of *rnc*.

In strains with an intact *rnc* gene, such as OP50, Ribonuclease 3 (RNase III) precisely cleaves dsRNA, resulting in smaller, digest-inactive fragments that are less effective upon ingestion. Conversely, in *rnc*-mutant strains like OP50(xu363) and HT115, dsRNA is not processed into smaller fragments, which preserves its activity through ingestion and promotes its anti-aggregation effect. Literature supports that longer dsRNA fragments tend to induce more robust RNAi effects than their shorter counterparts during ingestion (PMID: 12970568 & PMID: 26067272). When dsRNA is injected directly into *C. elegans* tissue, RNA length does not impact silencing efficiency as it does with ingestion. Moreover, we cannot exclude that RNA isolated from OP50 would partially basepair and hence provide also dsRNA species. Both scenarios explain why injection results were similar across all strains.

6. The text describing Figure 4F is misleading. In line 212, the authors write that “a mutant in *PPW-1* accumulated polyQ40 aggregates in muscle cells, independent of the diet”. However, the data in Fig. 4F show that the influence of food is statistically significant.

We thank the reviewer for this comment. The data in Fig. 4f does show that the OP50(xu363) diet significantly differs from the others in terms of aggregate accumulation, and our original wording may not accurately convey this. Our intent was to emphasize that aggregate accumulation persists across all diets in the *ppw-1* mutant, but we see how our statement could be misleading.

To address this, we revised the text and now state: “In the *ppw-1* mutant, the number of aggregates remains high across all three diets, ...”

7. Figure 5: It is surprising that the authors chose to focus their analysis on genes that are differentially expressed in polyQ40 vs. WT animals. It is known that polyQ40 affects muscle structure, and therefore it is not surprising that genes involved in muscle structure are upregulated, nor that KD of these genes worsens polyQ40 aggregation. However, the beneficial effect of bacterial dsRNA should be independent of polyQ40. polyQ40 should only be a proteostasis sensor (as they state in the beginning), and therefore genes affected by polyQ40 expression should not be considered. The correct comparison would therefore be between WT animals grown on OP50 vs HT115 or better OP50 vs OP50(xu363). Genes that are only expressed in animals grown on OP50(xu363) and not in animals grown on OP50 plates should be tested for their positive effect on proteostasis (and not only polyQ40 aggregation).

We thank the reviewer for this insightful comment. The polyQ40 strain indeed serves as a sensor for proteostasis disruptions; however, it also presents a stress condition due to the toxic nature of the aggregates in the cells. Initially, our approach was to compare the dietary effects by identifying proteins that would differ between the OP50, OP50(xu363), and HT115 diets across both polyQ40 and wild-type (WT) strains. We did this also to check whether we specifically lose a protein/a set of proteins in OP50(xu363) and HT115 diets, indicative that the bacterial dsRNA could

act as siRNA. However, we could not detect such changes. These results are consistent with our bacterial RNAseq data in which we also did not detect specific RNAs that could act as siRNAs.

However, we observed that changes to the proteome were more pronounced in the polyQ40 strain than in WT animals, likely because the dsRNA-mediated protection against aggregate accumulation primarily becomes evident in a compromised proteostasis environment (as in the polyQ40 strain). In contrast, WT animals at this young age are not under stress, which may explain the minimal dietary effects on their proteostasis markers.

Although we considered the comparisons suggested, our GO term analysis did not reveal other significant changes in WT animals related specifically to dsRNA response with this setup. This observation is consistent with our findings in Figure 5C, where the volcano plot shows only a few differential expression changes in WT animals (OP50 vs OP50(xu363)), underscoring the limited impact of the dietary dsRNA on WT gene expression under non-stress conditions. We would gladly identify additional significant pathways and proteins, if possible, yet only muscle-specific pathways emerged in our analysis. This finding remains highly informative, highlighting the importance of muscle health under stress and the role of diet in maintaining it. These results underscore the necessity of preserving muscle function to effectively manage cellular stress, which could be essential in extending healthspan.

8. Statistics

Some of the mean values appear to differ only slightly, but are nevertheless statistically highly significant. Unfortunately, there is no information about which multiple comparison test was used in the figure legends. There is also no information about which correction was used for the multiple comparison test. Please specify.

Also, information about the number of animals tested in each replicate or experiment is missing. Please add.

The figure legends have been updated to include detailed information on the multiple comparison tests used, including the type of correction applied. Additionally, we have now added the number of animals tested.

9. Supplementary figures: all figures concerning one main figure should be combined to increase readability.

Where possible, supplementary figures have been combined as suggested. However, due to the volume of data added throughout the revisions, consolidating all figures into a single supplementary figure was not feasible. Previous supplementary figure S1 and S2, S3 and S4, S8 and S9, have been merged.

10. Figure S2: A positive control for methylcobalamin is missing. Is it actually taken up and active? The description in the methods section suggests that the assay was done differently than in the cited paper (e.g., different methylcobalamin concentration, dead vs. alive OP50). The authors need to provide evidence that methylcobalamin is active (e.g., by repeating some of the experiments from the cited paper assessing mitochondrial health).

As the reviewer suggested, we checked whether methylcobalamin is active in our setup. It has been shown by Revtovich *et al.*, (PMID: 30865620), that methylcobalamin supplementation improves resistance to peroxide stress. To make sure that the methylcobalamin supplementation is effective, we monitored survival and developmental rate in the presence of hydrogen peroxide (Thermo Scientific) (0, 10% and 20%). The hydrogen peroxide-induced death was significantly rescued in the presence of methylcobalamin (Fig. S1c and below), recapitulating the data from Revtovich *et al.*

Interestingly, the developmental rate of their progeny was also affected by hydrogen peroxide and significantly increased in the presence of methylcobalamin (Fig. S1d-e and below).

The data are now included in the manuscript, showing that methylcobalamin supplementation is effective and it is not responsible for the effects in aggregation.

11. In the results section (line 123), the text describes polyQ40 aggregation, but the figure S5D is labelled Q35: which strain was used in this experiment?

The polyQ40 strain is the one that is developmentally arrested. Here we want to show that the polyQ repeats are responsible for that. Specifically, with shorter, non-toxic repeats like Q0 and Q24, no developmental defects are observed. However, at Q35, where toxicity begins, there is a noticeable delay, and in the highly toxic polyQ40, the worms arrest entirely. To clarify this, we have revised the text to better explain the progression of toxicity with increasing repeat lengths. The revised text now reads: "This effect was dependent on the expression of the number of polyQ repeats in muscle cells. Specifically, with shorter, non-toxic repeats like Q0 and polyQ24, no developmental defects were observed. However, at polyQ35, where toxicity begins, there was a noticeable delay, and in the highly toxic polyQ40, the worms arrested entirely."

12. Figure S6B: The statistics should also include a comparison between dead and alive bacteria, as it was done in figure S6D.

The statistics are now included as recommended.

13. Figure S9: there are two figures labelled S9.

We apologize for the mistake. This is now corrected.

14. Figure S13: this type of analysis should be part of the main figure and is more informative than the volcano plots in the main figure.

Figure 5 and S13 have been rearranged.

Reviewer #3 (Remarks to the Author):

In general, the manuscript entailed “Bacterial RNA promotes proteostasis through inter-tissue communication in *C. elegans*” described an interesting story about the role of bacterial-derived RNAs in *C. elegans*. DIA-based proteomics analyses were applied to investigate the effects of different diets on the proteomes of different types of worms. The experiments were well-designed and results were interesting. However, some issues in the proteomic parts should be clarified and supplied.

We wish to thank the reviewer for their positive assessment!

1. Since I can't get access to the raw MS data and identification information of the DIA-based proteomics, no information about the biological replicates of MS experiments and quantification results of the DIA proteomic analysis was provided in the manuscript.

We sincerely apologize for the inconvenience in accessing the raw MS data. All of the relevant data, including the biological replicates and quantification results from the DIA-based proteomic analysis, are available on the MassIVE repository, via the following link, which was provided in the initial submission and is still active:

<http://massive.ucsd.edu/ProteoSAFe/status.jsp?task=9775373687f4439b80b87162378b63af>

The password is PCF. We hope this resolves the issue and that the reviewers can access the data.

2. For quantitative proteomic data analysis, detailed information should be provided, for example, what were the search parameters for Spectronaut? Whether cross-run normalization was applied for the data analysis? Whether imputation for missing values was performed and how?

We thank the reviewer for this comment. A .txt file, named SpectronautSettings, is now provided as supplemental data, which contains the complete search setting/parameters used for Spectronaut. Additionally, Materials and Methods have been updated accordingly, addressing the normalization as well as the imputation of the data.

3. The cutoffs for screening of differentially-expressed proteins in the DIA-based proteomic analysis wasn't provided in the manuscript.

The cutoffs for screening were provided in the figure legend for Fig. 5 and in the Materials and Methods in the section GO term analysis and functional association networks. For clarification, a q-value of less than 0.05 was used to filter significant changes (or log₁₀ value of 1.3) shown as blue dots in the volcano plots and the

cutoff of -0.3 and +0.3 fold change (log2ratio) were used, depicted by the dotted lines.

4. Heatmap and principle components analysis are important to demonstrate the reproducibility and clusters of proteomic data. The author should provide this information in the manuscript.

We thank the reviewer for this comment. As suggested, we now include the heatmap and the principle component analysis of the proteomic data (Fig. S12). The description is provided in Materials and Methods.

5. In the manuscript, only 84 proteins that were significantly up-regulated were analyzed with GO enrichment analysis and STRING network analysis. How about the 110 down-regulated proteins? What are the roles of these down-regulated proteins in the different diets?

Interestingly enough, the 110 down-regulated proteins did not result into something worth following up. Therefore, we decided to not include this information. Below is the GO term analysis of these 110 proteins.

Image generated by ShinyGO 0.77

Reviewer #4 (Remarks to the Author):

The authors of this manuscript "Bacterial RNA promotes proteostasis through inter-tissue communication in *C. elegans*" investigated how OP50 and HT115, two *E. coli* strains commonly used as food for *C. elegans* worms raised in the laboratory. A lot of efforts were put into this study, as can be seen from the number of displayed items (5 figures in the main text and 16 supp figures) and the quality of the figures. However, after careful reading of this manuscript, I think that the main text, abstract included, needs a major revision. My concerns are as follows.

We thank the reviewer for the recognition of the amount of work and the quality of data.

Major concerns:

1. This study tries to establish a mechanism that starts with dsRNA in the bacterial diet (component #1). Ingested dietary dsRNA is processed by the RNAi machinery in the worm body (component #2), leading to activation of autophagy/aggrephagy (component #3) and consequently, less polyQ40::YFP aggregation in body wall muscles (BWMs) (component #4). This is thought to be a systemic effect that somehow requires intact RNAi function in the germline, in addition to that in the intestine. Component #1 is backed up by overwhelming evidence that the authors have gathered from elaborate experiments. Component #2 is also on solid footing, although the role of RNAi in the germline and other non-intestinal tissues is unclear. Component #3 is weak. The only data that supports it is an increase in the number of GFP::LGG-1 puncta in the seam cells of worms raised on HT115 relative to that of worms raised on OP50 (Fig. 2C-D). Whether autophagy/aggrephagy is activated in BWMs of worms raised on HT115 was not examined. The other data (Fig. 2 A-B, E-G, and Fig. S5), although sizable, only show that autophagy/aggrephagy is required for worms feeding on HT115 to keep polyQ40::YFP aggregation at a low level. They lend no direct support to activation of autophagy/aggrephagy in BWMs, neither do RNA-seq or proteomic data (Fig. S12 and Fig. S13). It may be right that autophagy is elevated in worms on HT115 diet and it is possible that enhanced autophagy is realized through post-translational regulation, but to this end there is not sufficient experimental evidence for it. Component #4 is all right by itself but becomes problematic when it is treated as if it were the same as health span, as discussed below. The wording of the abstract smoothed over the gaps to the effect that it's easy for readers to connect the dots in their mind when there is not yet a line connecting them.

We reworded the abstract and we reduced the reference to healthspan, rather stating effects on proteostasis.

To address the reviewer's request about autophagy, we assessed autophagy in muscle cells. We initially chose seam cells as a model because it is easier to monitor autophagy accurately in this distinct tissue at the L4 stage. Autophagy in muscle cells, by contrast, presents challenges: autophagy levels are generally very low in young muscle cells, which complicates quantification (see PMID: 25569839, PMID: 28675140). Additionally, the signal from other tissues can mask or overestimate the signal in muscle tissues. However, we conducted further analysis in muscle tissue by performing confocal microscopy on muscle cells of day 8 adult *C. elegans*. We used the MAH215 (mCherry::GFP::lgg-1) strain that expresses LGG-1 fused with mCherry and GFP under its own promoter. mCherry and GFP signal is present on autophagosomes, whereas when autolysosomes are formed the GFP signal is quenched due to the more acidic environment and only the mCherry is present. In this way we can differentiate autophagosomes from autolysosomes. Our findings

indicate that HT115 (and OP50(xu363)) diet led to a decrease in the relative GFP/mCherry ratio suggesting an increase in the amount of LGG-1 that arrived in autolysosomes in muscle cells, when compared to the OP50 diet (Fig. 2e, f, Fig. S7c and below). Thus, we observed an increase in autophagy in muscle cells. Moreover, these results confirmed that the observed dietary effects on autophagy are not limited to a single cell type, supporting our hypothesis of a diet-driven modulation of systemic autophagic activity.

Arrows show some of the autophagosomes (both GFP and mCherry).

2. Health span can be measured using different assays. The measurement of health span in this study relies almost exclusively on aggregation of polyQ40::YFP in BWMs. I think it is more appropriate to deemphasize health span and adjust the wording towards protection against proteotoxicity, proteostasis, or the like.

We agree that our study mainly focuses on aggregation. However, we have the data on worm motility in advanced age, which we have expanded and include more conditions and we now also added two more assays.

We assessed pharyngeal pumping rates in eight-days old wild-type and polyQ40 animals fed on the different diets. In both wild-type and polyQ40 animals we observed a higher pharyngeal pumping rate when they were on a bacterial diet missing the ribonuclease 3 (rnC) gene (OP50(xu363) and HT115) compared to OP50, which expresses functional rnC. Thus, we can ameliorate muscle function with dsRNA diet, even when we do not challenge the animals with a proteotoxic stress. These data are included in the manuscript (Fig. S8c and d).

For wt N2:

For polyQ40-expressing worms AM141:

Furthermore, we monitored body wall muscle morphology of 8-day old worms. For this assay we used the MAH19 (*rrf-1(pk1417) I*; *myo-3(st386) V*; *stEx30 [myo-3p::GFP::myo-3 + rol-6(su1006)]*) strain where the MYO-3 is fused with GFP and expressed under its own promoter, in a *myo-3* mutant background. Muscle morphology was classified into three categories (normal, intermediate and severe phenotype) (see below and Fig. S8e).

After growing the worms on the three diets muscle morphology was assessed (below and Fig. S8f).

We observe a significant difference in both OP50(xu363) and HT115 conditions versus the OP50 in the normal phenotype, suggesting a positive effect of those diets.

Still as this reviewer suggested, we de-emphasize our conclusions when possible from healthspan to proteostasis or similar, starting from the abstract, where we now mention “we show that dietary RNA species improve proteostasis in *C. elegans*”, instead of “we show that dietary RNA species extend healthspan in *C. elegans*”.

3. The experimental evidence in Fig. 5 does not, in my opinion, support the conclusion that “the dietary RNA mediates this protective effect by increasing the levels of key proteins required for muscle function,” unless RNAi reduced the abundance of a targeted UNC protein to the same level as that found in worms reared on OP50. In addition, genes knocked down or mutated in Fig. 5 I-N do not correspond to proteins up-regulated in worms raised on HT115 or OP50(xu363).

We acknowledge that not all muscle-related proteins may have been detected in the proteomic analysis due to technical limitations, such as incomplete coverage of the entire *C. elegans* proteome. Additionally, some proteins critical for muscle function may not show significant changes in abundance despite having functional importance. However, our proteomic analysis revealed a clear increase in the levels of specific muscle proteins, involved in sarcomere structure and muscle function, when worms were fed OP50(xu363) or HT115 compared to OP50. The experiments presented in Fig. 5 demonstrate that the protective effect of these diets on protein aggregation is abrogated when these key muscle proteins are downregulated or absent, supporting our hypothesis. To strengthen our hypothesis that muscle function

and sarcomere integrity are critical, we further knocked out (KO) or knocked down (KD) additional muscle-related genes.

To address the potential confusion, we revised the statement in the text to more accurately reflect the evidence and implications of our findings. The conclusion now reads:

“Moreover, we find that ribonuclease 3-deficient bacteria increased the levels of proteins required for muscle function. Reducing or abrogating the function of these proteins reversed the positive effect of the diet and aggregates accumulated.”

Minor problems

1. L246, the section title “Body-wall muscle contraction and sarcomere integrity alleviate accumulation of aggregates” is difficult to understand.

The title has been changed and it now reads: “Body-wall muscle contraction and sarcomere integrity are required for the reduction of aggregates”.

2. L182-184, “We confirmed that similar to HT115, the loss of aggregate formation on OP50(xu363) diet was also due to autophagy, and more specifically to aggrephagy (Fig. S10).” Conclusion is less than precise, for it implies that autophagy is THE reason, rather than A reason. I think the data show that autophagy, more specifically aggrephagy, is required for the low-polyQ-aggregation effect of the HT115 or OP50(xu363) diet.

We thank the reviewer for this suggestion. We agree that the statement is strong and we rephrased it accordingly. Now the sentence reads: “We confirmed that similar to HT115 diet data, autophagy, and more specifically aggrephagy is required to maintain the low polyQ40 aggregation effect on the OP50(xu363) diet (Fig. S7a-c).”

3. Grammar errors in the titles of Figure 5 and Figure S14.

We apologize for the errors; they have been corrected.

4. Title of Fig. S9, change to “Ribonuclease 3 depletion protects *C. elegans* from protein aggregation.”

Figures S8 and S9 have been merged and so did their title.

5. Fig. S10 is mislabeled as Fig. S9 with a mismatched figure legend.

We thank the reviewer for spotting this error, which has now been corrected.

6. Remove “T1” from Fig. 2E.

The stray HT115 label has been removed.

7. Different laboratories reported different effects of HT115 vs OP50 on *C. elegans* lifespan and brood size. For example, in contrast to what’s seen in this study, PMID 33159120 reported no difference in brood size but a markedly longer lifespan for worms on HT115 diet relative to OP50 (table 1). Can the authors compare the different results and discuss what might be causing the discrepancy?

The discrepancies in the reported effects of HT115 versus OP50 diets on *C. elegans* lifespan and brood size, we believe are multifaceted and arise from variations in experimental protocols and environmental factors. Several factors might explain these differences:

1. **Lifespan Measurement Protocols:** Differences in when lifespan monitoring begins (e.g., from egg versus adulthood) can impact results. Lifespan measurements initiated at different life stages may capture varying environmental and developmental influences.
2. **Temperature Variability:** Temperature plays a crucial role, as some studies observed effects of HT115 versus OP50 at specific temperatures (e.g., small trends (smaller than in the Sturh & Curran study PMID: 33159120) at 20°C and 25°C that disappear at 16°C) (PMID: 25040785). Temperature fluctuations during assays might amplify or mask bacterial diet effects.
3. **Dietary Bacterial Preparation and Growth:** Variations in the preparation of NGM plates, autoclaving steps, and even water quality used during the preparation, can influence bacterial growth and its subsequent effect on worm physiology. We are actually testing this hypothesis currently and our preliminary data suggest significant impacts of such methodological details, which may differ across laboratories. Especially, autoclaving steps during the preparation of NGM plates influences bacterial growth and *C. elegans* development, physiology and lifespan. Since these results are out of the scope of this study we are planning to report them in a separate manuscript in the near future.
4. **Worm Handling:** Worm handling differs across labs. These factors could lead to inconsistencies in lifespan and reproductive outcomes

Although a longer lifespan in HT115 would be “better fitting” in our study and could have enhanced its physiological relevance, our experimental set up does not show a significant difference in the lifespan of wt worms when fed the different diets. To our knowledge the biggest difference is reported by the Sturh & Curran study (PMID: 33159120), whereas in other studies smaller or no differences were reported (PMID: 20520844, PMID: 30125273, PMID: 19844570, PMID: 25040785). More explanations in inter-lab variance in *C. elegans* lifespan, are published in the Urban et al., 2022 (PMID: 34793939).

Response to reviewers' comments

We wish to thank the reviewers for their positive and encouraging comments on our manuscripts. We addressed the remaining comments below.

Reviewer #1 (Remarks to the Author):

The authors have addressed all of my comments with additional experiments. The paper was significantly improved.

Thank you for the positive assessment of our paper!

I have a few minor comments:

-Line 45: I thought that the influence of dietary restriction on the lifespan was first reported in rodents in 1917 (Green et al., Nat Rev, 2022).

<https://www.nature.com/articles/s41580-021-00411-4>

Thank you for bringing this up, we changed the introduction and mention now Osborne's work in 1917.

-Figure 4B-F. What I meant by my original comment is as follows. I believe the authors want to compare the effects of diet on WT and mutants. For example, the dietary effect in the wt (difference between OP50 and OP50(xu363) in Fig 3a) is much larger than the dietary effect in the rde-2 mutants (difference between OP50 and OP50(xu363) in Fig 4e). I had no problem with their conclusion from the beginning because the data was clear. However, OP50 and OP50(xu363) are statistically different in rde-2. Thus, I thought it might be better to compare the effects of diet on WT and mutants. Alternatively, OP50(xu363) condition can be compared between wt (Fig. 3a) and sid-1(Fig. 3b) etc. If it is not appropriate to compare these, the authors can weaken the statement slightly.

We appreciate this comment. Since these experiments were not performed in parallel, it would not be entirely fair to directly compare them within a single graph. Our primary message is that in these mutants, the diet-dependent effect is not that strong. However, we acknowledge the reviewer's point and therefore we decided to weaken the statement.

Reviewer #2 (Remarks to the Author):

The manuscript "Bacterial RNA promotes proteostasis through inter-tissue communication in *C. elegans*" by Kyriakakis and colleagues explores the impact of the bacterial diet on proteostasis in *C. elegans*. The study showed that *C. elegans* fed with HT115 bacteria had fewer polyQ40 aggregates in body wall muscle cells and exhibited less age-related paralysis compared to animals fed with OP50 bacteria. Moreover, an increase in LGG-1::GFP foci formation and the aggravation of polyQ40 aggregation after silencing of aggrephagy genes in HT115-fed animals suggested a protective enhancement of autophagy in response to the HT115 diet. The beneficial effect of the HT115 diet appears to depend on the absence of ribonuclease III, an enzyme that cleaves dsRNA, in HT115 bacteria, since deletion of the *rnc* gene coding for ribonuclease III also rendered OP50 beneficial. The authors also showed that the beneficial

effect highly depends on the RNAi machinery.

Although the authors have addressed some concerns in their revised submission by incorporating additional experimental data, several critical issues remain unresolved and require further clarification. Please find my additional comments below.

We appreciate the time and effort in reviewing our manuscript and providing detailed feedback.

Regarding point 1A:

- Authors' answer: "We observe that polyQ40 aggregates accumulate over time, even in worms fed HT115, supporting that the polyQ40 transgene is expressed in these animals."

This observation is precisely what one would expect if polyQ40 protein expression is reduced. Given that the polyQ40 transgene is a high-copy transgene, RNAi-mediated KD would likely not fully suppress its expression.

We acknowledge the concerns regarding the expression levels of polyQ40 and the potential impact of bacterial diet on transgene expression. However, we respectfully disagree with the interpretation of our data and would like to clarify our findings.

The initial concern raised was that "there is hardly any polyQ40 signal in animals grown on HT115," suggesting that not only aggregation but also expression of polyQ40 is reduced. In response to this, we have provided several lines of evidence demonstrating that polyQ40 is indeed expressed in worms across all three dietary conditions. These include fluorescence microscopy images and Western blot analyses, all of which showed that polyQ40::YFP expression levels are maintained across the dietary conditions. Another piece of evidence was the fact that aggregates accumulate during ageing also in HT115 diets, indicating the polyQ40 is expressed under these conditions and can form aggregates.

In response to the renewed reviewer's concern: we performed qPCR (as requested by the reviewer, below) and repeated the western blot analysis with worm lysates that were subjected neither to centrifugation nor to sonication. No matter what analysis we performed, we could not detect a reduction in polyQ40 in worms fed on HT115 when compared to OP50, or to OP50(xu363). Thus, polyQ40 levels do not vary under the dietary conditions tested.

- Authors' answer: "We also provide images of worms expressing polyQ24::YFP, driven by the same unc-54 promoter (Fig. S2a). The polyQ24 animals show similar expression across all dietary conditions, including HT115. This result suggests that dietary dsRNA is unlikely to significantly alter polyQ40 expression, as polyQ24 (with fewer Q repeats) maintains comparable expression levels across conditions."

Similarly, polyQ24::YFP is a high-copy transgene, and RNAi-mediated knockdown would likely result in only a slight reduction in its expression. A 20-30% reduction in expression would not be readily detectable by microscopy. However, even such a modest reduction could significantly delay aggregation, as this process is highly sensitive to protein levels.

First of all, we would like to point out that RNAi efficiency is not correlated with expression levels of the targeted gene. In our hands, GFP (or YFP) tagged proteins are very efficiently silenced, when we use RNAi against GFP (PMID: 35941155); we actually commonly dilute the bacteria expressing GFP dsRNA 1:250 or 1:500 with other bacteria because otherwise the silencing is too strong. Likewise, when we perform RNAi against *C. elegans* ubiquitin, we dilute the bacteria

expressing ubi-1 dsRNA 1:250 with other bacteria to avoid a lethal phenotype (PMID: 39910226). And there are probably many more examples... While it is conceivable that some highly expressed genes are not efficiently silenced, this is by no means the rule. Regarding the point that a 20-30% reduction in expression would not be readily detectable by microscopy but could still significantly delay aggregation, we would like to highlight that no such reduction has been observed. A 30% reduction in fluorescence signal is easily detectable by microscopy, and certainly also by western blot. Furthermore, the assumption that such a modest reduction would meaningfully impact aggregation remains speculative.

We are not entirely sure we understand the reviewer correctly, but we assume that the reviewer is concerned that bacterial-derived RNA species would specifically knock-down polyQ40-YFP. As pointed out above, we do not observe a reduction in polyQ40-YFP, but also not in polyQ24-YFP. If the RNA species would target the polyQ or the YFP of the transgene, in both cases we should observe a reduction of polyQ-YFP levels. However, there are no dietary-related changes to the mRNA or protein levels detectable for polyQ24-YFP and polyQ40-YFP. PolyQ24-YFP does not lead to the formation of aggregates in *C. elegans* muscle cells and serves therefore as perfect tool to rule out that bacterial RNA species generated in HT115 and OP50(xu363) cause a reduction of polyQ40-YFP.

Let's be clear: we would be over the moon, if we had any evidence that bacterial derived RNA species would specifically reduce polyQ40 levels in *C. elegans* muscle. The clinical relevance and the therapeutical potential would be enormous. Unfortunately, this scenario is not supported by our data.

- Authors' answer:" Additionally, our proteomic data for expression of the unc-54 promoter in the presence of HT115 and OP50(xu363) diets versus OP50 do not suggest a lower expression. If bacterial dsRNA affected transgene expression, it would also likely affect unc-54 levels. However, we do not observe a decrease in unc-54 expression in HT115 or OP50(xu363)-fed worms; if anything, we see a slight increase rather than a reduction in the AM141 polyQ40-expressing strain. Here are the quantified values: For AM141 strain \log_2FC HT115 vs OP50 = 0.39, \log_2FC HT115 vs OP50(xu363) = -0.07, \log_2FC OP50 vs OP50(xu363) = -0.46 and for wt N2 strain: \log_2FC HT115 vs OP50 = 0.21, \log_2FC HT115 vs OP50(xu363) = 0.06, \log_2FC OP50 vs OP50(xu363) = -0.14, indicating that the expression is relatively stable across diets in the wt animals."

Why would KD of polyQ40 also impact endogenous unc-54 levels? The KD is sequence-specific and would target only the cDNA coding for polyQ40::YFP, without influencing expression driven by the unc-54 promoter.

We did not really understand in the previous round that this reviewer suggested the possibility that diet-derived RNA species would KD polyQ40::YFP. Since the transgene is driven by the unc-54 promoter, we argued that there is unlikely a reduction of polyQ40 levels due to a reduction in transcription. We referred to our proteomic data primarily to emphasize that polyQ40 protein levels do not appear to be significantly altered, and almost certainly not due to promoter regulation.

- Authors' answer: "Finally, to show this more directly. We performed Western blot analyses comparing polyQ40 protein levels in worms fed with OP50, OP50(xu363), and HT115 (Fig. S6 and below). As negative control we used wt N2 worms grown on OP50. Alpha-tubulin was used as loading control... These results indicate no significant differences in polyQ40::YFP

expression, confirming that the observed dietary effects primarily impact aggregation rather than transgene expression levels. These combined findings suggest that the bacterial diet specifically influences the aggregation phenotype of polyQ40 rather than silencing its expression, supporting our interpretation that the observed reduction in aggregates is due to altered proteostasis rather than transgene suppression.”

The authors’ procedure for analyzing polyQ40 levels presents significant limitations. Notably, sonication (which they used for lysis) can promote protein aggregation (PMID: 15459333), and the high-speed centrifugation of lysates described in the methods is likely to pellet polyQ40 aggregates, potentially underestimating total protein levels in the Western blot analysis. This concern is reinforced by the observation of a single, distinct Q40-YFP band in their Western blots, characteristic of soluble proteins. In contrast, previous studies using this construct frequently report smears or protein fractions trapped in gel pockets during electrophoresis, which is typical for aggregation-prone proteins.

The authors must provide evidence that their lysis procedure does not exclude polyQ40 aggregates and confirm that the total protein, not just the soluble fraction, is loaded onto the gel. Centrifugation at lower speeds - or avoiding it entirely - should be considered, and the pellet fraction should also be analyzed to ensure the absence of polyQ40 aggregates. Additionally, performing the suggested RT-PCR analysis of polyQ40 expression could have offered valuable validation without the complications associated with protein aggregation.

Our rationale for performing Western blot analysis rather than RT-PCR was that protein levels provide a more direct measure of polyQ40 levels. We understand now better the reviewer’s rational that dietary RNA species would specifically target polyQ40::YFP. Even though this should be translated into a strong reduction of polyQ40::YFP protein levels, we see the point to perform a RT-PCR analysis, which we did. In addition, we generated worm lysates without sonification and without centrifugation. These lysates were subjected to western blot analysis.

As the reviewer will be able to appreciate, we did not observe any significant diet-dependent differences, indicating that dietary intervention does not influence polyQ40::YFP mRNA levels (Fig. S6d and below). This further supports our conclusion that the observed effects in aggregation are not due to different expression levels.

Another concern raised was the potential for sonication to promote protein aggregation. While we acknowledge this possibility, we do not see how this effect would be diet-dependent. Also, to be able to see the already formed aggregates, we repeated the Western blot analysis (below and Fig. S6a-c), as suggested. Sonication and centrifugation were completely avoided. By omitting these steps, we ensured that neither soluble proteins were artificially aggregated nor pre-existing aggregates were removed. The results confirmed that soluble polyQ40::YFP levels remain similar across all dietary conditions.

Western Blot (left) and analysis (right) of polyglutamine in WT, polyQ24::YFP (Q24) and polyQ40::YFP (Q40) expressing worms. Antibodies for YFP and α -tubulin were used. S.e. (short exposure), l.e. (long exposure)

As per the reviewer's request, we also retained the stacking gel during transfer to visualize aggregated proteins (Fig. S6b and below). To further validate this, we analyzed a strain expressing polyQ24::YFP, which does not form aggregates. In this strain, we observed equal expression across all three dietary conditions, confirming that diet does not influence the polyQ40 expression levels.

Full blot stained for YFP and long (l.e.) exposure. At the top part we observed a faint smudge only in the OP50 fed polyQ40::YFP-expressing strain.

Together with the RT-PCR data, we are now confident that diet does not alter polyQ40::YFP expression levels, but instead affects its aggregation dynamics. We appreciate the reviewer's comments, as they have strengthened our analysis and allowed us to refine our conclusions. These results have been added into the revised manuscript.

Regarding point 1B:

The authors report a diet-dependent mortality in approximately 50% of CB1301 *unc-54(e1301)* mutant animals when fed on HT115 bacteria, which raises concerns regarding their experimental conditions. This strain has been utilized in numerous studies, including RNAi experiments involving HT115 bacteria (e.g., PMID: 32217667, PMID: 39128917), without similar issues. Such mortality could indicate confounding factors unrelated to proteostasis, potentially compromising the reliability of their observations.

We respectfully disagree with the reviewer's interpretation. None of the studies actually measures diet-dependent mortality. So, there is no way of knowing whether or not they observed some mortality on HT115. Moreover, our experimental setup is different in that we keep the animals at a constant temperature at 20°C and do not perform the heat shocks. The reason, we did not perform heat shock was to keep the conditions comparable with the rest experiments. This means the experiments in the two publications and in our manuscript were done in different ways and are therefore not necessarily directly comparable. Again, we would like to stress, that in the papers referenced by the reviewer a direct comparison between different diets was not performed.

The alternative experiments presented by the authors to address these concerns are not convincing. The pharyngeal pumping rate, which they measured, is influenced by a variety of factors beyond proteostasis, including bacterial diet (PMID: 8462849). Therefore, it is not a reliable approach to assess proteostasis in the context of a different bacterial diet. Additionally, the observed improvement in muscle morphology is minimal and correlates only marginally with

the slightly reduced paralysis observed in aged wild-type worms. This provides insufficient evidence to conclude that proteostasis is substantially enhanced in HT115-fed non-transgenic animals and therefore my concern remains that the reported effects might be specific to the high-copy polyQ40 transgene.

To evaluate proteostasis in non-transgenic animals, the authors should consider testing additional ts mutants (e.g., as described in PMID: 22242008, PMID: 16469881) to verify that their findings are not limited to transgene-specific effects. Especially, since the study now emphasizes a role of the HT115 diet in enhancing proteostasis rather than improving aging, it is crucial to rigorously confirm that proteostasis is indeed improved.

We appreciate the reviewer's feedback and the suggestion to further assess proteostasis in non-transgenic animals. However, we would like to clarify several points regarding the concerns raised.

First, the reference to Avery et al. (PMID: 8462849) does not quantitatively measure pharyngeal pumping but merely mentions, as unpublished observations, "*Mutants with a starved appearance were examined by Nomarski microscopy, and only those with abnormal feeding motions were kept. Constipated, egg laying-defective, or flaccid paralyzed (ie., probable body wall muscle-defective) mutants whose pumping was slow were not kept, since these defects can cause slow pumping (my unpublished observations)*". that certain mutants with abnormal feeding motions were excluded from further analysis after the initial screen. This does not constitute direct evidence that bacterial diet broadly and systematically affects pharyngeal pumping in a way that invalidates its use as a readout in our study. Furthermore, our pharyngeal pumping analysis was performed in aged worms, where quality control mechanisms decline with age, making them more susceptible to proteostasis-related effects. Notably, Maier et al. (PMID: 20520844) showed that while there is a difference in the pumping rate of 1-day-old wild-type adults fed OP50 or HT115, this difference is abolished in 4-day-old worms, indicating an age- and/or stress-dependent effect of diet on pharyngeal pumping.

Second, our assays on paralysis and pharyngeal pumping were conducted to demonstrate that dietary RNA species can have beneficial effects in wild-type worms and that these effects are not limited to the polyQ40 proteostasis model. We do not claim that pharyngeal pumping is a direct readout of proteostasis, as the reviewer suggests. Furthermore, there seems to be an inconsistency in the evaluation of significance: on one hand, the reviewer suggests that a 20% reduction in expression could substantially impact aggregation, while on the other hand, they dismiss statistically significant improvements in muscle morphology and paralysis as minimal or marginal. Our analysis clearly demonstrates significant differences, reinforcing that the observed effects are not solely due to transgene-specific mechanisms. Expanding the study to include additional mutant strains, as suggested, goes beyond the current scope of our manuscript, as there is no evidence indicating that the effects we observe are transgene-specific.

Finally, in the initial round of reviews, another reviewer specifically requested that we shift our focus from aging to proteostasis, which we implemented. Now, this reviewer suggests shifting the emphasis back to aging. To address this concern while maintaining consistency with prior reviewer feedback, we have refined our framing in this case to emphasize the promotion of healthspan, which encompasses both proteostasis and age-related functional decline.

Regarding point 3:

I do not agree with the authors' interpretation of the results in the provided figure. If we consider there is basal autophagy in OP50-fed animals, then in HT115-fed animals, there appear to be fewer (or even no?) GFP-positive autophagosomes and a similar number of only RFP-positive autolysosomes. Based on these observations, I would interpret this as a reduction in autophagic activity rather than an induction. To my knowledge, autophagic flux is considered increased when there is a rise in both yellow (GFP-RFP colocalized) and red (RFP-only) puncta in cells, while a blockade in autophagic flux is indicated by an increase in yellow puncta without a corresponding rise in red puncta (PMID: 18188003). This interpretation also aligns with the sequencing data, which do not indicate an induction of autophagy.

We observe a lower percentage of yellow puncta and a higher number of red-labeled puncta in HT115-fed animals compared to OP50-fed ones. We express this difference as the relative GFP/mCherry ratio (Fig. 2f). According to PMID: 30355753, unlipidated mRFP-EGFP-LC3 produces a yellow cytoplasmic background, which decreases as autophagy is induced. Upon autophagy induction, autophagosomes rapidly form and fuse with lysosomes, leading to an increase in total vesicles and a higher red-to-yellow vesicle ratio (figure below). Conversely, when autophagic flux is blocked, yellow puncta accumulate due to impaired degradation. Based on these principles, we argue that autophagy is increased in HT115-fed animals.

[FIGURE REDACTED]

Figure from PMID: 30355753

However, we respectfully note that our manuscript does not discuss autophagic flux. Our study measures polyQ40 aggregates in two-day-old adults, while autophagy in muscle was assessed in older worms, as autophagosomes are scarce in wild-type young worms. To directly measure autophagic flux, lysosomal function would need to be disrupted, which would also impact proteostasis. For this reason, we focused on GFP and RFP data and made only mild statements regarding autophagy.

We would like to mention: the guidelines cited by the reviewer (PMID: 18188003) were followed in our seam cell autophagy measurements in younger animals.

Regarding our proteomics data, we agree that no significant increase in autophagy-related proteins was detected. However, we propose that dietary intervention induces a modest increase in autophagy, potentially as part of a stress response that does not require strong transcriptional upregulation. Often, autophagy regulation relies more on increased lipidation of LGG-1 (LC3 equivalent) rather than upregulation of its expression, as discussed in PMID: 30355753.

Collectively, our data, including microscopy in young and older animals and RNAi silencing of key autophagy components, support the involvement of autophagy in diet-dependent aggregate formation. While we do not explicitly measure autophagic flux, our findings indicate that dietary conditions influence autophagy, which in turn impacts protein aggregation dynamics.

Regarding point 5:

The authors show that injection of RNA isolated from either bacterial strain (OP50, OP50(xu363) or HT1115) does lead to a reduction on polyQ40 aggregation in the progeny of injected animals. These results are important and should not be relegated to a supplementary figure. Instead, it should be prominently featured in Figure 3G, replacing the current panel, especially since the existing figure lacks quantification of PolyQ40. Additionally, the accompanying explanation requires revision, as the current wording is unclear and misleading. The data presented highlight an effect on PolyQ40 aggregation in the progeny, which is likely independent of intertissue transport mediated by SID-1. This is because the RNA is already present in the stem cells and is distributed to progenitor cells through cytosolic diffusion rather than intercellular transmission. The claim that they "directly injected the RNA into the tissue" is incorrect. The RNA is injected only into the gonads of the animals, and for an effect to be observed in other tissues, such as muscle cells, intercellular RNA transmission via SID-1 would be required. However, such transmission is likely inefficient for the reasons the authors have described.

We are somewhat confused by this point and the statement regarding the claim that we "directly injected the RNA into the tissue." Nowhere in our manuscript do we make this claim. In both the main text and the Materials and Methods section, we explicitly state that RNA was injected into the gonad, not directly into other tissues. Therefore, we do not fully understand the concern raised here. The only place we use the words "injected directly into the tissue" was in the previous rebuttal letter where we described the literature and not our methodology. We may have not been precise enough in our wording, but we meant injected directly into the gonad to emphasize the injection instead of ingestion. Just to clarify, what we are doing here is providing the RNA species to the progeny, using a protocol that is commonly employed to silence specific genes through RNAi. We apologize, if we have not made this clear enough. Also, since the main figure is already crowded and the conclusion is still the same, we would prefer not to change the order of our figure. However, if the reviewer insists, we are of course obliged to change the figures.

Regarding point 7:

Regarding the authors' finding that diet-dependent differences were observed only in polyQ40 worms and not in WT worms: the gene expression changes could simply be a consequence of reduced polyQ40 aggregation rather than a causative factor. As such, this observation may not provide substantial mechanistic insight. Nevertheless, the authors should explicitly discuss this possibility in the manuscript to allow readers to form their own conclusions.

We thank the reviewer for their suggestion. We now mention this possibility and say, “However, we cannot exclude the possibility that these gene expression changes are merely a consequence of reduced polyQ40 aggregation, rather than a causative factor.”

Regarding point 8. Statistics:

In some experiments, the authors tested as many as 900 worms, which raises concerns about potential oversampling. In contrast, for other assays, they tested 30-40 worms, which seems more appropriate and consistent with standard practices in the field. The authors should clarify the rationale for these discrepancies in sample size.

We generally used sample sizes within the range described by the reviewer. In some cases, certain experimental conditions served as positive controls, which may have increased the number of worms analyzed. We assume the reviewer is particularly concerned about the high numbers in the developmental rate assays, where the total sample size exceeded several hundred worms.

For these assays, we started with approximately 12-16 synchronized, 2-day-old mothers, all raised under identical experimental conditions for over two generations. These mothers were placed on fresh plates for about 3 hours to lay eggs, ensuring a highly synchronized population for analysis. Given this setup, it is expected that a large number of progeny would be analyzed (reaching several hundred). Reducing the number of mothers would increase variability and reduce the robustness of the assay. Therefore, while the final sample size is high, we do not consider this to be oversampling, as it reflects the most accurate approach for assessing developmental rate.

Regarding point 10:

There is no reference to the methylcobalamin experiments in the text.

Since the methylcobalamin experiments were not the main focus of our manuscript, we included them in the Materials and Methods section. There, we reference the control experiments confirming that methylcobalamin is active under our experimental conditions and the data is presented in Fig S1d and e. However, if the reviewer believes these should be mentioned elsewhere in the text, we are happy to consider their suggestions.

Reviewer #3 (Remarks to the Author):

The authors addressed the reviewer’s comments by analyzing the data carefully, supplementing some data and revising the manuscript carefully. The responses and the revised manuscript are reviewed as satisfactory. It seems a good shape for publication in Nature Communication.

Thank you for the positive comments and recommending publication.

Reviewer #4 (Remarks to the Author):

The authors have addressed my concerns. I have no more questions.

Thank you!

Reviewer #2 (Remarks to the Author):

Review of "Bacterial RNA promotes proteostasis through inter-tissue communication in *C. elegans*" by Kyriakakis and colleagues.

I appreciate the authors' efforts to address some of my concerns. However, my main concern remains: There is still no convincing evidence that the bacterial diet directly improves proteostasis of wild-type worms. The observed effects on polyQ40 aggregation could be attributed to interactions with the endogenous RNAi pathway (affecting the transgene and thereby only indirectly proteostasis).

The main concern of this reviewer in the 1st round of reviews was that on HT115 diet polyQ40 was not expressed to the same extent than on OP50 diet, and that therefore we would not see any polyQ40 aggregation in animals on HT115 diet. We quote from the 1st review: "1. My biggest concern is that there is hardly any polyQ40 signal in animals grown on HT115..." In the first revision, we provided ample evidence that this is not the case, and that the polyQ40 expression levels were virtually identical on the different diets. The reviewer then raised the point that the animal's RNAi pathway per se may interfere with the polyQ aggregation. We had already data arguing against such an involvement in the initial manuscript, but still added more data to convince the reviewer. In third version of the manuscript, we showed that not only protein levels remain constant, but also the mRNA levels of the transgene are not altered by the bacterial diet as the reviewer suggested. Still, the reviewer maintains that our data could be explained by the RNAi pathway affecting the transgene. We did the experiments the reviewer asked us to do in the first two rounds, they remain unconvinced - the reason why, remains unclear. Unfortunately, the reviewer did not provide any guidance or suggest any experiments that would convince them.

We would also like to point out that we provide evidence that the bacterial diet of HT115 and OP50(xu363) improves fitness during aging in N2. Together with our data on the improved proteostasis in the proteostasis model, it is not unreasonable to suggest a model in which a similar mechanism is in place in N2 wild-type worms.

Point-by-point response to the authors' rebuttal letter:

Regarding point 1 (proteostasis of WT animals):

The authors acknowledge that the pharyngeal pumping assay is not a direct readout of proteostasis. Therefore, to substantiate the claim that the bacterial diet enhances proteostasis in wild-type animals, the use of mutant strains expressing metastable proteins as folding sensors would be more appropriate. The explanation provided for the high mortality observed with the CB1301 unc-54(e1301) mutant strain (that it might result from the absence of using heat shock) is not entirely convincing. Heat shock would typically exacerbate, rather than alleviate, stress-induced mortality. Other groups, including our own, have successfully used this strain under standard growth conditions. Even if this particular strain was challenging in their hands, the use of an alternative folding sensor strain would have strengthened their conclusions. As it currently stands, the observed effects seem primarily associated with transgenic animals expressing polyQ (or Aβ), which might be

particularly sensitive to influences from the endogenous RNAi machinery.

We respectfully disagree with the reviewer's interpretation 'The authors acknowledge that the pharyngeal pumping assay is not a direct readout of proteostasis.' We never stated that pharyngeal pumping is a direct readout of proteostasis. We always considered it as a readout of fitness, which is commonly accepted in the field.

We used polyQ40 aggregate formation in muscle cells as a proteostasis model. We showed that bacteria lacking rnC activity and therefore producing dsRNA alleviate the symptoms and improved fitness. Moreover, we showed that also in wild-type worms, this diet was beneficial for fitness. The reviewer asked for an additional model, a temperature-sensitive myosin folding mutant to be used. We were happy to comply, got the mutant and performed experiments. To address the reviewer's concerns the best as we could, we choose conditions that would be comparable to our other experiments: growth at 20°C, at least two generations on the same diet to exclude any possible confounding effects from the diet. When we grew the ts myosin mutant (*unc-54(e1301)*) continuously at 20°C, we observed an about 50% diet-dependent mortality rate, indicating that this was already a suboptimal starting point. We performed the experiment nevertheless and concluded that the results are very hard to interpret because of the confounding factors. This is what we already reported in the answer to the reviewers during the first round of revision, in which we included the experimental data. The reviewer remained unconvinced and implied we have a problem in worm husbandry or performing well-reported and sort-of-standard experiments.

We would like to point out that the two publications the reviewer cited during the second review did not state anything about mortality rates of *unc-54(e1301)* mutant animals at 20°C or 25°C. The absence of reporting does not mean that the mortality rate was not high, it solely means the authors did not report it. In PMID: 32217667, the authors say that they selected 35 animals for analysis/experiment. Naturally, they would choose life animals for the analysis. The animals in their analysis expressed polyQ35 in muscle and polyQ44 in the intestine; we use polyQ40-and polyQ24 as control-in muscle. Their biggest effect was on polyQ44 in the intestine and not in muscle. From the description in the paper, it is unclear how the experiment was performed, i.e. at which temperature were the animals raised, at which point and for how long have they been shifted to 25°C. Have they ever been raised at constant 20°C? What seems to be clear, however, is that the animals were raised on OP50 diet and then transferred as L1s onto HT115(DE3) for RNAi experiments. This is clearly something, we cannot do because we are working on the dietary input and the animals need to be raised on the same diet for more than one generation to eliminate any confounding effects. In PMID: 39128917, the authors described strong paralysis already at the permissive temperature was exacerbated at 25°C. The exact temperature was not given, but they reported a shift from 15°C to 25°C, which could be interpreted that the permissive temperature was 15°C. Therefore, obviously there is already big problem in *unc54(e1301)* mutant animals at lower temperatures, they also did not report on mortality, though. However, the authors clearly state that the animals were raised at 15°C and then shifted to 25°C for 24 hrs to measure body bends in liquid in 1-day old adults. As in the other paper,

they shifted the diet from OP50 to HT115 at L1 stage. We also contacted the senior author of this study and asked for guidance. They had never done experiments at constant 20°C or constant HT115 diet. They always shift L3/4 animals from 15°C to 20°C and also changed the diet from OP50 to HT115 when performing RNAi experiments. Moreover, they also already observe egg-laying defects even after the short shift to 20°C, indicating that long-term growth at 20°C is an issue.

The reviewer stated in their review 'Other groups, including our own, have successfully used this strain under standard growth conditions.' What does standard growth conditions mean? For N2, most researchers in the worm field would probably assume that this means growth on agar plates with OP50 at 20°C. However, when using *ts*-mutants this is not as easy. *unc-45(e1301)* animals apparently are raised as permissive temperature as low as 15 °C, while other *ts*-mutant are maintained at either 18°C or at times even at 20°C, and then only the shift to 25°C induces the *ts*-phenotype. As outlined above, we ideally would maintain the *unc-54(e1301)* worms at 20°C on different diets. In our lab settings, this comes at the cost of an about 50% mortality rate. We would be very interested to learn how the reviewer maintains this strain at 20°C and perhaps they could share their mutant with us, as it seems to be more thermotolerant than the mutant used by at least the Hoppe group, who reports strong phenotypes already at the permissive temperature. We might have gotten the same mutant, which is very already very temperature-sensitive at 20°C.

We would also like to point out that seemingly promoting proteostasis under certain instances may not do so, while in other cases it may. For example, in a recent publication it was shown that depletion of HSP-110 prevented alpha-synuclein aggregation. These beneficial effects were overridden by the negative effects of HSP-110 loss on protein folding, which was measured in *unc-54(e1303)* animals after a shift from 15°C to 25°C (PMID: 32449565). We have shown beneficial fitness effects on aging worms on HT115 and OP50(xu363) bacterial diets. Employing more and more proteostasis models, as suggested by the reviewer, will allow us to determine for which proteostasis models our model holds true and for which ones it won't. While this is a very interesting and important point, it goes way beyond the aims of this manuscript.

We find it hard to understand why this reviewer suggests that we are having a problem with our worm husbandry. There are of course differences and lab-to-lab variations in husbandry and feeding plate composition etc, some might be as simple as the water and the agar source. However, our data are robust. We show in N2 and in proteostasis model animals a positive effect on fitness when raised on HT115 or OP50(xu363) bacteria compared to OP50.

For clarity, we appended below how we have responded to the changing points during the revisions.

A. Quantification of polyQ protein levels under all three diets.

Initially, Western blotting was performed using a protocol similar to that used in our proteomic analysis. While the reviewer raised concerns about sonication and centrifugation potentially affecting protein aggregation, we do not see how this would introduce a diet-specific bias. Nevertheless, we repeated the Western blot using alternative conditions: we omitted sonication, centrifugation, analyzed the whole gel (including the stacking region),

and assessed both polyQ40 and polyQ24 levels. Our analysis (below) showed again no significant changes in protein levels.

Western Blot (left) and analysis (right) of polyglutamine in WT, polyQ24::YFP (Q24) and polyQ40::YFP (Q40) expressing worms. Antibodies for YFP and α -tubulin were used. S.e. (short exposure), l.e. (long exposure).

Full blot stained for YFP and long (l.e.) exposure.

B. Quantification of polyQ mRNA levels under all three diets to assess the involvement of endogenous RNAi machinery

We performed RT-PCR as suggested and found no significant differences in polyQ40::YFP mRNA levels between OP50, OP50(xu363) and HT115-fed animals (below), indicating that the polyQ40 levels are not particularly sensitive to potential induction of the endogenous RNAi machinery. Therefore, we do not understand why the reviewer insists that our data could be explained by the transgene being more sensitive to RNAi machinery when fed on HT115 or OP50(xu363). We would like to note that, if anything, a diet-derived RNAi-mediated reduction of polyQ40 expression would represent an intriguing and impactful finding. However, we found no evidence supporting such an effect.

For the second part of the same first point, we were requested to assess the age-dependent paralysis while knocking down autophagy and the RNAi machinery in wt animals and to test temperature sensitive (ts) mutants and the role of diet. We performed the age-dependent paralysis in wt (below left), but also in AM141 (polyQ40 expressing) strain (below right).

Of course as suggested, we also performed a motility assay in the *ts* mutant using the CB1303 strain (*unc-54(e1301)*). These experiments were independently performed by two researchers, one of whom was not involved in this study, to minimize bias, and we used similar conditions and approaches as the rest of the experiments we performed, to be able to directly compare them with the rest of the study. As outlined above and already in the response to the reviewer's comment in the first round, we do not feel comfortable with the experiment given the high mortality rate (about 50%) from the start. If the reviewer insists, we will include the data.

We would like to emphasize that mortality is not used as an "excuse" but as a potential confounding variable. Our goal was to interpret the data transparently and rigorously. All experiments were designed to closely replicate our primary conditions and were conducted with care to reduce bias. Also, we did not claim that heat shock alleviates stress-induced mortality. The reviewer implied that we were not performing the experiments properly because the diet-dependent difference in mortality is not a phenotype that has been described before. For us this is hard to judge, since we have not found any publication using the same conditions. Nevertheless, because we were unsure of the underlying mechanism, we decided not to include those data in our manuscript and instead perform other experiments that would address the reviewer's concern).

Supporting the significance of bacterial RNA we provided the pharyngeal pumping, and muscle morphology, (in addition to motility in advanced ages) all of which showed a diet dependent difference even in wild-type worms, excluding the hypothesis that the effects are only observed in the polyQ40 expressing strain.

For WT N2:

For polyQ40-expressing worms AM141:

For the body wall muscle morphology we used the MAH19 (*rrf-1(pk1417) I; myo-3(st386) V; stEx30 [myo-3p::GFP::myo-3 + rol-6(su1006)]*) strain where the MYO-3 is fused with GFP and expressed under its own promoter, in a *myo-3* mutant background. Muscle morphology of 8-day old worms, was classified into three categories (normal, intermediate and severe phenotype) (below).

After growing the worms on the three diets muscle morphology was assed (below).

We agree with the reviewer that pharyngeal pumping is not a direct measure of proteostasis, but we never claimed otherwise. Our intention was to provide additional functional data on muscle health beyond transgenic aggregation models. Our big picture is, that bacterial-derived RNA species have an impact on worms, in our case a positive one.

We are open to alternative terminology if “proteostasis” is considered inappropriate, although we use polyQ and A β transgenics, well-established proteostasis models in the *C. elegans* community, and we show an increase in autophagy.

Regarding the suggestion to use mutant animals to demonstrate enhanced proteostasis in wild-type animals, we respectfully note that such an approach appears contradictory, as mutant animals are by definition not wild-type.

Regarding point 2 (Interpretation of the autophagy reporter):

I respectfully disagree with the authors' interpretation. To distinguish between induction and blockage of autophagy, measurement of autophagic flux would be necessary, which has not been performed. Therefore, the conclusion that the bacterial diet decreases polyQ aggregation via enhanced autophagy is not sufficiently supported. The fact that KD of autophagy genes increases polyQ aggregation alone does not allow such a conclusion.

As stated in our previous responses, we have performed autophagy measurements using two well-established approaches in *C. elegans*: (1) quantifying GFP::LGG-1-labeled autophagosomes in seam cells, and (2) utilizing a dual-fluorescent LGG-1 reporter to differentiate autophagosomes from autolysosomes. These methods are commonly used in the field to assess autophagic activity in *C. elegans* (PMID: 39070663, PMID: 18188003, PMID: 30355753). It is important to note that in our manuscript, we were very cautious and avoided making definitive claims about autophagic flux.

To measure autophagic flux more directly, we would have to inject worms with bafilomycin A1. We decided not to use bafilomycin in this study, primarily because at the specific adult age (8-day old) at which we assess autophagy, bafilomycin has been reported (PMID: 28675140) to have minimal impact on autophagic flux in muscles (below). Therefore, this experiment would not add anything to the manuscript. Because injections are stressful for the worms, this procedure may rather introduce artefacts and make proper interpretation challenging.

In our manuscript, we mention that we observe a lower percentage of yellow puncta and a higher number of red-labeled puncta in HT115-fed animals compared to OP50-fed ones. We express this difference as the relative GFP/mCherry ratio, which according to PMID: 30355753 (below) would suggest that autophagy is increased in HT115-fed animals or at least that there is a difference in autophagy when different diets are used.

[FIGURE REDACTED]

Finally, in the first review round, the reviewer's suggestion was to either (1) interpret our findings more cautiously or (2) perform autophagy flux measurements and assess lysosomal

degradation. We actually did both: we toned down our interpretation and also performed new experiments, which more accurately measure autophagy. We believe our experimentation and interpretation of the data are consistent with the standard in the field.

Regarding point 5 (dsRNA uptake and SID-1 function):

The authors have misunderstood my original comment. In the manuscript, they state: "We assume that the size of the dsRNA species matters for the uptake and processing from the intestinal cells. Previous work showed that ingestion of long dsRNA is more effective than shorter pieces, but the length of dsRNA had no effect when injected into the gonad (33, 34)." The papers they cite in Ref. 33 and 34 found that the dsRNA transporter SID-1 transports more efficiently long dsRNA. However, both feeding and gonadal injection require SID-1 for the transport of dsRNA into muscle cells expressing polyQ40. Thus, the difference the authors observe cannot be explained solely by differences in dsRNA lengths.

We would like to cite the initial point 5 of reviewer 2: '5. Figure 3G: Quantification of polyQ40 aggregation is missing. To substantiate their claim, the authors should also inject RNA isolated from OP50 and OP50(xu363) and examine the effects on polyQ40 aggregation.'

We performed the requested experiments and also did the quantification. We showed that also RNA isolated from OP50 had a beneficial effect. We argued that we cannot exclude basepairing at one point during the RNA isolation and injection. Alternatively, we proposed that the lengths of dsRNA might play role. These explanations were just meant to provide possible scenarios for our observed results.

Then in the second review the Reviewer 2 assumes that the SID-1 transporter, responsible for intercellular RNA transport, is also responsible for the import of dsRNA into muscle cells and that this then would lead to the reduction in polyQ40 aggregates in muscle. This is not what we suggest. We do not have any direct evidence that this is the case. In contrast, our RNAseq of bacterial RNA and the lack of evidence that those bacterial RNA could act as siRNAs would rather suggest a different mechanism. Moreover, our *C. elegans* proteome data failed to detect any strong decrease in specific proteins on HT115 or OP50(xu363) diets compared to OP50.

We report that RNA from all three bacterial strains reduced aggregate formation in polyQ40 animals when injected into the gonad. We also provided evidence that the signal(s), which prevented polyQ40 aggregation on HT115 and OP50(xu363) diets, should involve the germline as germline RNAi-incompetent animals did not reduce aggregate formation, independent of the bacterial diet. The nature of the signal from the germline to the muscle remains elusive. We do not even know whether the bacterially derived dsRNA is transported into body-wall muscles or whether dsRNA length is a discriminator. Therefore, we are agnostic about a possible role of SID-1 in the process. While these are intriguing possibilities, their demonstration is very challenging to impossible at this point and clearly beyond the scope of this paper.

In the manuscript, we do not make any strong point about the nature of the signal because we simply do not know it. Our current and future work is geared towards understanding the

precise mechanism of how bacterially-derived RNA provides beneficial effects on fitness or healthspan through the proteostasis system to maintain homeostasis. Clearly, the finding the signal(s) is a major undertaking that is going well beyond what we can achieve at this point.

Regarding point 8 (sample size and statistical power):

Although there were many progenies on the plates, the authors could have randomly selected fewer animals for the analysis. A smaller sample size, would have allowed for more meaningful statistics.

In the previous round the reviewer was concerned about oversampling, and our previous explanation why we did the experiment the way we did it, apparently did not satisfy the reviewer. First of all, eliminating data points is not something we would ever like to do. Second, oversampling would be if we would have samples of very different sizes in which, for example, sample A would have many more data points than sample B. This is, however, not the case. It is true that in case when we counted progeny and their development the sample sizes are in the hundreds, while for other experiments like pharyngeal pumping, we tested 30-40 animals. In our lifespans, we start with 120 animals. Nevertheless, for each experiment we used comparable sample sizes. We did not compare 30 offspring to 900 offspring, which would be oversampling. Big sample sizes are not same as oversampling in statistical terms. Our sample sizes are likewise not so big that the outcome was statistically significant but not biologically significant.

We would like to respectfully clarify once again our rationale regarding the sample sizes used in the developmental rate assay. The higher sample size was used only and specifically for the developmental rate assay, where we analyzed the progeny of a small number (12–16) of synchronized, 2-day-old mothers. These mothers were allowed to lay eggs over a short 3-hour window to ensure tight synchronization. This approach naturally yields a large number of eggs, which we then analyzed.

We would also like to bring to the reviewer's attention that we are not the first to report this type of experiment. Maier *et al.*, also determined progeny development on OP50 and HT115 (see below) (PMID: 20520844). They use comparable numbers and the outcome was very similar. Therefore, we think our experiment, the analysis and the statistical testing was appropriate.

[FIGURE REDACTED]

When the goal is to calculate proportions (e.g., percentage of animals reaching a certain developmental stage), larger sample sizes improve statistical power and yield more accurate estimates. We also find it not scientifically sound to artificially reduce our data points. Finally, repeating the analysis with fewer worms and offspring does not change anything about the outcome and the interpretation of this experiment, which anyhow only provides supportive information.

We hope this clarifies why we used for this specific assay a higher number to perform the analysis and why it does not consist an oversampling problem.